# Ethylene-triggered subcellular trafficking of CTR1 enhances the response to ethylene gas

Hye Lin Park[1,2,8], Dong Hye Seo[1,2,5,8], Han Yong Lee[1,2,6,8], Arkadipta Bakshi[3,7], Chanung Park [1,2], Yuan-Chi Chien [1,2], Joseph J. Kieber[4], Brad M. Binder[3] & Gyeong Mee Yoon [1,2] ✉

The phytohormone ethylene controls plant growth and stress responses. Ethylene-exposed dark-grown Arabidopsis seedlings exhibit dramatic growth reduction, yet the seedlings rapidly return to the basal growth rate when ethylene gas is removed. However, the underlying mechanism governing this acclimation of dark-grown seedlings to ethylene remains enigmatic. Here, we report that ethylene triggers the translocation of the Raf-like protein kinase CONSTITUTIVE TRIPLE RESPONSE1 (CTR1), a negative regulator of ethylene signaling, from the endoplasmic reticulum to the nucleus. Nuclear-localized CTR1 stabilizes the ETHYLENE-INSENSITIVE3 (EIN3) transcription factor by interacting with and inhibiting EIN3-BINDING F-box (EBF) proteins, thus enhancing the ethylene response and delaying growth recovery. Furthermore, Arabidopsis plants with enhanced nuclear-localized CTR1 exhibited improved tolerance to drought and salinity stress. These findings uncover a mechanism of the ethylene signaling pathway that links the spatiotemporal dynamics of cellular signaling components to physiological responses.

The ability of organisms to respond to and integrate environmental signals into the cellsis critical for optimal growth and development, particularly for plants, which are non-motile. Plants adapt to a wide variety of abiotic stresses, and after the removal of stress, they need to rapidly restore basal cellular homeostasis. One key signal for the acclimation of plants to abiotic stress is the plant hormone ethylene. Ethylene regulates multiple aspects of growth and development, including fruit ripening, leaf and floral senescence, cell elongation, seed germination, root hair formation, and responses to biotic and abiotic stress[1–3]. Ethylene-mediated stress acclimation includes, but is not limited to, the rapid elongation of rice internodes in response to flooding, salt tolerance, heavy metal tolerance, and morphological changes in roots in response to nutrient deficiency[4–8].

Extensive molecular genetic studies have elucidated the basic ethylene signaling pathway[2,9–12]. In the absence of ethylene, the endoplasmic reticulum (ER)-localized ethylene receptors activate CONSTITUTIVE TRIPLE RESPONSE1 (CTR1) protein kinase, which in turn phosphorylates ETHYLENE INSENSITIVE 2 (EIN2), an ER membrane-localized Nramp homolog that positively regulates ethylene responses. CTR1-mediated phosphorylation of EIN2 prevents EIN2 from signaling in the absence of ethylene[13–15]. In response to ethylene, the receptors and hence CTR1 are inactivated, leading to reduced phosphorylation and increased accumulation of EIN2. EIN2 is then proteolytically cleaved, and the C-terminal domain (EIN2-CEND) is released. EIN2-CEND translocates into the nucleus and indirectly activates the ETHYLENE INSENSITIVE 3 (EIN3) and EIN3-like (EIL) paralogs, which are central transcription factors in ethylene signaling[13–15]. Full-length of EIN2 may also localize to the nucleus[16]. EIN2-CEND also associates with the mRNAs of two E3 ligase components, EIN3-Binding F-box 1 (EBF1) and EBF2, in the processing body, and subsequently

[1]Department of Botany and Plant Pathology, Purdue University, West Lafayette, IN 47907, USA. [2]Center for Plant Biology, Purdue University, West Lafayette, IN 47907, USA. [3]Department of Biochemistry & Cellular and Molecular Biology, University of Tennessee, Knoxville, TN 37996, USA. [4]Department of Biology, University of North Carolina, Chapel Hill, NC 27599, USA. [5]Present address: Department of Systems Biology, Yonsei University, Seoul 03722, Korea. [6]Present address: Department of Biology, Chosun University, Gwangju 61452, Korea. [7]Present address: Department of Botany, UW-Madison, Madison, WI, USA. [8]These authors contributed equally: Hye Lin Park, Dong Hye Seo, Han Yong Lee. ✉e-mail: yoong@purdue.edu

represses their translation, thus ultimately blocking the degradation of EIN3/EIL proteins[17,18]. In the nucleus, EBF1 and EBF2 degrade EIN3 via the 26S proteasome[19]. The function of CTR1 beyond phosphorylating EIN2 at the ER, if any, has not been characterized.

Hypocotyls of dark-grown seedlings exposed to ethylene show a dramatically diminished growth rate. However, after the removal of ethylene, seedlings return to a basal growth rate within 90 min[20,21], although the proteolytic cleavage of EIN2 is irreversible. Increased levels of negative regulators of ethylene signaling (the receptors and CTR1) as well as a receptor-clustering model have been suggested to play a role in the rapid return to basal growth rates after removal of ethylene[22–26], though additional mechanisms may exist. Here, we report that CTR1 rapidly translocates from the ER to the nucleus in response to ethylene. Unexpectedly, the nuclear-localized CTR1 enhances ethylene responses through a mechanism that does not require its kinase activity. This inhibits the fast recovery of seedling growth back to basal levels and enhances salt and drought tolerance. These results suggest a new paradigm for the dynamic regulation of the ethylene signaling involving the translocation of CTR1 to the nucleus, thereby strengthening EIN2-mediated EIN3 activation in the nucleus.

## Results

### Ethylene-induced ER-to-nucleus translocation of CTR1

CTR1 consists of an N-terminal regulatory domain and a C-terminal kinase domain that is homologous to the catalytic domain of the Raf kinase family. CTR1 lacks any canonical organelle targeting sequences, including nuclear localization sequences (NLSs)[11] (Fig. 1a). To determine the role of CTR1 beyond its regulation of EIN2, we examined the subcellular localization of CTR1 after plant exposure to exogenous ethylene in the dark. To this end, we introduced a -7.6 Kb genomic *CTR1* transgene containing a *CTR1* promoter region driving expression of the CTR1 coding region fused to a GFP reporter (*CTR1p:GFP-gCTR1*)[11]. The *CTR1p:GFP-gCTR1* transgene fully complemented *ctr1-2* (a null allele) in both light- and dark-grown seedlings, including decreased rosette and inflorescence size, and the triple response phenotype (Fig. 1b and Supplementary Fig. 1), indicating that the fusion protein was functional. In the presence of an inhibitor of ethylene perception, silver nitrate ($AgNO_3$), or in an ethylene-insensitive *etr1-1* mutant background, the GFP-CTR1 fusion protein localized to the ER (Fig. 1c, d), in agreement with findings from previous reports[27,28]. Unexpectedly, in response to either 1-aminocyclopropane-1-carboxylate (ACC, a direct precursor of ethylene) or exogenous ethylene, GFP-CTR1 accumulated in the nucleus (Fig. 1c, e, f). A low level of nuclear-localized GFP-CTR1 was occasionally detected in a small number of cells in dark-grown seedlings in the absence of exogenous ethylene (Supplementary Fig. 2a), likely reflecting basal levels of ethylene. Disruption of either *EIN2* or *EIN3/EIL1* did not prevent the nuclear accumulation of GFP-CTR1 (Fig. 1g), suggesting that EIN2 and EIN3 are not required for CTR1 nuclear translocation. Furthermore, when GFP-CTR1 was expressed under the control of the strong CaMV 35S promoter (Supplementary Fig. 3), a large fraction of GFP-CTR1 localized to the nucleus in etiolated seedlings even in the absence of exogenous ACC (Fig. 1h and Supplementary Fig. 2b), presumably because of the limited number of ethylene receptors tethering CTR1 to the ER. Consistent with prior findings, ACC treatment increased the stability of GFP-CTR1 protein (Supplementary Fig. 4). Given the nuclear localization of GFP-CTR1 in the overexpression lines in the presence of silver nitrate, ACC-induced CTR1 stabilization may promote CTR1 nuclear translocation once CTR1 protein abundance reaches the limit at which it can be bound by ethylene receptors at the ER. Fractionation analyses confirmed that both GFP-CTR1 and endogenous CTR1 in *CTR1p:GFP-gCTR1/ctr1-2* and wild-type seedlings, respectively, were enriched in the nuclear fraction in extracts of ACC-treated seedlings, but remained in cytosol in extracts derived from seedlings grown in the presence of $AgNO_3$ (Fig. 1i, j).

ACC is often used to induce the ethylene response in plants. However, recent studies indicate that ACC acts independently of ethylene as a signaling molecule in multiple processes, including cell wall function[29], guard cell differentiation[30], pollen tube attraction[31], growth of thalli and rhizoids[32], and pathogen interactions[33,34]. In our study, both ACC and ethylene caused equivalent CTR1 nuclear translocation (Fig. 1f); thus, we used ACC for further studies.

EIN2-CEND migrates into the nucleus within 10 min after ethylene treatment[14]. We monitored the dynamics of CTR1 movement in response to increasing ACC treatment duration (Fig. 1k and Supplementary Fig. 5). GFP-CTR1 first accumulated to detectable levels in the nucleus 30 min after ACC treatment, and a further increase in nuclear protein levels was observed after an additional 30 min. However, in the absence of ethylene, the abundance of GFP-CTR1 in both the cytoplasm and nucleus rapidly decreased. The GFP fluorescence of nuclear- and ER- localized GFP-CTR1 was gradually reduced until 40 min after ethylene removal, but by 60 min after ethylene removal, little detectable fluorescence remained in both the ER and nucleus (Fig. 1l), consistent with ACC-induced stabilization of GFP-CTR1. Corroborating this, the steady-state levels of GFP-CTR1 were gradually decreased after ethylene removal (Supplementary Fig. 6). Together, these findings revealed that ethylene stimulates the translocation of CTR1 from the ER to the nucleus in an EIN2 and EIN3/EIL independent manner, and that nuclear CTR1 protein is either degraded via an unknown mechanism or may be exported back to the cytoplasm after ethylene removal.

### CTR1 nuclear trafficking is regulated through the CTR1 N-terminus

CTR1 is recruited to the ER via interaction with ethylene receptors, where it acts to prevent ethylene signaling by phosphorylation of EIN2[11,13,35]. Nuclear translocation of CTR1 requires the dissociation of CTR1 from the ER and presumably disassociation from the ethylene receptors. CTR1 interacts with the ethylene receptor via its N-terminal domain[27]. Therefore, we examined whether the N-terminal domain of CTR1 might affect its nuclear translocation. To test this possibility, we expressed a fusion protein of the CTR1 lacking the N-terminal domain (ΔNT-CTR1) (Fig. 1a) from its native promoter in stable transgenic plants. The *CTR1p:GFP-ΔNT-gCTR1* transgene did not rescue *ctr1-2* in either etiolated or light-grown plants (Fig. 2a and Supplementary Fig. 1). Consistent with this finding, *CTR1p:GFP-ΔNT-gCTR1* seedlings showed constitutive expression of the ethylene-inducible *ERF1* gene, to a level comparable to that observed in *ctr1-2* (Fig. 2b). The failure of the *ΔNT-gCTR1* transgene to complement *ctr1-2* likely resulted from the decreased ER membrane targeting of the ΔNT-CTR1 protein. Indeed, the ΔNT-CTR1 protein was constitutively localized to the nucleus in the dark-grown seedlings, whether expressed from its own, or the CaMV 35S promoter (Fig. 2c, d, and Supplementary Figs. 2c–d and 7).

We further explored whether binding of CTR1 to the ethylene receptors might play a role in its nuclear localization. Previous studies have shown that the *ctr1-8* mutation (G354E) blocks the interaction of CTR1 with ETR1 in yeast-two-hybrid assays but does not affect the intrinsic kinase activity of the protein (Fig. 1a)[36]. In agreement with the weak hypermorphic nature of *ctr1-8*, the *GFP-CTR1^{ctr1-8}* transgene partially complemented *ctr1-2* in both light- and dark-grown seedlings (Fig. 2e and Supplementary Fig. 1). Previous fractionation studies have demonstrated that the *ctr1-8* mutant protein is found primarily in the soluble fraction, and a minor portion remains associated with the membrane fraction[28], in contrast to the predominant ER localization of wild-type CTR1. However, we observed that a large fraction of GFP-CTR1-8 still appeared to localize to the ER (Fig. 2f). We speculate that this finding might have been due to weak association of the CTR1-8 protein with the receptors, thus resulting in a rapid equilibrium being reached between the receptor-bound and unbound states of CTR1-8. Unexpectedly, GFP-CTR1-8 did not translocate to the nucleus in either

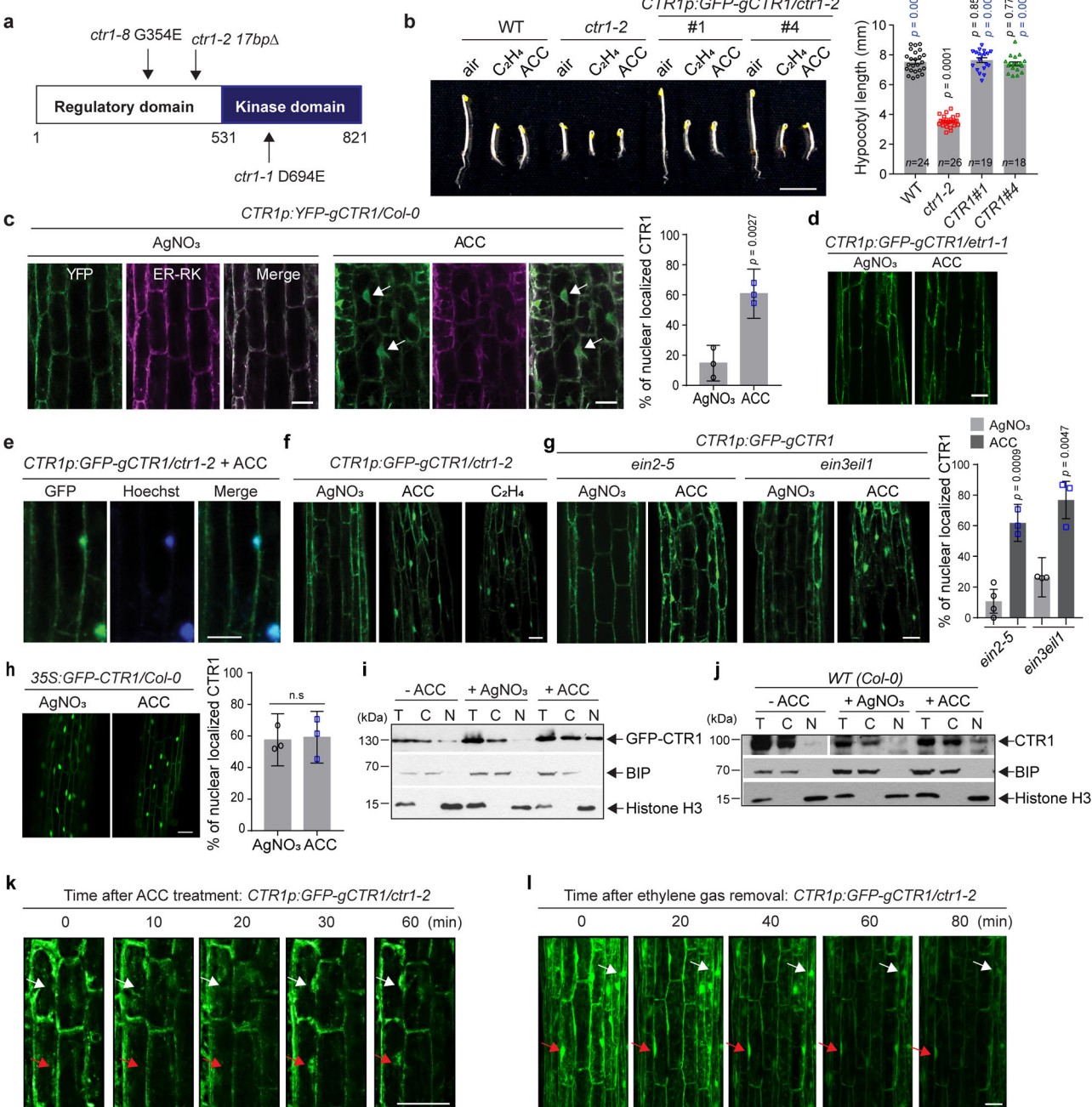

**Fig. 1 | Ethylene-activated CTR1 translocation to the nucleus does not require EIN2 and EIN3. a** Diagram of the CTR1 protein and the positions of *ctr1* mutant alleles. **b** The GFP-fused wild-type genomic *CTR1* fragment fully rescues *ctr1-2* and confers an ethylene response. Seedlings were grown for 3 d in the dark with or without ACC or ethylene. The graph represents the quantification of hypocotyl lengths of seedlings grown on MS without ACC. MS, Murashige and Skoog medium. Significance was determined by one-way ANOVA with Dunnett's multiple comparisons test to compare the results to the WT (black) and *ctr1-2* (blue) controls. Error bars, SE. Scale bar, 5 mm. **c** YFP-CTR1 co-localizes with ER-RK and translocates into the nucleus in the presence of ACC. The areas below the hook and above the elongation zone were imaged. ER-RK[62], an RFP-fused ER marker. Arrows indicate nuclear-localized YFP-CTR1. The graph represents the ratio of nuclear-localized CTR1 in dark-grown seedlings. Error bars, SE (*n* = 3 seedlings). **d** GFP-CTR1 does not translocate into the nucleus in *etr1-1* mutant. **e** GFP-CTR1 fluorescence overlaps with Hoechst nuclear staining under ACC treatment. **f** Ethylene and ACC activate CTR1 nuclear translocation. **g** Seedlings expressing GFP-CTR1 in *ein2-5* or *ein3eil1*

mutants were grown on MS medium with or without AgNO₃ or treated with ACC for 2 h. Error bars, SE (*n* = 3–4 seedlings). **h** Overexpression of CTR1 from 35S promoter leads to nuclear localization of CTR1 in the presence of silver nitrate. Error bars, SE (*n* = 3 seedlings). **i–j** Total protein extract (T) of seedlings treated with or without ACC (200 μM) or grown on media with silver nitrate (100 μM) was fractionated into nuclear (N) and cytoplasmic (C) fractions, followed by immunoblotting with anti-GFP, -BIP, and -CTR1, and anti-Histone H3 antibodies. **k** Time-lapse image series of hypocotyl cells expressing GFP-CTR1 in 3-d-old etiolated *CTR1p:GFP-gCTR1/ctr1-2* seedlings after exposure to 200 μM ACC. **l** GFP-CTR1 fluorescence decreased after ethylene gas removal. *CTR1p:GFP-gCTR1/ctr1-2* seedlings were pretreated with 10 ppm ethylene gas and imaged different time points after the ethylene removal. All scale bars represent 50 μm, except the scale bar in (**b**). The presented images at each time point (**k–l**) are merged Z-stack images that combine 10 successive Z-stack images with 2 μm intervals. All imaging was repeated at least of three times with similar results. Two-tailed Student's *t* test was performed to determine statistical significance for (**c**), (**g**), and (**h**). n. s, statistically not significant.

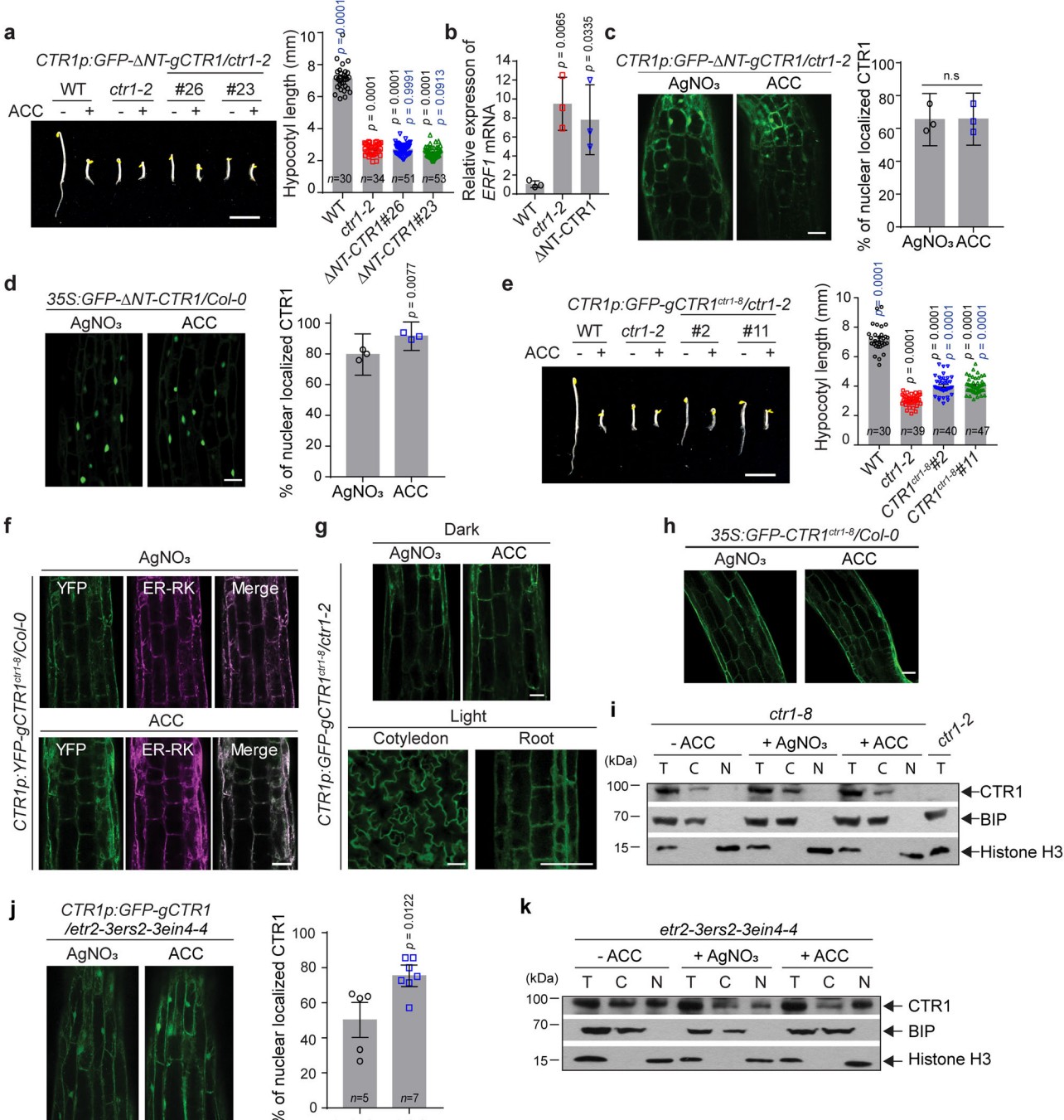

**Fig. 2 | The N-terminus of CTR1 inhibits ACC-induced CTR1 nuclear movement.** **a** Seedlings were grown on MS medium with or without 10 μM ACC for 3 d and photographed. The graph shows quantification of the hypocotyl lengths of seedlings grown on MS without ACC. Scale bar, 5 mm. Significance was determined by one-way ANOVA with Dunnett's multiple comparisons test to compare the results to the WT (black) and *ctr1-2* (blue) controls. Error bars, SE. **b** Quantitative gene expression analysis of ethylene-responsive *ERF1* in *ctr1-2* and *CTR1p:GFP-ΔNT-gCTR1* seedlings. Expression was normalized to an *Actin* control and is presented relative to the WT control. Error bars, SD ($n = 3$ biological replicates). **c** Constitutive nuclear localization of GFP-ΔNT-CTR1 expressed from the native promoter in the hypocotyls of dark-grown seedlings. The graph represents the ratio of nuclear-localized CTR1 in dark-grown seedlings treated with ACC or AgNO₃. Error bars, SE ($n = 3$ seedlings). **d** Constitutive nuclear localization of GFP-ΔNT-CTR1 expressed from the CaMV 35S promoter in 3-d-old dark-grown. The graph represents the ratio of nuclear-localized CTR1 in dark-grown seedlings treated with ACC or AgNO₃.

Error bars, SE ($n = 3$ seedlings). **e** Seedlings expressing GFP-CTR1^ctr1-8 from the native promoter were grown on MS medium with or without 10 μM ACC for 3 d and photographed. The graph represents the quantification of hypocotyl lengths of seedlings grown on MS without ACC. Scale bar, 5 mm. One-way ANOVA with Dunnett's multiple comparisons test to compare the results to the WT (black) and *ctr1-2* (blue) controls. Error bars, SE. **f** Seedlings co-expressing YFP-CTR1-8 and ER-RK were grown in the dark and used to visualize co-localization. **g–h** GFP-CTR1-8 does not translocate into the nucleus in both dark and light conditions. **i** Fractionation analysis of *ctr1-8* mutant seedlings treated with or without ACC and silver nitrate. **j** Constitutive nuclear localization of GFP-CTR1 in the *etr2-3ers2-3ein4-4* mutant background in the dark-grown seedlings. Error bars, SE. **k** Fractionation analysis of *etr2-3ers2-3ein4-4* mutant seedlings treated with or without ACC and silver nitrate. Scale bars in (**c**, **d**, **f**, **g**, **h**, **j**) represent 50 μm. All imaging was repeated at least of three times with similar results. Two-tailed Student's *t* test was performed to determine statistical significance for (**b**, **c**, **d**, **j**). n.s, statistically not significant.

etiolated or light-grown seedlings, regardless of ACC treatment (Fig. 2f–h). The compatible levels of GFP-CTR1-8 protein to WT CTR1 protein indicates that the lack of GFP-CTR1-8 in the nucleus was not the result of reduced levels of GFP-CTR1-8 protein (Supplementary Figs. 2e, f, 7). In agreement with these results, fractionation studies showed that endogenous CTR1-8 is absent in the nucleus in the presence of ACC or silver nitrate (Fig. 2i). To further examine the role of CTR1's interaction with ethylene receptors in regulating its nuclear translocation, we examined the localization of GFP-CTR1 in various ethylene receptor mutant backgrounds. GFP-CTR1 constitutively localized to the nucleus in an *etr2-3ers2-3ein4-4* triple mutant (Fig. 2j), in which three of the five ethylene receptors were disrupted, as well as in other multiple loss-of-function receptor mutants (Supplemental Fig. 8). Fractionation studies further demonstrated that CTR1 is constitutively localized to the nucleus in the *etr2-3ers2-3ein4-4* mutant (Fig. 2k). These results suggest that in the absence of ethylene, ethylene receptors likely tether CTR1 to the ER via direct interaction, thus preventing CTR1 nuclear translocation.

## The nuclear movement of CTR1 is independent of kinase activity

To address the role of kinase activity in CTR1 localization, we expressed a catalytically inactive CTR1 mutant (*CTR1p:GFP-gCTR1^ctr1-1*) in the *ctr1-2* mutant background (Fig. 1a)[36]. The *GFP-CTR1^ctr1-1* transgene did not rescue *ctr1-2* in either the dark or light conditions (Fig. 3a and Supplementary Fig. 1) consistent with its strongly hypomorphic nature. Similar to the full-length wild-type GFP-CTR1, full-length GFP-CTR1-1 expressed from its native promoter translocated to the nucleus after ACC treatment (Fig. 3b). Fractionation studies confirmed that ACC treatment promotes the nuclear localization of CTR1-1 protein (Fig. 3c). When expressed from the CaMV 35S promoter, GFP-CTR1-1 was constitutively localized in the nucleus, similarly to wild-type GFP-CTR1 (Fig. 3d). In addition, inactive ΔNT-CTR1-1 expressed from either the native promoter or the CaMV 35S (*CTR1p:GFP-ΔNT-gCTR1^ctr1-1/ctr1-2* and *35S:GFP-ΔNT-CTR1^ctr1-1/Col-0*) constitutively localized to the nucleus (Fig. 3e, f), and this *ΔNT-CTR^ctr1-1* transgene did not rescue *ctr1-2* in either dark- or light-grown seedlings (Supplementary Figs. 1 and 9).

ΔNT-CTR1 autophosphorylates on four residues (S703/T704/S707/S710) within the kinase activation loop, and this autophosphorylation is critical for CTR1 kinase activity and homodimer formation[37]. To test the role of this autophosphorylation, we altered three of these target S/T residues (T704/S707/S710) to Ala (CTR1^AAA), which has been shown to disrupt homodimer formation[37]. We confirmed that ΔNT-CTR1^AAA was catalytically inactive toward the EIN2 substrate; wild-type ΔNT-CTR1 but not ΔNT-CTR1^AAA phosphorylated EIN2-CEND when co-expressed in *Arabidopsis* mesophyll protoplasts (Fig. 3g). Whereas wild-type ΔNT-CTR1 interacted with itself in a yeast-two-hybrid assay, the ΔNT-CTR1^AAA did not interact with itself (Fig. 3h). Similar to wild-type ΔNT-CTR1, ΔNT-CTR1^AAA constitutively localized to the nucleus (Fig. 3i). Together, these results indicate that kinase activity and probably homodimerization are not required for CTR1 nuclear translocation.

## Nuclear-localized CTR1 delays growth recovery of seedlings after ethylene-induced growth inhibition

Both the *ctr1-1* and *ctr1-8* mutants show constitutive ethylene responses[36], despite the observation that CTR1-1 and CTR1-8 proteins showed different nuclear translocation responses to ethylene (Figs. 2f–h, 3b, d). This indicates that CTR1 nuclear movement does not control the primary ethylene response but rather may influence ethylene response kinetics by fine-tuning nuclear ethylene signaling. To test this hypothesis, we measured the ethylene growth response kinetics of the hypocotyls of etiolated seedlings, which have been widely exploited to analyze the kinetics of various ethylene mutants[20,38]. There are two phases of growth inhibition of wild-type *Arabidopsis* hypocotyls in response to ethylene. Phase I begins 10 min

after ethylene treatment and is characterized by a rapid deceleration in growth rate. After a transient (15 min) plateau in the growth rate, phase II growth inhibition is initiated, thus resulting in further growth suppression lasting 30 min until the growth rate reaches a new low steady-state rate[20,38]. Genetic studies have revealed that EIN2 is necessary for both phases, but EIN3/EIL1 is only required for phase II[22]. Interestingly, after removal of ethylene during phase II, hypocotyl growth rapidly recovers to the pre-treatment growth rate within 90 min[20,38], indicating the existence of a mechanism to rapidly shut off the ethylene response.

To examine whether nuclear-localized CTR1 might play a role in the rapid growth inhibition when ethylene is added or in the recovery kinetics when ethylene is removed, we performed time-lapse analyses of the ethylene response growth kinetics of seedlings overexpressing GFP-ΔNT-CTR1 (*35S:GFP-ΔNT-CTR1*) and GFP-ΔNT-CTR1-1 (*35S:GFP-ΔNT-CTR1^ctr1-1*), both of which showed strong constitutive nuclear localization and exhibited an ethylene response in the dark (Figs. 2d, 3f). After ethylene exposure, both *35S:GFP-ΔNT-CTR1* and *35S:GFP-ΔNT-CTR1^ctr1-1* seedlings had similar onset and strength of phase I and II growth inhibition to those in the wild-type seedlings. However, the recovery of the hypocotyl growth rate of both *35S:GFP-ΔNT-CTR1* and *35S:GFP-ΔNT-CTR1^ctr1-1* seedlings after ethylene removal was slower than that of the wild type (Fig. 4a and Supplementary Fig. 10a, b). Similar kinetics of ethylene response and growth recovery was also observed in seedlings expressing active or inactive wild-type full-length CTR1 (*35S:GFP-CTR1* and *35S:GFP-CTR1^ctr1-1*), both of which showed some level of constitutive CTR1 nuclear localization (Figs. 1h, 3d, 4b, and Supplementary Fig. 10c, d). The quantification of the intrinsic growth rate of seedlings expressing active or inactive full-length CTR1 or truncated CTR1 in air showed that all have a similar growth rate as compared to wild-type seedlings, indicating that the slower growth recovery of the seedlings is not the result of differences in the basal growth rate of the seedlings (Supplementary Tables 1, 2). The delayed hypocotyl growth recovery in *35S:GFP-ΔNT-CTR1*, *35S:GFP-ΔNT-CTR1^ctr1-1*, *35S:GFP-CTR1* and *35S:GFP-CTR1^ctr1-1* resembled that reported in previous studies with EIN3 overexpression or loss of EBF2, which results in higher EIN3 levels[21].

The *ctr1-8* mutant recovered approximately 90 min later than the wild type after the removal of ethylene (Fig. 4c). This result was opposite to our expectations, because the CTR1-8 protein does not translocate to the nucleus, unlike ΔNT-CTR1 and ΔNT-CTR1-1 proteins. However, given the hypermorphic nature of the mutation, the slower growth recovery of the *ctr1-8* mutant is likely attributable to its weak interaction with the receptors, thus resulting in decreased EIN2 phosphorylation and consequently enhanced EIN3 levels. To further investigate the role of CTR1 in the recovery kinetics, we introduced a wild-type full-length genomic *CTR1* fragment into the *ctr1-8* mutant (*CTR1p:GFP-gCTR1/ctr1-8*). The *GFP-gCTR1* transgene rescued the phenotypes of *ctr1-8* in the dark (Supplementary Fig. 11) and restored the slower growth recovery kinetics of *ctr1-8* to levels comparable to those of the wild type (Fig. 4c, Supplementary Tables 1, 2), thus confirming that the mutant CTR1 is responsible for the slower recovery kinetics in *ctr1-8* (Fig. 4c). This result is consistent with the observation that a *GFP-gCTR1^ctr1-8* transgene expressed from the native CTR1 promoter in *ctr1-2* did not confer a wild-type-like growth recovery, in contrast to a WT *GFP-gCTR1* transgene (Supplementary Fig. 12). Next, we examined the correlation between CTR1 nuclear translocation and ethylene growth recovery. To this end, we generated a transgenic line expressing simian virus (SV40) NLS-fused CTR1-8 protein in a *ctr1-8* mutant background (*CTR1p:GFP-gCTR1^ctr1-8-SV40/ctr1-8*) to determine whether enhanced levels of nuclear-localized CTR1 might affect the slower recovery kinetics of the *ctr1-8* mutant. The *GFP-gCTR1^ctr1-8-SV40* transgene did not complement *ctr1-8* and exhibited a comparable ACC response to that of *ctr1-8* (Supplementary Fig. 11). The addition of SV40-NLS to CTR1-8 also did not change the function of CTR1-8 in ethylene response kinetics (Supplementary Fig. 13). As expected, GFP-CTR1-8-

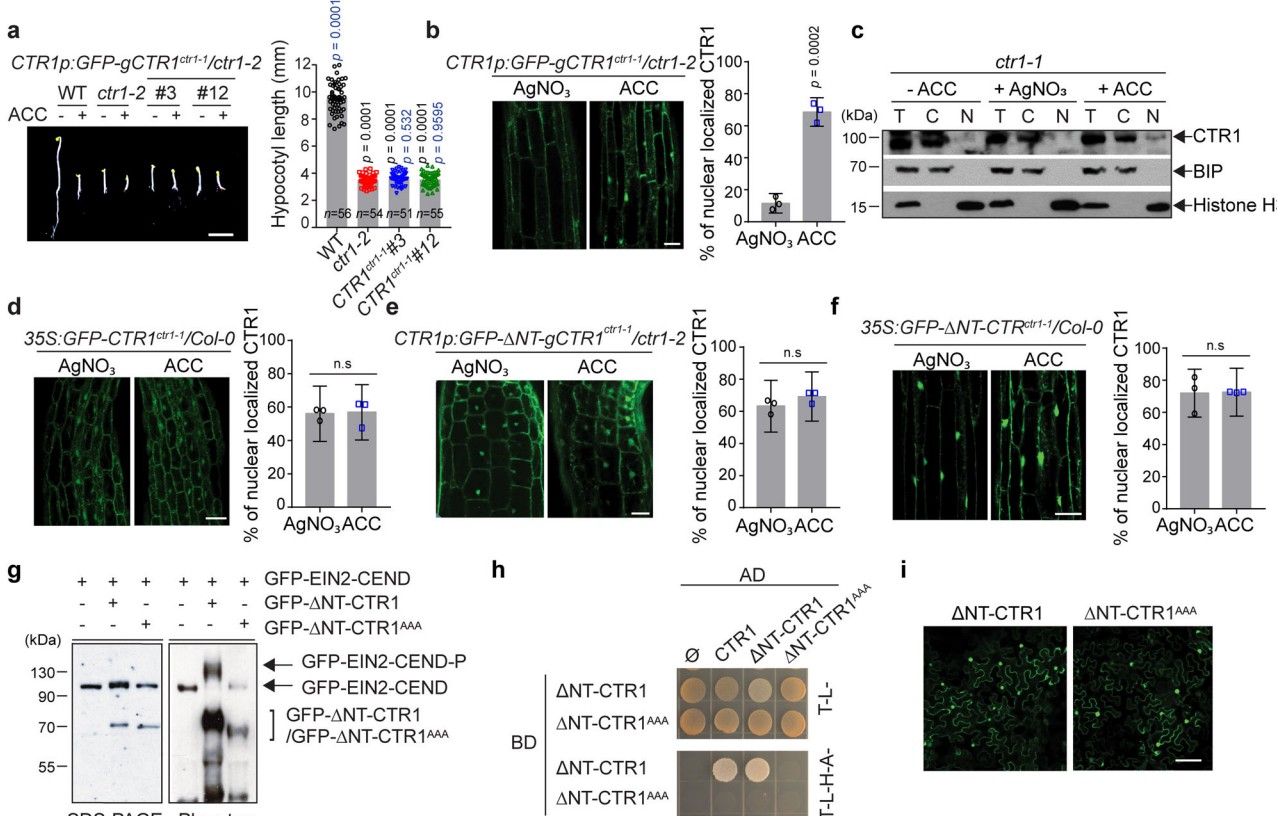

**Fig. 3 | The kinase activity of CTR1 is not necessary for ACC-induced CTR1 nuclear translocation. a** Seedlings were grown on MS medium with or without 10 μM ACC for 3 d and photographed. Scale bar, 5 mm. The graph represents the quantification of hypocotyl lengths of seedlings grown on MS without ACC. Significance was determined by one-way ANOVA with Dunnett's multiple comparisons test to compare the results to the WT (black) and *ctr1-2* (blue) controls. Data represent the means and SE. **b** Seedlings expressing GFP-CTR1-1 from its native promoter were grown on MS medium with or without AgNO3. The graph represents the ratio of nuclear-localized CTR1 in dark-grown seedlings treated with ACC or AgNO3. Error bars, SE (*n* = 3 seedlings). **c**, Fractionation analysis of *ctr1-1* mutant seedlings treated with or without ACC and silver nitrate. **d** Overexpression of CTR1-1 from 35S promoter leads to nuclear localization of CTR1 in the presence of silver nitrate and ACC. Error bars, SE (*n* = 3 seedlings). **e, f** Seedlings expressing GFP-ΔNT-

CTR1-1 from its native promoter (**e**) or the CaMV 35S promoter (**f**) were grown on MS medium with AgNO3 or treated with ACC. The graph represents the ratio of nuclear-localized CTR1 in dark-grown seedlings treated with ACC or AgNO3. Error bars, SE (*n* = 3 seedlings). **g** In vitro phos-tag analysis of active and inactive ΔNT-CTR1 in *Arabidopsis* protoplasts. **h** The indicated bait and prey constructs were co-transformed into the AH109 yeast strain, and the transformed yeast were grown on selection medium. **i** Tobacco leaves were infiltrated with Agrobacterium transformed with ΔNT-CTR1 or ΔNT-CTR1^AAA plasmid construct, followed by a 3 d incubation and visualization of nuclear signals by confocal microscopy. All scale bars represent 50 μm except the scale bar in (**a**). The areas below the hook and above the elongation zone of hypocotyls of dark-grown seedlings were used for imaging. Two-tailed Student's *t* test was performed to determine statistical significance for (**b, d, e, f**). n.s, statistically not significant.

SV40 proteins constitutively localized in the nucleus (Supplementary Fig. 11). *CTR1p:GFP-gCTR1^ctr1-8-SV40/ctr1-8* hypocotyls recovered approximately 45 min more slowly than *ctr1-8* after ethylene removal (Fig. 4c), thus indicating that nuclear-localized CTR1 promotes slower growth recovery. Together, these results suggest that nuclear-localized CTR1 delays the growth recovery of seedlings after ethylene removal, presumably via the stabilization of EIN3 proteins.

### Nuclear-localized CTR1 stabilizes EIN3 via EBFs without kinase activity

The correlation between nuclear-localized CTR1 and the delayed growth recovery kinetics suggests that CTR1 positively regulates nuclear ethylene responses. Therefore, we explored whether CTR1 might modulate EIN3 function in the nucleus in an ethylene-dependent manner. To test this possibility, we first determined whether CTR1 interacts with ethylene signaling components in the nucleus, including EIN2-CEND, EIN3, and EBFs, using yeast-two-hybrid and BiFC assays. We used CTR1 with an N-terminal deletion in the yeast-two hybrid assay because the full-length protein autoactivated in this assay (Supplementary Fig. 14a). Both ΔNT-CTR1 and ΔNT-CTR1-1 interacted with EIN2-CEND, EBF1, and EBF2, but not with EIN3 (Fig. 5a and

Supplementary Fig. 14b). Full-length CTR1 interacted with the EBFs in a BiFC assay regardless of ACC treatment (Fig. 5b and Supplementary Fig. 15), but not with EIN3 and EIN2-CEND. EIN2-CEND reconstituted YFP signals with its known nuclear interacting protein, EIN2 NUCLEAR-ASSOCIATED PROTEIN 1 (ENAP I) in the nucleus (Supplementary Fig. 16)[39], thus indicating that EIN2-CEND was expressed and present in the nucleus in these assays. Co-immunoprecipitation assays further confirmed that CTR1 interacts with EBF2 in vivo (Fig. 5c).

The ethylene response kinetics results showed that the enhanced nuclear localization of CTR1 delayed the restoration of seedling growth to basal levels after the removal of ethylene. Because EIN3 overexpression or a lack of EBF2 leads to similar recovery response kinetics, we examined endogenous EIN3 protein levels in the wild type and seedlings overexpressing ΔNT-CTR1-1 (*35S:GFP-ΔNT-CTR1^ctr1-1*) in response to different ACC concentrations. In agreement with findings from prior studies[19,40], ACC stabilized EIN3, displaying an ACC dose-dependent increase in the wild type, whereas, *35S:GFP-ΔNT-CTR1^ctr1-1* seedlings expressed significantly higher basal levels of endogenous EIN3, which were not further stabilized by ACC (Fig. 5d and Supplementary Fig. 17). Likewise, *ctr1-8* and *CTR1p:GFP-gCTR1^ctr1-8-SV40/ctr1-8* seedlings, both of which showed delayed growth recovery after

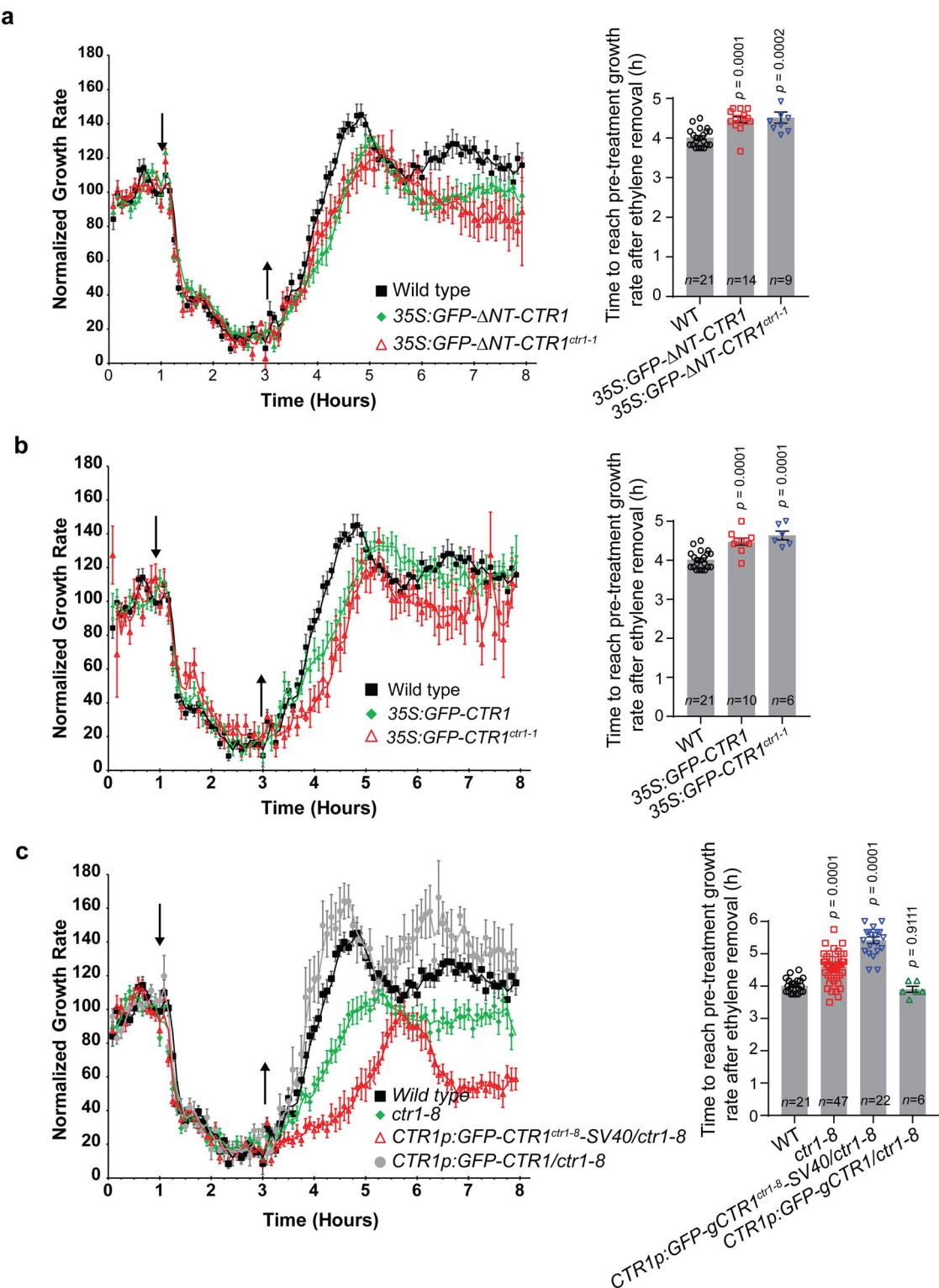

**Fig. 4 | Nuclear-localized CTR1 delays growth recovery of hypocotyls after removal of ethylene.** The hypocotyl growth rate in response to ethylene was recorded for 1 h in air, followed by 2 h exposure to 10 ppm ethylene and then 5 h recovery in air. Ethylene was introduced 1 h after measurements were initiated (down arrow) and then removed 2 h later (up arrow). The responses of wild-type seedlings are shown in each graph. **a** *35S:GFP-ΔNT-CTR1* and *35S:GFP-ΔNT-CTR1ctr1-1*. **b** *35S:GFP-CTR1* and *35S:GFP-CTR1ctr1-1*. **c** *ctr1-8*, *CTR1p:GFP-gCTR1ctr1-8-SV40/ctr1-8*, and *CTR1p:GFP-gCTR1/ctr1-8*. Data were normalized to the growth rate in air before treatment with ethylene. The experiments were repeated at least twice and generated similar results. The graphs indicate the quantification of time to reach pre-treatment growth rate after ethylene removal. Error bars, SE, Two-tailed Student's *t* test.

ethylene withdrawal (Fig. 4c), expressed higher basal levels of EIN3 than the wild-type seedlings, which did not show further EIN3 stabilization with higher ACC treatment (Fig. 5e, f). Corroborating the positive role of nuclear-localized CTR1 on EIN3 stabilization, a

higher steady-state level of EIN3 protein was detected in *35S:GFP-ΔNT-CTR1ctr1-1* seedlings than in wild-type seedlings after the removal of ACC (Fig. 5g). However, despite having higher EIN3 levels, the *35S:GFP-ΔNT-CTR1ctr1-1* did not display a strong triple response as compared to

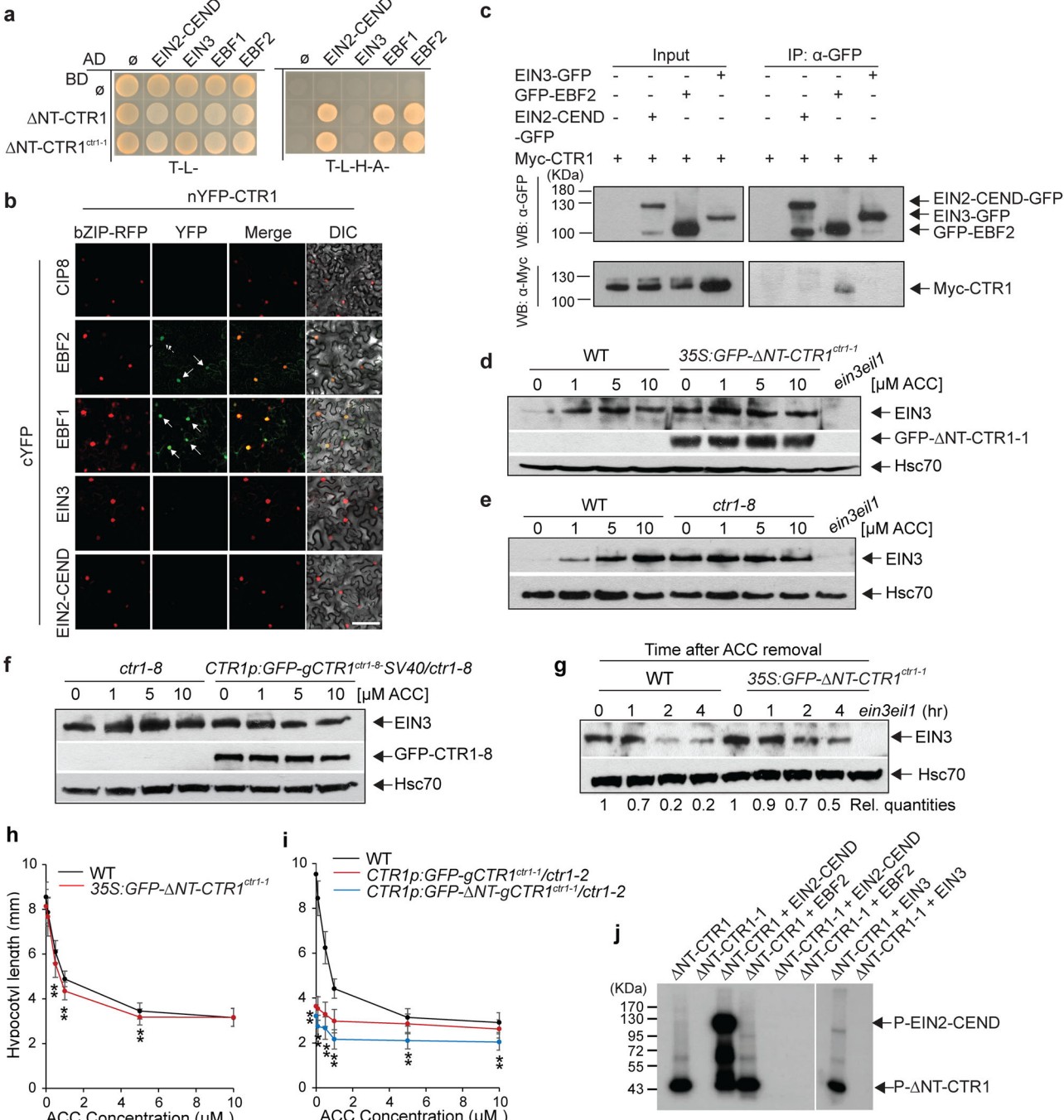

**Fig. 5 | Nuclear-localized CTR1 stabilizes EIN3 via a non-catalytic function.**
**a** CTR1 interacts with EBF1 and EBF2, but not EIN3 in yeast-two-hybrid assays.
**b** BiFC assay for full-length wild-type CTR1 and nuclear ethylene signaling proteins
in *N. benthamiana* in the presence of ACC. COP1-interacting protein 8 (CIP8) and
bZIP-RFP were used as a negative control and a nuclear subcellular marker,
respectively. **c** Co-immunoprecipitation analysis of CTR1 and EBF2 in *N. ben-
thamiana*. **d** Three-day-old dark-grown seedlings were treated with the indicated
concentrations of ACC for 2 h, and total protein extracts were used for immuno-
blotting with anti-EIN3, GFP, and Hsc70 antibodies. **e**, **f** *ctr1-8* (**e**) and seedlings
expressing GFP-CTR1-8-SV40 NLS in the *ctr1-8* mutant (**f**) expressed higher basal
levels of EIN3 than wild-type seedlings. The experiments were repeated at least
three times with similar results. **g** *35S:GFP-ΔNT-gCTR1ctr1-1* expressed a significantly
higher level of EIN3 protein than the wild type after removal of ACC. Seedlings
pretreated with 200 μM ACC for 2 h and total protein extracts were harvested at
the indicated times after ACC removal. Rel. quantities represent the ratio of the

intensity of the EIN3 band to Hsc70 band signals, and these values are expressed
relative to the intensity of EIN3/Hsc70 in the wild type with that in time 0 samples,
which was set to 1. **h** ACC dose-response curves for the hypocotyl length of 3-d-old
dark-grown wild-type and *35S:GFP-ΔNT-CTR1ctr1-1* seedlings. Control treatments
included no ACC. Error bars, SE ($n \geq 24$ seedlings for each ACC concentrations).
Two-tailed Student's $t$ test. **i** Seedlings expressing *GFP-ΔNT-gCTR1ctr1-1* from its
native promoter in *ctr1-2* mutant background showed an enhanced ACC response
compared to *CTR1p:GFP- gCTR1ctr1-1/ctr1-2* seedlings in the dark. Error bars, SE
($n \geq 46$ seedlings for each ACC concentrations). Two tailed Student's $t$ test. **j** CTR1
does not phosphorylate EBF2. In vitro kinase assay for purified ΔNT-CTR1 or ΔNT-
CTR1ctr1-1 with EIN2-CEND, EIN3, or EBF2. EIN2-CEND and EIN3 were used as a
positive and negative control, respectively. The indicated proteins were incubated
together in kinase reaction buffer and then separated by SDS/PAGE, and the
incorporated radiolabel was detected by autoradiography. Experiments were
repeated three times with the similar results.

previously described EIN3 overexpression lines (Fig. 5h). This line only produced significantly shorter hypocotyls than the wild type at lower concentrations of ACC[41]. Unlike *3SS:GFP-ΔNT-CTR1^{ctr1-1}*, *CTR1p:GFP-ΔNT-gCTR1^{ctr1-1}/ctr1-2* seedlings exhibited significantly shorter hypocotyls than the wild-type seedlings over a broader range of ACC concentrations (Fig. 5i). The hypersensitive ACC response of *CTR1p:GFP-ΔNT-gCTR1^{ctr1-1}/ctr1-2* as compared to *CTR1p:GFP-gCTR1^{ctr1-1}/ctr1-2* is consistent with the positive role of nuclear-localized GFP-ΔNT-CTR1-1 on ethylene responses. In vitro kinase assays did not show any phosphorylation of EBF2 by CTR1 (Fig. 5j), supporting the observation that CTR1 kinase activity is not involved in CTR1's nuclear movement and the regulation of EIN3 protein stability. Together, these results demonstrate that, after translocation to the nucleus, CTR1 directly interacts with the EBFs and consequently inhibits the EBF-mediated degradation of EIN3.

### The increased nuclear accumulation of CTR1 enhances stress tolerance to abiotic stress

Several studies have demonstrated that overexpression of EIN3 confers salinity tolerance to several plant species, including *Arabidopsis*[42–44]. Therefore, we asked whether *3SS:GFP-ΔNT-CTR1^{ctr1-1}* plants have enhanced tolerance to salt stress. *3SS:GFP-ΔNT-CTR1^{ctr1-1}* seedlings showed a higher survival rate (~47%) than wild type (~12%) on medium with 175 mM NaCl and had higher expression of salt-responsive genes upon salt stress (Fig. 6a, b). Consistent with the stress results from seedlings grown on agar media, *3SS:GFP-ΔNT-CTR1^{ctr1-1}* plants showed strong salt-tolerance phenotypes on soil irrigated with 300 mM NaCl for 4 weeks (Fig. 6c and Supplementary Fig. 18a). Intriguingly, *3SS:GFP-ΔNT-CTR1^{ctr1-1}* plants displayed a wild-type-like seedling phenotype, which is inconsistent with its higher EIN3 levels (Fig. 5d). To determine if the enhanced salt tolerance of *3SS:GFP-ΔNT-CTR1^{ctr1-1}* is due to the truncated CTR1, we examined the effect of salinity stress on seedlings expressing a full-length, inactive CTR1 that translocates into the nucleus (*3SS:GFP-CTR1^{ctr1-1}*). *3SS:GFP-CTR1^{ctr1-1}* expressed higher levels of EIN3 than the wild type and showed enhanced tolerance to salt stress, but it produced significantly shorter hypocotyls and smaller seedlings than that of the wild type (Supplementary Fig. 18a, b, c, and d). Similar to *3SS:GFP-CTR1^{ctr1-1}*, seedlings expressing WT CTR1 (*3SS:GFP-CTR1*) also displayed enhanced salt tolerance phenotypes, suggesting that kinase activity of CTR1 does not play a major role in salt stress resilience (Supplementary Fig. 18e). Furthermore, seedlings overexpressing CTR1-8 (*3SS:GFP-CTR1^{ctr1-8}*) showed comparable salt stress responses to the wild-type seedlings (Fig. 6d, e), which is consistent with its inability to translocate to the nucleus and comparable ethylene response and recovery kinetics to the wild-type (Supplementary Fig. 19). Additionally, seedlings expressing CTR1-8 or CTR1-8-SV40 from their native promoter in the *ctr1-8* background demonstrated a contrasting response to salinity stress; the introduction of a *GFP-gCTR1^{ctr1-8}* transgene compromised the salt-resistant phenotype of the *ctr1-8*, whereas SV40 NLS-mediated nuclear localization of GFP-CTR1-8 conferred comparable levels of strong salt resistance phenotypes to the seedlings (Fig. 6f). We postulate that the contrasting phenotypes may be a result of the CTR1-8, which is unable to translocate into the nucleus, suppressing EIN2 in the cytoplasm, thus rescuing the weak hypermorphic nature of the *ctr1-8* mutant. The analysis of the salt tolerance of various ethylene signaling mutants and transgenic lines reveals a correlation between a strong ethylene response and enhanced salt tolerance (Supplementary Fig. 20).

The role of ethylene in drought and water stress is relatively elusive compared to its role in salinity stress. However, a few studies on *Arabidopsis*, *Morus*, and rice have suggested that ethylene plays a role in drought responses as well[42,45,46]. Similar to the response to salt stress, *3SS:GFP-ΔNT-CTR1^{ctr1-1}* and *3SS:GFP-CTR1^{ctr1-1}* showed significantly enhanced stress tolerance to drought (Fig. 6g–j). By contrast, *3SS:GFP-CTR1^{ctr1-8}* displayed a wild-type-like response to drought stress (Fig. 6k).

Similar to the salinity stress responses, *CTR1p:GFP-CTR1^{ctr1-8}/ctr1-8* and *CTR1p:GFP-CTR1^{ctr1-8}-SV40/ctr1-8* exhibited contrasting responses to drought stress, showing that the nuclear localization of CTR1-8 confers enhanced tolerance to drought (Fig. 6l). The germination analysis of seedlings in the presence of increasing concentrations of mannitol showed that *ctr1-1*, *ctr1-2*, as well as an EIN3 overexpression line and *ebf2-3* exhibited a significantly higher germination rate than the wild-type, indicating a link between ethylene signaling and drought resistance (Supplementary Fig. 21). Furthermore, the germination rate of *3SS:GFP-ΔNT-CTR1^{ctr1-1}* and *3SS:GFP-CTR1^{ctr1-1}* was comparable or slightly higher than the *ctr1-1*, *ctr1-2*, *ebf2-3*, and *EIN3ox* in the presence of high levels of mannitol, consistent with their enhanced tolerance to drought in soil-grown conditions. Taken together, these results demonstrate that an increasing level of nuclear-localized CTR1 confers enhanced tolerance to abiotic stress, likely through reinforcing ethylene signaling by stabilizing EIN3 proteins.

## Discussion

As a direct modulator of EIN2 function, the role of CTR1 has been firmly established as a negative regulator in the ethylene signaling pathway. In this study, we demonstrated that CTR1 also acts as a positive regulator for ethylene response when it translocates into the nucleus upon the perception of ethylene by the receptors in the ER. CTR1 nuclear movement does not require EIN2, yet it stabilizes EIN3 upon entering the nucleus, suggesting that ethylene signaling can be activated in part, in an EIN2-independent manner. Kinetic analysis showed that CTR1 translocates into the nucleus later than EIN2. However, it is also possible that a low level of CTR1 nuclear translocation occurs earlier or concurrently with EIN2 nuclear translocation after its release from the ER. Ethylene significantly increases CTR1 protein abundance (Fig. Supplementary Fig. 4)[28], and thus the CTR1 nuclear movement that we observed may primarily result from the presence of excess CTR1 that is not bound to ethylene receptors. In any case, given the complete ethylene insensitivity of the *ein2-5* null mutant[12], the nuclear-localized CTR1 likely plays a secondary role in ethylene responsiveness, likely fine-tuning the EIN2-mediated primary responses in the presence of abiotic stress via the additional stabilization of EIN3 in the nucleus (Fig. 7).

The failure of CTR1-8 protein to translocate to the nucleus was surprising given the weak interaction of CTR1-8 with the ethylene receptors, which would increase the release of CTR1-8 from the ER. Expression of full-length, wild-type CTR1 protein in loss-of-function ethylene receptor mutants constitutively localized in the nucleus, as do N-terminal deletion versions of CTR1 in the wild-type, suggesting that the interaction with ethylene receptors prevents CTR1 release from the ER. We speculated that the modification of CTR1 after ethylene binding to the receptors may relieve N-terminus-mediated inhibition of CTR1, thereby enabling CTR1 release from the ER and subsequent nuclear translocation. Given the weak interaction with the ethylene receptors, the CTR1-8 mutant protein may not undergo this conformational change and consequently fails to translocate to the nucleus. Alternatively, *ctr1-8* mutation might prevent the mutant CTR1-8 protein from interacting with an unknown cargo protein that transports CTR1 to the nucleus, given that CTR1 lacks a canonical NLS.

CTR1 forms a complex with EBFs and consequently influences EIN3 protein stability, although the detailed underlying mechanism remains elusive. Intriguingly, this process is independent of the catalytic function of CTR1. Many non-catalytic functions of protein kinases have been reported in yeast and mammalian systems[47–50], demonstrating that their extensive roles in biological processes as scaffolds, allosteric regulators, and molecular switches beyond phosphorylating proteins[47,48,51–53]. Given that CTR1 does not interact with EIN3, CTR1 might allosterically inactivate the activity of EBFs, thus inhibiting EIN3 degradation. Alternatively, CTR1 might decrease free nuclear EBF

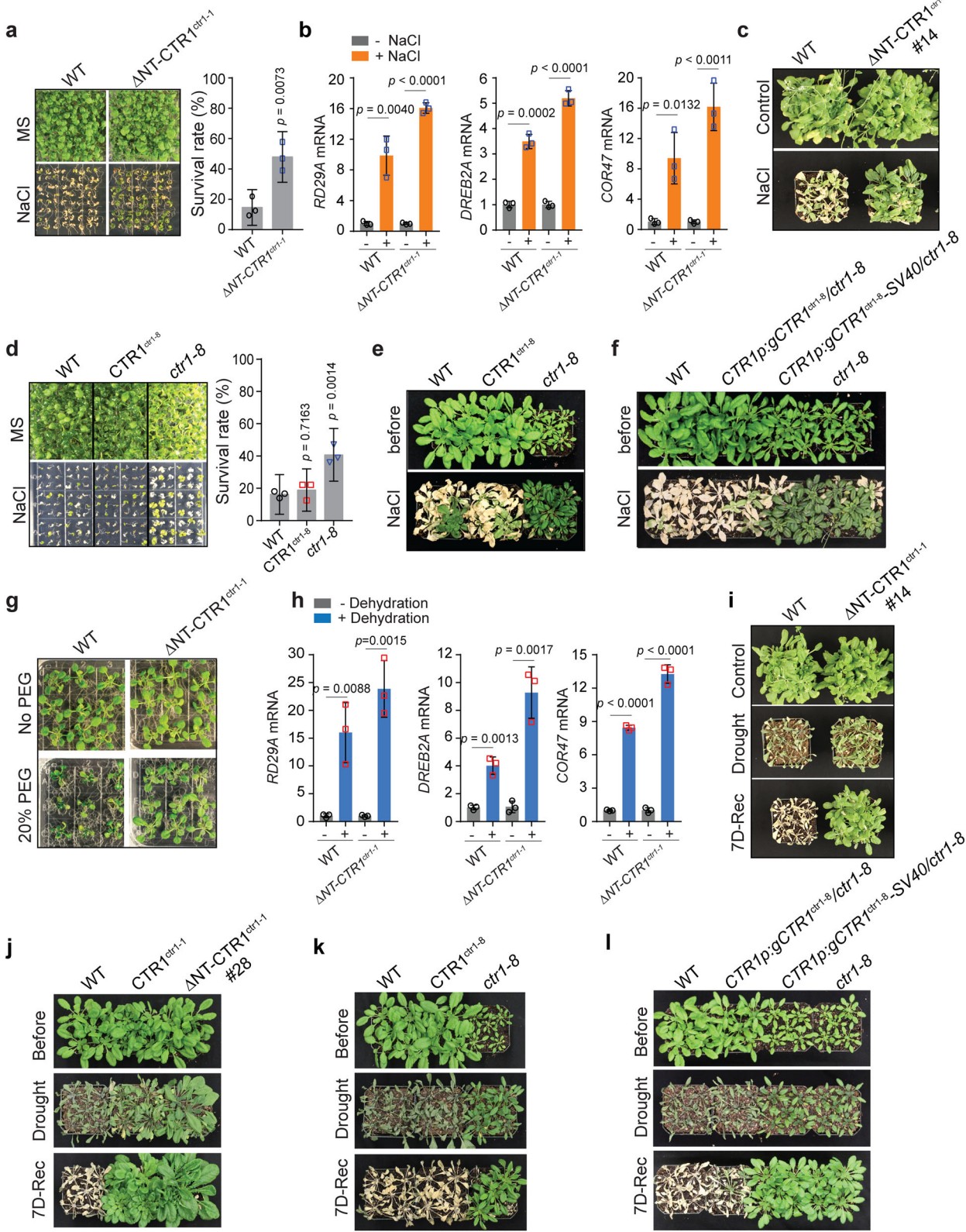

pools that interact with EIN3 through the formation of CTR1-EBF complexes. The kinase activity-independent role of CTR1 is consistent with the enhanced stress response of plants expressing either an active or inactive form of CTR1. Given that activation of ethylene signaling likely leads to inactivation of CTR1 kinase activity, the kinase-active form of CTR1 expressed in plants may be inactivated upon stress-induced activation of the ethylene signaling pathway. Further studies

examining the kinase activity of CTR1 upon ethylene or stress treatment could provide additional insight into this process.

Analysis of ethylene growth response kinetics has been instrumental in examining various ethylene mutants; however, the detailed mechanism underlying the recovery kinetics remains unknown. Previous studies have shown that increased abundance of EIN3 is a major factor underlying the delayed recovery kinetics after ethylene

**Fig. 6 | Increased CTR1 nuclear accumulation enhances plant's resilience to abiotic stress. a** Seedlings were grown on MS medium containing with or without 175 mM NaCl for 3-weeks and the survival rate of the seedlings were counted. A graph represents the quantification of survival rate of the seedlings. Error bars, SE (*n* = 3 biological replicates), Each biological replicate contains 64 seedlings per genotype. **b** Relative expression of salt-induced genes. One-week-old seedlings were treated with 175 mM NaCl solution for 3 h and RNA was extracted. Error bars, SD (*n* = 3 biological replicates). **c** Two-week-old seedlings were irrigated with 300 mM NaCl solution every 4 days for 28-d, then watered normally for 7-d. **d** The survival rate of the *ctr1-8* and *3SS:GFP-CTR1^{ctr1-8}* seedlings on MS medium containing with or without 175 mM NaCl. Error bars, SD (*n* = 3 biological replicates), Each biological replicate contains 72 seedlings per genotype. **e**, **f** Four-week-old

seedlings were irrigated with 300 mM NaCl solution every 4 days for 12 d, then watered normally for 7-d. **g** One-week-old light-grown seedlings were transferred MS medium containing 20% PEG8000 for a week and photographed. **h** Relative expression of drought-induced genes. One-week-old seedlings were placed on filter papers and dried for 3 h at room temperature and used for the analysis. Error bars, SD (*n* = 3 biological replicates). **i**. 2-week old WT and *3SS:GFP-ΔNT-CTR1^{ctr1-1}*(#14) seedlings were subjected for water stress by withdrawing water for 28-d, followed by 7-d rewatering. **j–l** Four-week-old wild-type, *3SS:GFP-CTR1^{ctr1-1}*, *3SS:GFP-ΔNT-CTR1^{ctr1-1}*(#28), *3SS:GFP-CTR1^{ctr1-8}*, *ctr1-8*, *CTR1p:GFP-gCTR1^{ctr1-8}/ctr1-8* and *CTR1p:GFP-gCTR1^{ctr1-8}-SV40/ctr1-8* mutant seedlings were subjected to water stress, followed by 7-d of re-watering recovery (7D-Rec). Statistical significance we determined by Student's *t* test relative to control for (**a**, **b**, **d**, **h**).

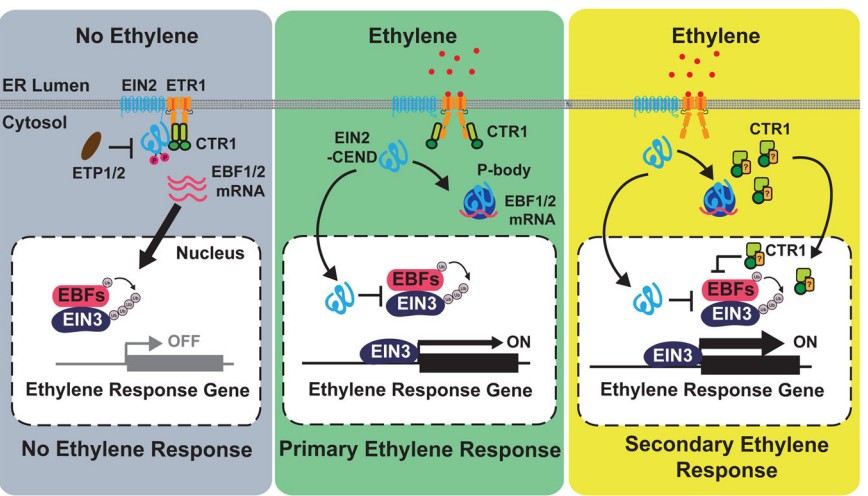

**Fig. 7 | Model for ethylene-induced CTR1 nuclear translocation and stimulation of the ethylene response.** In the absence of ethylene, the ethylene receptors, CTR1, and EIN2 mainly localize to the ER. Receptor-activated CTR1 phosphorylates EIN2, thus leading to the proteolytic degradation of EIN2 via EIN2-TARGETING PROTEIN 1 (ETP1) and ETP2. In the presence of ethylene, the receptors, and hence CTR1, are inactivated. Inactivated CTR1 no longer phosphorylates EIN2, and the C-terminal domain of EIN2 (EIN2-CEND) is proteolytically cleaved and translocates to the nucleus, where it initiates the primary ethylene response via EIN3 activation.

EIN2-CEND is also targeted to processing bodies (P-bodies) and suppresses EBF mRNA translation. After the nuclear translocation of EIN2-CEND, CTR1 is released from the receptors and relocates to the nucleus via an unknown mechanism. The nuclear-localized CTR1 directly interacts with EBF2, thus stabilizing EIN3 via an unknown mechanism and ultimately stimulating the ethylene response. The yellow square with a question mark indicates an unknown cargo protein that delivers CTR1 to the nucleus. Arrows indicate the movement of ethylene signaling components or positive influence; blunted ends indicate inhibition.

removal[21]. Our studies have revealed that nuclear-localized CTR1 plays a role in the growth recovery kinetics of *Arabidopsis* hypocotyls by increasing EIN3 stability. Given its negative roles in the ethylene response, the CTR1-mediated stimulation of the ethylene response in the nucleus was surprising; however, many proteins have dual functions in signal transduction pathways, and some perform opposite activities in the same biological process[54–57]. For example, in human memory control, AGAP3, an NMDA receptor-interacting signaling protein, not only enhances the activity of memory formation but also decreases the activity of memory formation, depending on the perception of different signals[54]. The positive role of nuclear-localized CTR1 in ethylene responses was also apparent in the tolerance of *Arabidopsis* seedlings to salinity and drought stress. The stress response analysis of seedlings expressing CTR1-8 or CTR1-8-SV40 from its native promoter showed that the enhanced stress tolerance of *3SS:GFP-ΔNT-CTR1^{ctr1-1}* and *3SS:GFP-CTR1^{ctr1-1}* is not an artifact derived from overexpression of CTR1 proteins. While it is uncertain to what extent nuclear-localized CTR1 affects the stress response of plants, the level of EIN3 protein appears to be correlated with stress tolerance. This suggests that nuclear-localized CTR1 is important for responding to abiotic stresses, particularly those that require fast or prolonged activation of the ethylene signaling pathway. One of the intriguing observations of the 35S CaMV promoter lines is the dissonance between the wild-type-like phenotypes of *3SS:GFP-ΔNT-CTR1^{ctr1-1}* and its

high level of EIN3. One possible explanation for this discrepancy is that the overexpressed, truncated CTR1 protein may interact or lack interaction with an unknown pathway that influences plant development, such plant size. However, given the complexity of signaling pathways and the unpredictable outcomes of perturbing these pathways, this is just one of the many possible explanations for the phenomenon. Further studies are needed to fully understand the mechanisms involved. Together, the CTR1-mediated stimulation of ethylene responses in the nucleus might be an elegant way in which plants maximize ethylene responses by turning a negative regulator into a positive regulator, thereby enabling rapid acclimation of plants to stresses.

CTR1 lacks a canonical NLS. The failure of the CTR1-8 protein to undergo nuclear translocation supports the absence of an NLS in CTR1 as the CTR1-8 only differs from the wild type in its binding affinity to ETR1. However, it is still possible that CTR1 may be able to enter the nucleus through a non-canonical NLS. Furthermore, elucidation of the mechanism underlying CTR1-mediated EBF regulation and the identification of a nuclear pathway that promotes rapid growth recovery would provide more insights advancing the mechanistic understanding of ethylene response regulation. Ethylene interacts with a myriad of internal and external stimuli in regulating plant growth and stress responses. Thus, revisiting crosstalk with other pathways by considering the nuclear role of CTR1 should be of great interest.

## Methods

### Plant materials and growth conditions

*Arabidopsis thaliana* Col-0 was used as the WT reference throughout the study. All plants were grown in either long-day or short-day conditions at 22 °C ± 2 °C or in vitro on Murashige and Skoog (MS) basal medium supplemented with 0.8% plant agar (pH 5.7) in a continuous light chamber at 21 °C. All plants used were homozygous or T2 lines. Homozygous transgenic lines were identified by the segregation of antibiotic resistance followed by the confirmation of protein expression via immunoblot analyses. Mutants used in this study: *ein2-5*[12], *ein3eil1*[40], *etr2-3ers2-3ein4-4*[58], *etr1-6etr2-3ein4-4*[59], *ctr1-2*[11], *ctr1-1*[11], *ctr1-8*[36], *etr1-1*[60], *ebf2-3*[21] were previously described.

### CTR1 constructs and site-directed in vitro mutagenesis for *Arabidopsis* transformation

All molecular cloning was performed with the Gateway (Invitrogen) or infusion cloning (Takara Bio USA) strategies unless otherwise specified. Table S3 list all primers used for cloning and mutagenesis. To create the *CTR1p-GFP-gCTR1* construct, we PCR-amplified three separate overlapping fragments (CTR1 promoter, GFP, and the genomic fragment of CTR1). The 0.96 Kb CTR1 promoter and 4.7 kb full length CTR1 genomic fragment were amplified with Col-0 genomic DNA as a template, and the GFP coding sequences were amplified with the binary vector pEarleyGate 104. Three overlapping fragments were subsequently subjected to an infusion reaction with *StuI*- and *XbaI*-digested pEarleyGate 104, thus yielding the GFP-full-length CTR1 clone in the pEarleyGate 104 backbone. Genomic *CTR1* fragment mutations (G354E *ctr1-8* or D694E *ctr1-1*) were introduced into the WT genomic fragment via PCR-based mutagenesis, and the resulting fragments were used for infusion reactions, as described above. *CTR1p-GFP-gCTR1* and its mutant variant constructs were also constructed as described above. To construct *35S:GFP-CTR1*, *GFP-CTR1ctr1-1*, *GFP-ΔNT-CTR1*, *GFP-ΔNT-CTR1ctr1-1*, and *GFP-CTR1ctr1-8*, we cloned the coding sequences of full length CTR1 or its kinase domain into the pENTR entry vector and subsequently transferred them to the binary vector pSITE2CA. Mutations were introduced in the coding sequences of CTR1 in the pENTR vector, and the sequences were further transferred to pSITE2CA. To create the *CTR1p-GFP-gCTR1ctr1-8-SV40 NLS* construct, we added *SV40 NLS* sequences to the *CTR1p-GFP-gCTR1ctr1-8* fragment via overlapping PCR and cloned the resulting fragment into the pEarleyGate 104 vector via infusion as described above. All 35S promoter-driven lines were transformed into WT background.

### Bimolecular fluorescence complementation constructs

The following coding sequences including a stop codon were transferred from the pENTR vector into pCL112 or pBatTL to generate N-terminal nYFP-fusions and C-terminal cYFP fusions: *CTR1, ENAP* (nYFP fusion), and *CIP8* (cYFP fusion). The following coding sequences without a stop codon were transferred from the pENTR vector into pBAT-YFPc to generate C-terminal cYFP-fusions: *EIN3, EIN2-CEND*, and *CIP8*. The coding sequences of *EBF1* and *EBF2* in pENTR were transferred into pCL113, thus creating *cYFP-EBF1* and *cYFP-EBF2*.

### Phos-Tag gel analysis

Preparation of Phos-Tag polyacrylamide gels and subsequent immunoblotting were performed according to the manufacturer's instructions. The Phos-Tag gel was prepared with 5 ml of 8% acrylamide and 20 μM Phos-Tag (Wako) for the resolving gel and 4 mL of 4.5% acrylamide for the stacking gel. After completion of electrophoresis, the gel was incubated in 100 mL transfer buffer containing 0.2% (w/v) SDS and 10 mM EDTA for 30 min. Protein was blotted onto nitrocellulose, which was then blocked with 5% nonfat milk. The membranes were then probed with a 1:5000 dilution of Roche anti-GFP (Sigma Aldrich Cat#

11814460001) or a 1:5000 dilution of anti-mouse HRP secondary antibody (ThermoFisher Cat# 31430).

### Time-lapse growth recovery analysis

To measure the ethylene response growth kinetics of seedlings, we grew seedlings on vertically oriented Petri plates in the dark to a height of 3 to 4 mm (42–46 h) before the beginning of the growth-rate measurements. The agar plates were placed vertically in a holder and fitted with a lid for continuous gas flow (100 mL min⁻¹). Seedlings were grown in air for 1 h, and ethylene (typically 1 or 10 ppm) was applied for 2 h and then removed. Images were acquired every 5 min with a CCD camera fitted with a close-focus lens with illumination provided by infrared LEDs. The growth rates of the hypocotyls in every time interval were then calculated. Under these conditions, the equilibration time of the chamber at these flow rates was approximately 30 s, which was much faster than the image acquisition time. From the results, we determined how various mutations affected growth inhibition kinetics when ethylene was added and affected the recovery kinetics when it was removed.

### Yeast-two-hybrid assays

The coding sequences of full-length *CTR1* or its kinase domain (*ΔNT-CTR1*) with or without *ctr1-1* mutation (D694E) in the pENTR GW entry vector were transferred into the pGADT7. The resulting bait clones were paired with *EIN2-CEND*, *EIN3*, *EBF1*, or *EBF2* in the pGBKT7 or pGADT7 vector, and their interactions were tested in yeast. Positive interactions between the prey and bait were selected on medium lacking histidine, tryptophan, leucine, and adenine. Because of the autoactivation of full-length *CTR1* in pGBKT7, we used *ΔNT-CTR1* or *ΔNT-CTR1ctr1-1* to generate *CTR1* bait constructs. The transformants were extracted by Zymo protein kit (Cat# Y1002) and resolved through SDS-PAGE followed by immunoblotting with anti-Myc (Sigma, Cat# 11814150001, 1:5000) or anti-HA (Roche, Cat#11867423001, 1:10000).

### Confocal microscopy

All imaging of GFP, YFP, and RFP was performed with a laser-scanning confocal microscope (Zeiss LSM880 upright) and the areas below the hook and above the elongation zone of the hypocotyls of dark-grown seedlings were imaged. Samples were directly mounted on a glass slide in water containing with or without ACC. For imaging of *Arabidopsis* seedlings transformed with CTR1 constructs, the seedlings were grown on MS medium with or without 100 μM AgNO₃ supplementation in the dark for 3 d. For ACC treatment, seedlings on MS without AgNO₃ were treated with 200 μM ACC dissolved in water for 2 h. For ethylene treatment, seedlings were grown directly on MS medium in GC vials for 3 d. The GC vials were subsequently capped, injected with 10 ppm ethylene gas, and incubated for 2 h. For nuclear imaging, 3-d-old dark-grown seedlings were treated with 200 μM ACC for 2 h, incubated with Hoechst33342 solution (Invitrogen) for 30 min, and briefly washed before mounting on slides. For imaging light-grown seedlings, plants were grown on MS in constant light for 5 d. For imaging of fluorescence signals in protoplasts, transfected protoplasts were incubated with 200 μM ACC for 2 h in the dark and then examined. To image BiFC, leaf disks of infiltrated tobacco leaves with CTR1 and counterpart constructs (*EIN2-CEND, EIN3, EBF1, EBF2, ENAP1*, and *CIP8*) were mounted on glass slides in water.

### Coimmunoprecipitation assays

*N. benthamiana* leaves were infiltrated with Agrobacteria co-transformed with the plasmids of interest and incubated for 3 d in a growth chamber. Total proteins were extracted from the infiltrated leaves and homogenized in co-immunoprecipitation buffer (25 mM Tris, pH 7.5, 150 mM NaCl, 5 mM EDTA, pH 8.0, 1 mM DTT, 1 mM PMSF, and 1× protease inhibitor cocktail (Roche, Cat#

04693132001). After quantification of the protein concentration with a Bradford assay, an equal amount of total protein extracts was incubated with anti-GFP (Sigma Aldrich Sigma, Cat# 11814460001) overnight at 4 °C, then incubated with Protein A/G magnetic beads (Fisher) for 1 h at room temperature with gentle shaking. The total protein suspension containing the magnetic beads was applied to a magnetic column, washed three times with co-IP buffer, eluted with boiled 2× SDS sample buffer, and subjected to immunoblotting analysis with anti-GFP (Sigma, Cat# 11814460001: 1:5000) and anti-Myc (Sigma, Cat# 11814150001).

### Nuclear-cytoplasmic fractionation

Three-d-old etiolated *Arabidopsis* seedling was grinded with ice-cold lysis buffer (20 mM Tris, pH7.4, 25 % glycerol, 20 mM KCl, 2 mM EDTA, pH 8.0, 2.5 mM MgCl$_2$, 250 mM sucrose, 40 μM MG132, 5 mM DTT, 1 mM PMSF, 1× Protease inhibitor cocktails) followed by filtration with 2 layers of Miracloth. The flow-through sample was collected and 200 μl was saved as total proteins. The flow-through was spun at 1,500 g for 10 min to pellet the nuclei and the supernatant was transferred to new tubes and saved as the cytoplasmic fractions. The resulting pellet was resuspended in washing buffer (20 mM Tris, pH 7.4, 25% glycerol, 2.5 mM MgCl$_2$, 25% Triton X-100, 20 μM MG132, 5 mM DTT, 1 mM PMSF, 1x Protease inhibitor cocktails) and spun at 1,500 g for 10 min. After repeating washing the pellet with the washing buffer five times, the pellet was resuspended in the RIPA buffer (50 mM Tris, pH7.4, 50 mM NaCl, 2 mM EDTA, 0.1% SDS, 40 μM MG132, 5 mM DTT, 1 mM PMSF, and 1× Protease inhibitor (Roche). The samples were resolved by SDS-PAGE and transferred to nitrocellulose membrane for immunoblotting analysis with anti-GFP (Sigma, Cat# 11814460001: 1:5000 dilution), -BIP (Santa Cruz, Cat# sc-8017: 1:3000 dilution), -Histon H3 (Novusbio, Cat# NB500-171: 1:5000 dilution), and anti-CTR1 antibodies (in-house: 1:3000 dilution).

### In vitro kinase assays

His$_6$-ΔNT-CTR1 and other proteins were expressed in BL21 (Rosetta) and the soluble fraction of total proteins were purified by using Ni-NTA His-Bind Resin (Millipore Sigma, Cat# 70666). A total of 20 ng purified His$_6$-ΔNT-CTR1 or His$_6$-ΔNT-CTR1$^{ctrl-1}$ protein was incubated with 100 ng of His$_6$-EIN2-CEND-His$_6$, His$_6$-EBF2, or His$_6$-EIN3 in kinase reaction buffer [50 mM Tris pH 7.5, 10 mM MgCl$_2$, 1× Roche Complete Protease Inhibitor mixture, and 1 μCi [γ−$^{32}$P] ATP] for 30 min at room temperature. After incubation, the reactions were terminated by boiling in 6× Laemmli SDS sample buffer for 3 min. Samples were subjected to SDS/PAGE, dried, and visualized by autoradiography.

### Real-time quantitative PCR analysis

Total RNA was prepared with RNeasy Plant Mini Kits (QIAGEN, Cat# 74904) and reverse transcribed with SuperScript II reverse transcriptase (Invitrogen) according to the manufacturers' instructions. Quantitative RT-PCR was performed with PowerUP$^{TM}$SYBRGreen Master Mix (Applied Biosystems, Cat# A25741). The primers used are listed in Table S3. Three biological replicates were analyzed with three technical replicates per sample. The relative expression of candidate genes was normalized to *Actin 2*.

### Immunoblot analysis

Three-day-old dark-grown seedlings were treated with different concentrations of ACC for 2 h. The harvested seedlings were weighed, and 2× SDS sample buffer was added to the seedlings in proportion to their weight. The samples were then homogenized with a pestle. Subsequently, the same amount of total protein extract from each sample was boiled for 3 min and resolved through SDS-PAGE followed by immunoblotting with anti-EIN3 (Agrisera, Cat# As194273, 1:3000 dilution) or anti-Hsc70 (Enzo Life Science, Cat# ADI-SPA-818-F,

1:20000 dilution). Signals were detected with SuperSignal West Pico or Femto Maximum Sensitivity Substrate (Thermo Fisher Scientific, Cat# 34580, 34095), and band intensities were measured in Image J software.

### *Arabidopsis* drought and salt stress experiments

For drought recovery experiments, *Arabidopsis* seedlings were grown on soil in short-day conditions (12 h L/12 hD) for 4 weeks, and water was withdrawn for 10-d followed by re-watering for 7-d. For germination assays of seedlings on growth media containing different concentrations of mannitol, surface-sterilized seeds were sown on media with or without mannitol and transferred to 4 °C for 5-d of cold treatment, followed by incubation in a growth chamber (24 h light). The germination of seeds was monitored and photographed for 5-d. For PEG experiments, one-week-old light-grown seedlings were transferred MS medium containing 20% PEG8000 for a week and photographed. For the salt stress treatment, *Arabidopsis* seedlings were grown on soil in short-day conditions for 4 weeks. Four-week-old seedlings were treated with 300 mM NaCl every 4-d for 12 d, followed by 7-d of recovery. For survival assays of seedlings on salt plates, seedlings were sown on plates containing 175 mM NaCl and incubated in the light for 10-d The experiments were repeated three times with similar results.

### Reporting summary

Further information on research design is available in the Nature Portfolio Reporting Summary linked to this article.

## Data availability

All data are available in the main text or the supplementary materials. Source data are provided with this paper.

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

## Acknowledgements

The authors thank Alison Delong for critical reading of the manuscript, Eric Schaller for providing ethylene receptor mutants, Jose Alonso for providing EIN3 overexpression lines, McNickle Gordon for statistics advice, and Purdue Imaging Facility for confocal microscope service. This work was supported by grants from NSF (MCB-1817286) and Ralph W. and Grace M Showalter Research Trust to GMY, and NSF to JJK (IOS-1856431) and BMB (MCB-1817304).

## Author contributions

G.M.Y. and J.J.K. conceived the experiments; G.M.Y. planned and supervised experiments; H.L.P., D.H.S., H.Y.L., C.P., Y.C. conducted experiments except ethylene-response kinetics; A.B. and B.M.B. conducted ethylene response kinetics analysis; G.M.Y., J.J.K., B.M.B., H.L.P., D.H.S., H.Y.L., and C.P. contributed to the interpretation of the data; G.M.Y. wrote the manuscript with support from J.J.K. and B.M.B. H.L.P., D.H.S., and H.Y.L. equally contributed to the manuscript.

## Competing interests

The authors declare no competing interests.
