## [Peer Review File · Nature Communications]

Ethylene-triggered subcellular trafficking of CTR1 enhances the response to ethylene gasReviewer #1 (Remarks to the Author):

The work of Park et al provides compelling evidence for a novel function of CTR1 in ethylene signaling. They nicely demonstrated that CTR1 is able to migrate to the nucleus after an ethylene signal has been perceived. This information is totally novel and quite exceptional, making it a big breakthrough in our understanding of the ethylene signaling pathway. The release of CTR1 from the ethylene receptor at the ER seems a crucial process that enables the CTR1 nuclear translocation, as demonstrated by the use of different reporter lines. The authors also show convincingly that CTR1 is interacting with the EBFs in the nucleus, and that this interaction is probably stabilizing EIN3 proteins. As such, the authors nicely reveal a first important glimpse on the possible mode of action of nuclear CTR1. I feel these findings provide a lot of novelty to the community, as they reveal a totally novel mechanism how ethylene signaling can occur independent of EIN2 and how ethylene nuclear events are fine-tuned.

I have a few unresolved questions that might be addressed or discussed in the manuscript or elucidated with a few extra simple experiments. For example, why is the CTR1 fluorescence increasing after an ethylene treatment at the ER and the nucleus, while the CTR1 fluorescence level is not decreasing at the EC (only the nuclear signal) when the ethylene signal is removed. Another small thin I noticed is that the CTR1-EBF interaction also occurs outside of the nucleus (Figure 5), which is not discussed in the manuscript. What would this mean or is this an artifact of the 35S construct used?

The latter brings me to two key points that require some more substantial attention. The authors often use 35S constructs or lines in which CTR1 is constitutively present in the nucleus, even in the absence of ethylene. I understand the logic behind this, but I wonder if it could also not lead to false interpretations. Under normal conditions, CTR1 levels are supposed to reduce in the nucleus if the ethylene signal drops. This reasoning actually is used by the authors to link the potential nuclear function of CTR1 with a delayed response upon ethylene release. However, the growth kinetics experiments (Figure 4) lack data with lines that do not have the 35S or constitutive nuclear localized CTR1. I wonder how these lines would perform, as the main reasoning is that ethylene will lead to nuclear localization of CTR1 and ethylene release would remove that nuclear localization. I wonder if the GFP:CTR1 line in the *ctr1-2* mutant has a normal wild type growth response, or has the GFP:CTR1(*ctr1-8*) line (no CTR1 nuclear transfer) an abolished growth response? If the release of ethylene is the actual physiological event where nuclear CTR1 is operational via the action of EIN3 stability, then the authors should show that EIN3 stabilization occurs during the release of ethylene, and not during the onset of ethylene signaling. I believe they have all the CTR1-lines available to show this EIN3 response.

A final remark that also concerns me, is the use of the constitutively nuclear localized CTR1 is its role during stress resilience, especially drought stress. First of all, ethylene does not play a dominant role in drought stress responses (opposed to its role in salinity stress), and yet the authors make it seem that the nuclear CTR1 is important to respond to drought stress. If ethylene production (or signaling) is not triggered by drought stress in *Arabidopsis* (which I think is the case as it has never been reported in literature to my knowledge), no CTR1 translocation shall occur, making the nuclear function of CTR1 not possible. I think the drought response observed in Figure 6 is an artifact from the developed constitutive nuclear CTR1 line. Therefore, this figure requires to have proper controls that prove that drought stress is able to activate the nuclear translocation of CTR1. It would also benefit from a positive control showing that nuclear stabilized EIN3 is involved in drought tolerance, as this is postulated to be a CTR1-dependent action.

Despite these last two major comments, this work is outstanding, highly original and extremely interesting. Given some revisions or additional clarifications, this work has a major impact in the field of ethylene biology.

Below are some other detailed comments, including more information on the minor and major concerns.

Introduction.

The authors mention that ethylene plays a role in drought stress resilience; however the references provided are not related to drought stress. To be honest, of all the abiotic stressors, drought stress is most likely least linked with ethylene. There are a few older reports on the link between drought stress and ethylene, but only in less commonly studied crops, and not in model plants. I suggest removing drought stress from the list or providing a reference. Especially because Figure 6 deals with the role of nuclear CTR1 in regulating drought stress tolerance, they need to provide explicit literature background information that ethylene is involved in drought stress tolerance in Arabidopsis.

Include the function of the EBF's in regulating EIN3 stability in the introduction.

The authors question how the release of an ethylene signal can lead to a rapid growth recovery. They might want to include the work in tomato on receptor abundance and ethylene feedback on receptor expression in the context of this question (e.g. doi: 10.3389/fpls.2019.01054 and doi.org/10.1016/j.postharvbio.2021.111573). And because the receptors are negative regulators, this feed-back mechanism might explain how ethylene sensitivity can be lost rapidly.

Results

Figure legend 1 indicates that a YFP was used while the main text speaks of a GFP fusion protein in the CTR1 complementation constructs. The authors should indicate which fluorophore was used and report it consistently.

The photobleaching experiment of supplemental Figure S3 is not proving that GFP:CTR1 is migrating to the nucleus within 150 seconds, but rather that the bleached GFP is able to recover. It is known that GFP proteins can recover rapidly from photobleaching if illuminated. I think this experiment is not contributing to the observation of the CTR1 migration, but is also not needed as this finding was already demonstrated via several other approaches.

Figure 1j describes the time kinetics of CTR1 migration. However, it is clear from this image that the background fluorescence (the ER localized CTR1?) is also increasing after 30-60 min (which is not the case for the EIN2 experiment of Supplemental Figure S4). So it is unclear if the GFP signal intensity increased over time (due to increase laser power or longer exposure time in the microscopy settings) or that CTR1 abundance increased in general (in both the nucleus and the ER), as suggested by the authors. This experiment should be clarified by means of fluorescence quantification of both the GFP channel and any other stable marker (e.g. ER-RK marker as used by the authors) over time, to rule out if the nuclear signal is caused by increased CTR1 stability (of CTR1 proteins already present in the nucleus before the ethylene treatment?) or by nuclear migration. The time-lapse western blot presented in figure 1k is also not convincing as this experiment shows CTR1 stabilization, not migration. Therefore, a time-lapse western blot of the total and nuclear fraction should be presented with and without CHX.

Furthermore, the removal of ethylene leads to a decrease of the nuclear GFP:CTR1 signal (Figure 1i), but not the ER-localized CTR1, which stays equally abundant over the 80 min time course. How can the authors explain this?

The protoplast assay of Supplemental Figure S5 should have a negative control where

GFP:CTR1 is expressed in wild type protoplasts with and without ACC/ethylene.

Why do the authors not show the growth reduction and recovery kinetics of the normal GFP:CTR1 line in the ctr1-2 mutant? This line should behave very similar as the wild-type as this line shows a normal ethylene-regulated nuclear CTR1 translocation (Fig 1c) and a nuclear CTR1 loss when ethylene is removed (Fig 1i). Similarly, why do they not show these growth responses for the GFP:CTR1(ctr1-8) line which has no nuclear localization, to analyze if the growth recovery is reduced or absent.

Figure 5b and Supplemental Figure S10: I wonder why there is a weak YFP signal outside of the nucleus for the CTR1-EBF1/2 BiFC experiment. Does this suggest that CTR1 and EBF1/2 also interact at the ER? Also, how come that CTR1 and EBF1/2 interact in the nucleus in the absence of ACC, as Figure 1 shows that ACC/ethylene is needed to translocate CTR1 to the nucleus. Is this an artifact of the 35S constructed used for the BiFC experiment?

Figure 5: The authors suggest, based on western blot results of EIN3, that the slower growth recovery (of Figure 4) is due to the nuclear action of CTR1 that stabilizes EIN3 (via the EBFs). However, the authors do not show that the removal of ethylene leads to a gradual EIN3 degradation and that this response is delayed or inhibited when CTR1 is retained in the nucleus. They only show that CTR1 stabilizes EIN3 in the absence of ethylene, not after removal of ethylene. I think this is crucial, as it hooks up data presented in Figure 4 and Figure 5, and would demonstrate the molecular action of CTR1 in this ethylene recovery phase.

Figure 6: What strikes me when looking at the stress images in Figure 6, is that the phenotype of the 35S:deltaNT-CTR1(ctr1-1) plants looks very similar as the wild type in the absence of stress. These mutants look way bigger in Figure 6, as compared to the phenotypes presented in Supplemental Figure S1. Are these lines not supposed to show a strong ctr1-2-like phenotype? Because the 35S:deltaNT-CTR1(ctr1-1) line has a strong constitutive nuclear localization of CTR1 (independent of ethylene signaling), it could lead to artifacts in the stress responses. Normally CTR1 should only be in the nucleus when ethylene signaling occurs, which is likely not to happen during drought stress. Unless the authors can show that their pCTR1:GFP:CTR1 line shows nuclear translocation upon drought stress? Therefore, I suggest to repeat these experiments and include a few extra controls. For example, how is the ctr1-2 mutant performing during drought? Or how does the pCTR1:GFP:CTR1 rescue line in ctr1-2 performs, which shows a "normal" ethylene-regulated nuclear translocation of CTR1. And as a positive control the authors should include a 35S:EIN3 line or ebf1,ebf2 mutant, which would show the involvement of nuclear stabilized EIN3 in drought tolerance. I think adding several of these controls would show that the drought resistant phenotype of the 35S:deltaNT-CTR1(ctr1-1) might be an artifact.

Reviewer #2 (Remarks to the Author):

This manuscript describes novel findings that advance our understanding of ethylene signaling and open new avenues for further investigation. CTR1 has long been known to be a negative regulator of ethylene responses with the CTR1 protein associating with the ethylene receptors at the ER membrane, but here the authors show in Arabidopsis that ethylene induces translocation of CTR1 into the nucleus where CTR1 seems to function as a positive regulator of ethylene responses, possibly through an inhibitory interaction with EBFs, which in turn leads to increased levels of the transcription factor EIN3, a known positive regulator of ethylene responses. The authors have convincingly demonstrated that ethylene induces translocation of CTR1 into the nucleus, that this nuclear localization involves control by the physical interaction of the CTR1 N-terminal

domain with the ethylene receptors. The nuclear translocation is independent of CTR1 kinase activity. Using kinetic growth analysis, the authors show that nuclear-localized CTR1 delays growth recovery upon the removal of ethylene, which inhibits plant growth. They correlate this delayed recovery to enhanced levels of the transcription factor EIN3, perhaps due to protein-protein interaction between CTR1 and EBF2, however the mechanism for the inhibition of EBF2 by CTR1 is not addressed in this study. The authors also show an interesting biological consequence of overexpressing the C-terminal domain of CTR1 in protecting Arabidopsis from abiotic stress. A variety of methods were used in this study, and the methodology is generally sound. In a few cases the interpretation and conclusions could be made clearer or with more caution.

Main suggestions:

The varying data/statements on the role of CTR1 in the nucleus are confusing and should be clarified. For example, the Abstract and Title indicate that CTR1 in the nucleus is a positive regulator of ethylene responses. However, the authors show that ethylene response does not require the nuclear localization of CTR1, given that the *ctr1-8* protein is not detected in the nucleus yet the *ctr1-8* mutant has a constitutive ethylene response. As the authors state, "This finding indicated that CTR1 nuclear movement does not control the primary ethylene response but instead may influence ethylene response kinetics by fine-tuning nuclear ethylene signaling" (page 8). This sentence on page 8, and the lack of *ctr1-8* in the nucleus, seem at odds with the paper's conclusion that "CTR1 enhances nuclear ethylene responses by stabilizing EIN3 (in the Discussion, page 13)". Moreover, to explain why the high, CTR1-induced levels of EIN3 do not confer the known phenotype of EIN3 overexpression, the authors speculate that the overexpressed CTR1 acts to suppress ethylene responses in the cytoplasm (page 11). This again feels in conflict with the main conclusion provided in the title of the paper. I suggest that the nuances of CTR1 action be clarified in the Discussion section.

The model in Figure 7 shows a "Secondary Ethylene Response", but I don't think this wording was used anywhere in the paper other than in this figure. Please explain what is meant by a secondary response.

The Figure 7 legend states, "After the nuclear translocation of EIN2-CEND, CTR1 is released from the receptors and relocates to the nucleus via an unknown mechanism." I don't think anything can be stated about the order and timing of the translocation of CTR1. In other words, the statement "after the nuclear translocation of EIN2-CEND" seems unrealistic even for a model. What is the basis for proposing this sequential order in the model? I presume there is almost always some basal ethylene, so the ethylene signaling system is not something that is fully turned off in the absence of ethylene and then turned on in a series of steps where CTR1 only goes to the nucleus after EIN2-CEND goes into the nucleus.

I think there should be a negative control for the Co-IP of Myc-CTR1 by GFP-EBF2, using some other Myc-tagged protein that is not expected to be pulled down by GFP-EBF2, to strengthen this interaction finding. Alternatively, there should be such a negative control for EBF1 and EBF2 in the Y2H and BiFC.

Why didn't the authors also test or show co-IP of CTR1 with EBF1?

It would be helpful to know whether the EIN3 constructs are actually expressed in the Y2H and BiFC experiments (Figs 5a, b), since the authors use this data to rule out a direct interaction between CTR1 and EIN3.

In the Discussion, the authors conclude, "We further discovered that the nuclear movement of CTR1 is tightly associated with a plant's response to abiotic stress" This sentence should be revised because the authors did not demonstrate that resilience to abiotic stress is a normal role of native CTR1 nor that the nuclear movement itself is the cause of the resilience. For the stress experiments, the authors used only an

overexpressed C-terminal domain of CTR1, and so they should adjust their conclusion to be more like their statement in the Abstract. (In addition, given that this effect was only shown in Arabidopsis, perhaps the conclusion should not be implied to be true for all plants.)

For the abiotic stress experiments, I think the authors should have included the ctr1-1 mutant and a line that overexpresses EIN3 as controls to see whether overexpressing the C-terminal domain of CTR1 provides even greater protection. Could the overexpression construct have caused gene silencing of CTR1 in those lines that were analyzed?

The abiotic resilience could be tested with the ctr1-8 version with and without the SV40 NLS in order to correlate the stress resilience with nuclear localization of CTR1-8.

Minor comments:

Please describe what kind of mutations are used for the following experiment for those who are unfamiliar with the mutant alleles: "Disruption of either EIN2 or EIN3/EIL1 did not prevent the nuclear accumulation of CTR1 (Fig. 1g)."

Since there is no quantification for the data in Figure 1h, the authors should not refer to "a large fraction" in this sentence: "a large fraction of GFP-CTR1 constitutively localized to the nucleus in etiolated seedlings, regardless of ACC treatment (Fig. 1h)..."

Cite Supplementary Figure 7 before the part of the sentence that says "both of which...." in the following sentence: "Similar kinetics of ethylene response and growth recovery was also observed in seedlings expressing active or inactive wild-type full-length CTR1 (35S:GFP-CTR1 and 35S:GFP-CTR1ctr1-1), both of which showed some level of constitutive CTR1 nuclear localization (Fig. 1h, 3c, and Supplementary Figure 7)."

For the list of ACC responses on page 5, reference 24 does not show ACC in pollen tube growth.

Minor edits:

In the Figure 5 legend, do not capitalize "B", and add the "S" to "35:GFP-..." for g. In the legend for 5h, replace "showed enhanced ACC response than" with "showed an enhanced ACC response compared to". Capitalize the first letter of "agrobacteria".

Reviewer #3 (Remarks to the Author):

My comments to this work can be found from the attached file.

Reviewer #3 Attachment on the following page.

The article by Park and co-authors reports a very interesting, eye-catching finding on the cytoplasmic-nuclear transport of the Raf-like CTR1 protein, an essential negative regulator of the ethylene signaling pathway, and provides another aspect of ethylene signaling regulation via the F-box protein EBF1/EBF2. The finding of CTR1 nuclear localization is convincing and exciting. My only concerns about the inference made in this work are 1) effects of the overexpressed proteins on ethylene signaling appear to be marginal and conditional, and it is undetermined whether the reported effects are also conferred by the native function of CTR1, 2) data acquisition and analyses for the laser scanning confocal microscopy (LSCM) primarily involved “independent samples” but not “paired samples” to compare effects of a treatment or genetic backgrounds. Large sample-to-sample variations are prone to false negatives or false positives in cell biology studies, 3) subcellular localizations and effects of the several different GFP-CTR1 variants involve data from independent transgenic lines (or events) in different genetic backgrounds, and it is unlikely to determine whether the different degrees of nuclear localization or effects are results of the protein levels or the treatment/genetic backgrounds, and 4) wording needs to be more precise to avoid misleading.

Specific comments:

1. According to the image data presented, as long as GFP-CTR1 exceeds a level, constitutive nuclear localization will be observed at various degrees. It is the level, instead of the transgene promoters (*CTR1p* or *35S*), that determines the percentage of nuclear localization. The authors, however, complicated their data presentations at the first place and gave the authors to an impression that GFP-CTR1 is constitutively nuclear localized by the *35S:GFP-CTR1* but not the *CTR1p:GFP-CTR1* transgene. As a fact, GFP-CTR1 by the *CTR1p:GFP-CTR1* transgene also exhibited a level of constitutive nuclear localization. It is misleading to describe the GFP-CTR1 movement to be “constitutive”. The wording “constitutive” needs to be revised for other GFP-CTR1 variants in this work.
2. Data acquisition and analyses from the laser scanning microscopy (LSCM) will be critical to the interpretation of GFP-CTR1 localizations in the study and need to be very carefully and rightly conducted.
 - Cells are in a three-dimensional shape, and an image acquired from a focal plane of the laser confocal microscopy (LSCM) only provides a slice of two-dimensional information. The constitutive nuclear fluorescence for lines with high GFP-CTR1 levels will be much evident than those with low levels, because strong nuclear fluorescence can be observed over a wide range of focal planes and weak fluorescence only observed within a narrower focal plane range (See the illustration below). This easily explains the observation of constitutive GFP-CTR1 nuclear localization of those *35S:GFP-CTR1* lines and partial nuclear localization of those *CTR1p:GFP-CTR1* lines. This also explains the uneven fluorescence (i.e., cell-to-cell variations) among cells of an image sample (for example, Fig 1f and 1g, where the fluorescence is much stronger in cells on both sides and weak in the center of the AgNO₃-treated hypocotyl, and the opposite for the ACC-treated). There will also be transgenic lines, regardless of the promoters, exhibiting various levels of the GFP-CTR1 proteins. In other words, one will observe constitutive nuclear localization for *CTR1p:GFP-CTR1* lines and partial nuclear localization for *35S:GFP-CTR1* lines. There will be *ctr1-2* lines exhibiting the WT phenotype while the GFP-CTR1 fluorescence barely detectable. It may be misleading to report constitutive GFP-CTR1 nuclear localizations only for *35S:GFP-CTR1* lines.
 - Following comment 1 (also the illustration), the cell-to-cell variation, generated

from different focal plane sampling, will distort the observed nuclear localization percentage reported in Fig 1 due to false negatives. Please describe how was the % of nuclear localized CTR1 measured to ensure that the analyses were properly conducted. When the data are presented in percentage, it means that that are nuclei truly determined without the fluorescence (did the authors conduct this measurement?). The authors will need to report the number of individual nuclei counted and the number of nuclei exhibiting the fluorescence (make sure that those without fluorescence is determined over the z-axis for). When the number of observed nuclei is less than 100, fraction but not percentage should be used; otherwise, the percentage is an estimate but not an actual percentage and cannot be subject to statistical analyses. There will be no error bars for the percentage/fraction; once the percentage or fractions are measured, it will be goodness of fit instead of *t*-test to determine statistical significance for data such as Fig 2c and 2i.

- The GFP-CTR1 nuclear localization affected by a treatment presented in most of the figures was not determined from the same samples after a treatment (i.e., dependent samples or paired samples that reveal the changes before-and-after a treatment), and the LSCM images comparing “treated” and “non-treated” (independent samples) cannot truly reveal effects of a treatment (or the mutations) because of complexity associated with different focal plane sampling and sample-to-sample variations as explained in comments 1 and 2. Without in-depth analyses, results from independent sampling in cell biology studies are prone to false positives/negatives.
- When “paired samples” are involved for LSCM image comparisons, caution should be taken to avoid focal plane shift that also affects the observation and inference (note, as an example, there is focal plane shift for Fig 1i/ 1j and Supplementary Fig 4; the presented images were not taken from the same focal planes, as inferred from the different outlines of adjacent cells). The focal plane shift makes it unlikely to determine whether the observed nuclear signal was a result of cytoplasmic-nuclear trafficking or simply stronger fluorescence signal captured from different focal planes. For instance, the total fluorescence

at 60 min was evidently much stronger than that at time 0 to 30 min for Fig. 1j. It is also recommended to present full images of a LSCM image (can be presented in expanded information) over the z-axis for some essential images, instead of a crop for a cell, so that more information can be available to assess the nuclear localization status from different cells. Besides, the cropped images for a single cell cannot be “presentative”, and one cannot make inference from single samples in the absence of sufficient information from other cells.

- Involving independent transgenic lines (events), there need negative controls (non-treated) for the GFP-CTR1 localizations/percentage to contrast effects of the AgNO₃ and ACC treatment in the indicated mutant backgrounds in Fig 1 (including *ctr1-2*, *etr1-1*, *ein2-5*, and *ein3 eil1*).

- Following the aforementioned comments and explanations on the problems associated with “independent sampling”, I strongly recommend the authors to carefully conduct “paired sampling” and avoid focal plane shift to provide better quantification and more reliable image evidence to characterize the cytoplasmic-nuclear transport event. This comment is also important because different transgenic lines in those genetic backgrounds are involved in this study and there are always cell-to-cell variations for a same line. Analyzing paired samples will minimize the uncertainty resulted from different protein levels. To avoid focal plane shift, which always happens for prolonged observations, one will need to re-adjust the focal plane manually to the time 0 plane by referencing the cell outlines. In this study, ACC treatment was conducted for 2 hrs. Given that a short ACC treatment is sufficient to induction of the ethylene response, the GFP-CTR1 movement in response to ethylene for paired samples can be easily conducted (as reported in Fig 1j and 1l).

3. The nuclear GFP-CTR1 rises from about 10% (AgNO₃) to 60% (ACC) (a 6-fold increase; Fig 1c), and the nuclear GFP-CTR1 is undetected for the silver treated (Fig 1i). Can a prolonged exposure detect GFP-CTR1 for the silver treated? A negative control (non-treated) will be needed as well to demonstrate inhibition of CTR1 nuclear transport by silver. This comment applies to both WT, *ein2* and *ein3 eil1*. In the western blot (Fig 1i), please explain why and which Hsc70 isoform was used as the internal reference and why the ER marker BiP2 not used. Hsc70 could exist in the cytoplasm and nucleus, and it is the ER marker that may better determine if there is any ER contamination in the nuclear fraction.

4. In Materials and Methods, please describe whether the purified proteins for the *in vitro* assays were from soluble or re-solubilized inclusion bodies. The description for His6-EIN2^{WT} is misleading because this work does not involve the full-length EIN2.

5. The conclusion that GFP-CTR1 constitutively localizes to the nucleus in *etr2 ein4 ers2* needs to be carefully made because the transgenes in WT and the mutant are expressed at a different locus (i.e., independent transgenic event), and levels of the GFP-CTR1 may differ and affect the inference. This argument agrees with the fact that GFP-CTR1 constitutively localize to the nucleus when the transgene is driven by the 35S promoter in WT.

Moreover, the constitutive nuclear localization of GFP-CTR1 in *etr2 ein4 ers2* appears to agree with the absence of ER-associated CTR1 reported by Gao and co-authors (Gao et al., 2003). On the other hand, the soluble fraction by Gao *et al.* involves 8,000×g and 10,000× centrifugations, which will greatly remove the nuclei, indicating that the detected CTR1 in the soluble fraction is from cytosol but not from the nucleus. This comment also applies to the CTR1-8 protein. *ctr1-8* is a relatively weak allele, producing a mild mutant phenotype throughout development, and CTR1-8 in part associates with the membrane. When the protein is in excess amount, its ER

localization is expected. On the other hand, when GFP-CTR1-8 in transgenic lines is expressed at a low level, it is less likely to be observed a nuclear signal, which does not mean the absence of nuclear localization. The different degrees of nuclear GFP-CTR1 localization may in part arise from different protein levels of those independent transgenic lines. To confirm ethylene-induced CTR1 localization to the nucleus, it is critical to provide evidence without involving transgene expression (determining nuclear CTR1 and CTR1-8 by immunoblotting). This will not be a challenging comment because one of the authors has publications involving anti-CTR1 antibodies.

6. There lacks a non-treated control for the percentage of CTR1 nuclear localization to determine effect of the AgNO₃ or ACC treatment.

It is important to biochemically determine the subcellular localizations of the endogenous CTR1, CTR1-8, and CTR1-1 to support the model and the LSCM results proposed in this study (good anti-CTR1 antibodies are available from one of the co-authors). To determine subcellular localizations of endogenous CTR1-8, CTR1-1, or CTR1 (in the triple mutant), proteins from the soluble, membrane, nuclear fractions will need to be determined by immunoblot analyses. Previous studies on CTR1/CTR1-8 were only conducted for the membrane and soluble fractions (Gao et al., 2003), which do not address whether the protein is in the nucleus.

7. At present, the inference is interfered by excess amount of the GFP-CTR1 proteins. Biological effects of the overexpressed GFP-CTR1 are marginal and not easily detected, interference of the excess amount protein on native CTR1 function or ethylene signaling needs to be considered.
8. The seedling growth recovery rate (Fig 4) was determined from transgenic lines involving independent transgenes at different loci. Because of the effects (or difference) are marginal, with large variations (data are means and SEs, $n \geq 6$, meaning that the SD for each sample mean will be approximately $SE \times \sqrt{6}$), and can only be observed conditionally, it is unlikely to determine whether the differences are results of different protein levels or the expressed proteins. Alternatively, excess amount of GFP-CTR1 or GFP-CTR1-1 may disturb the native CTR1 function (this comment applies to data for the ACC dose-response assay in Fig 5g and 5h). Studies on the GFP-CTR1-8-SV40 protein needs a positive control (GFP-CTR1-SV40) to determine whether the SV40 signal may impair function of the CTR1 or CTR1-8 protein. This control is important to addressing the biological significance of constitutive localization of CTR1 in the nucleus. Conversely, investigating whether an NES signal may impair CTR1 function is equally important.

The conversion of seedling growth to the recovery rate may somehow distort the growth curves for different genotypes, and the rate could be misleading. For example, the seedling hypocotyl will be longer for WT than for *ctr1* mutants, thus a larger denominator for WT than for *ctr1* mutants; as a result, the actual growth recovery will be under estimated for WT and overestimated for *ctr1* mutants. This also explains for the higher growth rate for *ctr1-8* and the two transgenic lines than for WT (Supplementary Fig 8b).

The conclusion for the different growth recovery reported in Fig 4 is somehow descriptive or arbitrary, lacking statistical testing. The authors will need to report the support for their inference with statistical evidence. With the conversion of growth recovery to recovery rate, the data are somehow distorted by the different denominators (especially for Fig 4b), and the authors will need to determine a reasonable way to analyze the data. I might not have problems for the interpretations for GFP-CTR1-8-SV40 of Fig 4b; at least, the difference appears prominent.

The description “the GFP-CTR1 transgene ...restored the slower growth recovery of *ctr1-8* to levels comparable to those of the WT” needs to be revised. The transgenic

- line appears to have a greater growth recovery rate than WT as shown in Fig 4b.
9. Levels of EIN3 shown in Fig 5d may not support the differences in the seedling hypocotyl lengths (Fig 5g). The levels of Fig 5e may not support the *ctr1-8* seedling hypocotyl lengths (Supplementary Fig 8a and 8b). EIN3 levels of *35S:ΔNT-ctr1-1* at 0 ACC were similar to that of WT at 1-10 mM ACC, but the difference between the seedling hypocotyl lengths of WT and *35S:ΔNT-ctr1-1* was not statistically significant at 0 ACC (Fig 5d and 5g). EIN3 levels of *ctr1-8* at 0 ACC are already much greater than that in WT treated with 10 mM ACC, but *ctr1-8* seedlings at 0 ACC are known longer than 10 mM ACC-treated WT (the authors did not report the actual seedling hypocotyl length for 10 mM ACC-treated WT). It is unexplained for why the similar level of EIN3 of 1-10 mM ACC-treated WT did not result in similar levels of the ethylene response.
 10. Data for the resilience to abiotic stress by the *35S:GFP-ΔNT-CTR1^{ctr1-1}* need to be carefully addressed. This evidence is only obtained from a single transgenic event, and it is to be determined whether it is the transgene insertion at certain locus (or loci; there may be more than one transgenes in this line) or the effects of GFP-ΔNT-CTR1-1 fragment. Moreover, EIN3 levels determined from *35S:GFP-ΔNT-CTR1^{ctr1-1}* do not agree with the rosette phenotype shown in Fig 6. The highly elevated EIN3 is expected to result in strong rosette growth inhibitions, and the *35S:GFP-ΔNT-CTR1^{ctr1-1}* plants are not different from WT.

The section “The increased nuclear accumulation of CTR1 enhances stress tolerance to abiotic stress” is misleading. It the *35S:GFP-ΔNT-CTR1^{ctr1-1}* transgene, its insertion at certain loci, or the truncated ΔNT-CTR1-1, instead of CTR1, that results in the phenotype. To support the argument by the authors, one will need data from the corresponding transgenic plants.

Other comments or suggestions:

1. The antibodies for EIN3 were from Agrisera, and there is no information about the validity of this commercial antibodies from the webpage (<https://www.agrisera.com/en/artiklar/ein3.html>). Many false positive reports are resulted from inappropriate antibodies (Baker, 2005), and can the authors provide any evidence that validate the anti-EIN3 antibodies including negative positive/controls (albeit the antibodies were previously cited in PNAS)? This comment is raised because EIN3 levels and degrees of the seedling hypocotyl growth inhibition are not tightly associated (see comment 8), and personally I had poor experience with the commercial antibodies. The increase immune signal in response to ACC does not mean it is the EIN3 protein. Similarly, please report the source of anti-EIN2 antibodies. Besides, please show the full-paged immunoblots for EIN2, EIN3, and CTR1 in supplementary files for the purpose of data transparency, with the molecular-weight markers.
2. For the seedling hypocotyl measurements (not the growth rates in Supplementary Fig 8), data for ACC- or ethylene-treated WT as controls are absent. Besides, the graphic presentations for the dose-response curves are erroneous. The distance between each concentration point at the x-axis should not be equal because the dosage is not linear.
3. The section title “The CTR1 N-terminus inhibits CTR1 nuclear trafficking” may be misleading. Nuclear trafficking still occurs for GFP-CTR1 that has the N-terminus. It is the association of CTR1 with the ethylene receptors via the N-terminus to retain the protein at the ER.
4. In introduction, the description for CTR1-mediated EIN2 phosphorylation and ethylene signaling repression may need updating another study showing full-length

EIN2 localization to the nucleus, complex EIN2 cleavages, and weak association of EIN2 phosphorylation status and ethylene signaling activation to better describe other aspects of ethylene signaling (Wen et al., 2012; Zhang et al., 2020). Inference made from different aspects of findings, instead of limited to the canonical model that may change from time to time and prevent new findings, is important to development of new discovery.

5. I noticed that some of the LSCM images are from cells with a relatively small size, and it appears that those images were taken from cells of tip regions (root tips or apical regions) of a seedling. Probably I was wrong; however, if any LSCM images were taken from cells of tip regions, background noises that gives false positive signals should be considered (Zhang et al., 2020).
6. Seeding hypocotyl measurement needs to be described in Materials and Methods; the WT seedlings appear to be unbelievably short (<6 mm). The seedling hypocotyl measurement is also problematic for WT in Fig 2e; it is only about 5 mm in length. Please refer to the typical seedling hypocotyl measurement by Hua and Meyerowitz, where the WT seedling is about ≥ 12 mm (Hua and Meyerowitz, 1998). A proper measurement may better unveil the difference in degrees of the ethylene response. When the hypocotyls are relatively short, minor differences cannot be unveiled and the inference will be made differently.
7. For graphical data presentations, error bars should be presented on both sides of a bar, and the y-axis should start from 0. The data analyses involve inferential statistics, and it is SE instead of SD to be presented (Cumming et al., 2007).
8. Fig 2d, the percentage for nuclear localized CTR1, fix CTR1 to Δ NT-CTR1.
9. In addition to the co-expression, data for Supplementary Fig 11 needs controls for EIN2-C and ENAP1 alone to show their subcellular localizations in the absence of each other.
10. The LSCM images for Supplementary Fig 4 for the nuclear EIN2 look more like at the ER network surrounding the nucleus. The nuclear EIN2 is characteristic of speckles that are revealed in independent LSCM images (Qiao et al., 2012; Zhang et al., 2020).

Baker, M. (2005). Reproducibility crisis: Blame it on the antibodies. *Nature* 521:274-276.

Cumming, G., Fidler, F., and Vaux, D.L. (2007). Error bars in experimental biology. *The Journal of cell biology* 177:7-11.

Gao, Z., Chen, Y.F., Randlett, M.D., Zhao, X.C., Findell, J.L., Kieber, J.J., and Schaller, G.E. (2003). Localization of the Raf-like kinase CTR1 to the endoplasmic reticulum of Arabidopsis through participation in ethylene receptor signaling complexes. *J Biol Chem* 278:34725-34732.

Hua, J., and Meyerowitz, E.M. (1998). Ethylene responses are negatively regulated by a receptor gene family in *Arabidopsis thaliana*. *Cell* 94:261-271.

Qiao, H., Shen, Z., Huang, S.-s.C., Schmitz, R.J., Urich, M.A., Briggs, S.P., and Ecker, J.R. (2012). Processing and Subcellular Trafficking of ER-Tethered EIN2 Control Response to Ethylene Gas. *Science* 338:390-393.

Wen, X., Zhang, C., Ji, Y., Zhao, Q., He, W., An, F., Jiang, L., and Guo, H. (2012). Activation of ethylene signaling is mediated by nuclear translocation of the cleaved EIN2 carboxyl terminus. *Cell research* 22:1613-1616.

Zhang, J., Chen, Y., Lu, J., Zhang, Y., and Wen, C.-K. (2020). Uncertainty of EIN2^{Ser645/Ser924} Inactivation by CTR1-Mediated Phosphorylation Reveals the Complexity of Ethylene Signaling. *Plant*

Communications 1.

REVIEWER COMMENTS

Reviewer #1 (Remarks to the Author):

The work of Park et al provides compelling evidence for a novel function of CTR1 in ethylene signaling. They nicely demonstrated that CTR1 is able to migrate to the nucleus after an ethylene signal has been perceived. This information is totally novel and quite exceptional, making it a big breakthrough in our understanding of the ethylene signaling pathway. The release of CTR1 from the ethylene receptor at the ER seems a crucial process that enables the CTR1 nuclear translocation, as demonstrated by the use of different reporter lines. The authors also show convincingly that CTR1 is interacting with the EBFs in the nucleus, and that this interaction is probably stabilizing EIN3 proteins. As such, the authors nicely reveal a first important glimpse on the possible mode of action of nuclear CTR1. I feel these findings provide a lot of novelty to the community, as they reveal a totally novel mechanism how ethylene signaling can occur independent of EIN2 and how ethylene nuclear events are fine-tuned.

I have a few unresolved questions that might be addressed or discussed in the manuscript or elucidated with a few extra simple experiments. For example, why is the CTR1 fluorescence increasing after an ethylene treatment at the ER and the nucleus, while the CTR1 fluorescence level is not decreasing at the EC (only the nuclear signal) when the ethylene signal is removed. Another small thin I noticed is that the CTR1-EBF interaction also occurs outside of the nucleus (Figure 5), which is not discussed in the manuscript. What would this mean or is this an artifact of the 35S construct used?

Response 1: We found the same comments in the introduction and results section below. Please see the response in below (**Response 9, 12**)

The latter brings me to two key points that require some more substantial attention. The authors often use 35S constructs or lines in which CTR1 is constitutively present in the nucleus, even in the absence of ethylene. I understand the logic behind this, but I wonder if it could also not lead to false interpretations. Under normal conditions, CTR1 levels are supposed to reduce in the nucleus if the ethylene signal drops. This reasoning actually is used by the authors to link the potential nuclear function of CTR1 with a delayed response upon ethylene release. However, the growth kinetics experiments (Figure 4) lack data with lines that do not have the 35S or constitutive nuclear localized CTR1. I wonder how these lines would perform, as the main reasoning is that ethylene will lead to nuclear localization of CTR1 and ethylene release would remove that nuclear localization. I wonder if the GFP:CTR1 line in the *ctr1-2* mutant has a normal wild type growth response, or has the GFP:CTR1(*ctr1-8*) line (no CTR1 nuclear transfer) an abolished growth response? If the release of ethylene is the actual physiological event where nuclear CTR1 is operational via the action of EIN3 stability, then the authors should show that EIN3 stabilization occurs during the release of ethylene, and not during the onset of ethylene signaling. I believe they have all the CTR1-lines available to show this EIN3 response.

Response 2: We found the same comments in the introduction and results section below. Please see the response in below (**Response 11, 13**).

A final remark that also concerns me, is the use of the constitutively nuclear localized CTR1 is its role during stress resilience, especially drought stress. First of all, ethylene does not play a dominant role in drought stress responses (opposed to its role in salinity stress), and yet the authors make it seem that the nuclear CTR1 is important to respond to drought stress. If ethylene production (or signaling) is not triggered by drought stress in Arabidopsis (which I think is the case as it has never been reported in literature to my knowledge), no CTR1 translocation shall occur, making the nuclear function of CTR1 not possible. I think the drought response observed in Figure 6 is an artifact from the developed constitutive nuclear CTR1 line. Therefore, this figure requires to have proper controls that prove that drought stress is able to activate the nuclear translocation of CTR1. It would also benefit from a positive control showing that nuclear stabilized EIN3 is involved in drought tolerance, as this is postulated to be a CTR1-dependent action.

Response 3: We found the same comments in the introduction and results section below. Please see the response in below (**Response 14**).

Despite these last two major comments, this work is outstanding, highly original and extremely interesting. Given some revisions or additional clarifications, this work has a major impact in the field of ethylene biology.

Below are some other detailed comments, including more information on the minor and major concerns.

Introduction.

The authors mention that ethylene plays a role in drought stress resilience; however the references provided are not related to drought stress. To be honest, of all the abiotic stressors, drought stress is most likely least linked with ethylene. There are a few older reports on the link between drought stress and ethylene, but only in less commonly studied crops, and not in model plants. I suggest removing drought stress from the list or providing a reference. Especially because Figure 6 deals with the role of nuclear CTR1 in regulating drought stress tolerance, they need to provide explicit literature background information that ethylene is involved in drought stress tolerance in Arabidopsis.

Response 4: Thank you for the comments. We removed the drought stress from the list and rephrased the sentences.

Include the function of the EBF's in regulating EIN3 stability in the introduction.

Response 5: We added the sentences regarding the role of EBF on EIN3 stability regulation in the introduction (lines 67-69).

The authors question how the release of an ethylene signal can lead to a rapid growth recovery. They might want to include the work in tomato on receptor abundance and ethylene feedback on receptor expression in the context of this question (e.g. doi: 10.3389/fpls.2019.01054 and doi.org/10.1016/j.postharvbio.2021.111573). And because the receptors are negative regulators, this feed-back mechanism might explain how ethylene sensitivity can be lost rapidly.

Response 6: Thank you for the comments. We included the work on tomato and the potential feedback regulation of ethylene response via the modulation of the receptor abundance and gene expression of the receptors in the text (lines 72-75).

Results

Figure legend 1 indicates that a YFP was used while the main text speaks of a GFP fusion protein in the CTR1 complementation constructs. The authors should indicate which fluorophore was used and report it consistently.

Response 7: We changed the text. Most transgenic lines used in this study express CTR1 in a form of GFP-fused CTR1. But the lines that coexpress CTR1 and ER-RK marker protein express YFP-fused CTR1 (Fig. 1c and Fig. 2f).

The photobleaching experiment of supplemental Figure S3 is not proving that GFP:CTR1 is migrating to the nucleus within 150 seconds, but rather that the bleached GFP is able to recover. It is known that GFP proteins can recover rapidly from photobleaching if illuminated. I think this experiment is not contributing to the observation of the CTR1 migration, but is also not needed as this finding was already demonstrated via several other approaches.

Response 8: We removed the figure as suggested.

Figure 1j describes the time kinetics of CTR1 migration. However, it is clear from this image that the background fluorescence (the ER localized CTR1?) is also increasing after 30-60 min (which is not the case for the EIN2 experiment of Supplemental Figure S4). So it is unclear if the GFP signal intensity increased over time (due to increase laser power or longer exposure time in the microscopy settings) or that CTR1 abundance increased in general (in both the nucleus and the ER), as suggested by the authors. This experiment should be clarified by means of fluorescence quantification of both the GFP channel and any other stable marker (e.g. ER-RK marker as used by the authors) over time, to rule of if the nuclear signal is caused by increased CTR1 stability (of CTR1 proteins already present in the nucleus before the ethylene treatment?) or by nuclear migration. The time-lapse western blot presented in figure 1k is also not convincing as this experiment shows CTR1 stabilization, not migration. Therefore, a time-lapse western blot of the total and nuclear fraction should be presented with and without CHX.

Furthermore, the removal of ethylene leads to a decrease of the nuclear GFP:CTR1 signal (Figure 1i), but not the ER-localized CTR1, which stays equally abundant over the 80 min time course. How can the authors explain this?

Response 9: Thank you for the comments. The use of seedling with ER-RK marker is good idea, but we found that YFP fluorescence of CTR1 is greatly photobleached in our experimental setting, which unable to use the coexpression lines for time-lapse experiments. To investigate if

the presented data resulted from possible focal plane changes or prolonged laser exposure, we repeated the experiments several times and included the Z-axis in data processing to minimize the effects of focal plane change. The newly acquired data (merged 10 successive Z-stack images at each time point) using *CTR1p:GFP-CTR1/ctr1-2* showed that ACC appeared not to influence the intensity of CTR1 fluorescence in the cytoplasm but increased the fluorescence of CTR1 in the nucleus over time compared to that in the seedlings without ACC treatment at time 0 (**Fig. 1k** and **Supplementary Fig. 6**). Consistent with the results in the original submission, CTR1 amassed in the nucleus approximately 30 min after ACC treatment. Normally, we image seedlings by placing them on a microscope slide followed by ACC solution mount and obtain images at different time points. During this process, seedlings in ACC solution gradually change their position over time. Thus, we speculate that the increased GFP-CTR1 signals in Fig.1i in the original submission may likely be due to focal plane changes of the cells while locating the same cells at the previous time points. We are sorry about the inadequate quality of the data in the original submission. We replaced it with the newly generated data obtained with a more careful imaging process and Z-axis inclusion (**Fig. 1k**).

For Fig.1j, we performed this experiment in a similar way to that of Fig.1i in the original submission. Thus, we repeated the experiment as described above. The overall signals in both the nucleus and ER were gradually reduced until 40 min after ethylene removal, followed by a steep reduction in GFP signals at 60 min after ethylene removal (**Fig. 1l**). To further investigate the protein abundance of CTR1 after ethylene removal, we examined the protein abundance of GFP-CTR1 after ethylene removal using immunoblotting. Seedlings were pretreated with ethylene for 2 h and harvested at different time intervals after ethylene removal. As shown in **Supplementary Fig. 7**, the steady-state level of GFP-CTR1 gradually reduces after the removal of ethylene.

Furthermore, immunoblotting analysis for ACC-induced CTR1 localization in the nucleus in the presence and absence of ACC and silver nitrate (**Fig. 1j** and **Supplementary Fig. 4**) showed that some minor levels of CTR1 were detected in the nucleus in seedlings without any treatment, likely due to the basal levels of ethylene in etiolated seedlings. However, the lower levels of nuclear-localized CTR1 detected in no treatment conditions were barely detected in seedlings grown on media with silver nitrate. This result suggests that the nuclear movement of CTR1 is ethylene-dependent and that CTR1 is not present in the nucleus in the absence of ethylene. Together with other supporting data in the manuscript, we feel that the newly added data is sufficient to prove that CTR1 is absent in the nucleus and its nuclear movement is triggered by ethylene.

For the connection between CTR1 nuclear movement and ethylene-induced stabilization of CTR1, our data suggest that the nuclear movement of CTR1 is triggered by the combination of migration and stabilization of cytosolic CTR1. Ethylene initially triggers the movement of CTR1 to the nucleus by displacing CTR1 from the ER. However, over time, the nuclear movement of CTR1 is facilitated by ethylene-induced increased CTR1 abundance in the cytoplasm. Ethylene increases CTR1 protein abundance. Therefore, when the levels of CTR1 reach above the threshold that can't be tethered by the receptors, free cytosolic unbound CTR1 is likely

transported to the nucleus. This is supported by the fact that CTR1 overexpression lines showed constitutive nuclear localization of CTR1 in the presence of silver nitrate (**Fig. 1h** and **Fig. 3d**).

The protoplast assay of Supplemental Figure S5 should have a negative control where GFP:CTR1 is expressed in wild type protoplasts with and without ACC/ethylene.

Response 10: We provided the suggested data (**Supplementary Fig. 9**)

Why do the authors not show the growth reduction and recovery kinetics of the normal GFP:CTR1 line in the *ctr1-2* mutant? This line should behave very similar as the wild-type as this line shows a normal ethylene-regulated nuclear CTR1 translocation (Fig 1c) and a nuclear CTR1 loss when ethylene is removed (Fig 1i). Similarly, why do they not show these growth responses for the GFP:CTR1(*ctr1-8*) line which has no nuclear localization, to analyze if the growth recovery is reduced or absent.

Response 11: Thank you for the comments. We provided the suggested data (**Supplementary Fig. 12**). A native promoter-driven expression of GFP-CTR1 (*pCTR1:GFP-gCTR1/ctr1-2*), complemented *ctr1-2*, resulting in comparable ethylene response and recovery kinetics to that of the wild-type. Similarly, *pCTR1:GFP-gCTR1^{ctr1-8}/ctr1-2* exhibited similar levels of ethylene response to the wild-type, but the *CTR1^{ctr1-8}* transgene did not rescue the recovery kinetics to that of the WT.

Figure 5b and Supplemental Figure S10: I wonder why there is a weak YFP signal outside of the nucleus for the CTR1-EBF1/2 BiFC experiment. Does this suggest that CTR1 and EBF1/2 also interact at the ER? Also, how come that CTR1 and EBF1/2 interact in the nucleus in the absence of ACC, as Figure 1 shows that ACC/ethylene is needed to translocate CTR1 to the nucleus. Is this an artifact of the 35S construct used for the BiFC experiment?

Response 12: The weak signal outside of the nucleus is not an artifact. We repeated the experiments several times with similar results. Our preliminary results showed that CTR1 itself also localizes not only to the nucleus but also to unknown cytoplasmic puncta in plants. Since EBF mRNA is trapped in the processing body by the C-terminus of EIN2 (1-2), the puncta outside of the nucleus where CTR1 and EBF2 interact may be the P-body. We did not mention this in the text as it is preliminary and requires further study.

The interaction of CTR1 and EBF2 in the nucleus in the absence of ACC is not an artifact. We speculate that the excess CTR1 expressed from the 35S promoter translocates into the nucleus due to the limited number of ethylene receptors that can hold the excess CTR1 in the ER. This is similar to what we observed in the CTR1 overexpression lines (**Fig.1h**), in which CTR1 localizes in the nucleus even with silver nitrate treatment.

References

1. EIN2-directed translational regulation of ethylene signaling in Arabidopsis (2015). Li et al., Cell.
2. Gene-specific translation regulation mediated by the hormone-signaling molecule EIN2 (2015). Merchane et al., Cell.

Figure 5: The authors suggest, based on western blot results of EIN3, that the slower growth recovery (of Figure 4) is due to the nuclear action of CTR1 that stabilizes EIN3 (via the EBFs). However, the authors do not show that the removal of ethylene leads to a gradual EIN3 degradation and that this response is delayed or inhibited when CTR1 is retained in the nucleus. They only show that CTR1 stabilizes EIN3 in the absence of ethylene, not after removal of ethylene. I think this is crucial, as it hooks up data presented in Figure 4 and Figure 5, and would demonstrate the molecular action of CTR1 in this ethylene recovery phase.

Response 13: Thank you for the comments. We provided the suggested data (**Fig. 5g**). The steady-state level of EIN3 protein in 35S: Δ NT-CTR1^{ctr1-1} is significantly higher than that in the WT at different time points after the removal of ACC, supporting the nuclear role of CTR1 in the stabilization of EIN3.

Figure 6: What strikes me when looking at the stress images in Figure 6, is that the phenotype of the 35S: Δ NT-CTR1(ctr1-1) plants looks very similar as the wild type in the absence of stress. These mutants look way bigger in Figure 6, as compared to the phenotypes presented in Supplemental Figure S1. Are these lines not supposed to show a strong ctr1-2-like phenotype? Because the 35S: Δ NT-CTR1(ctr1-1) line has a strong constitutive nuclear localization of CTR1 (independent of ethylene signaling), it could lead to artifacts in the stress responses. Normally CTR1 should only be in the nucleus when ethylene signaling occurs, which is likely not to happen during drought stress. Unless the authors can show that their pCTR1:GFP:CTR1 line shows nuclear translocation upon drought stress? Therefore, I suggest to repeat these experiments and include a few extra controls. For example, how is the ctr1-2 mutant performing during drought? Or how does the pCTR1:GFP:CTR1 rescue line in ctr1-2 performs, which shows a "normal" ethylene-regulated nuclear translocation of CTR1. And as a positive control the authors should include a 35S:EIN3 line or ebf1,ebf2 mutant, which would show the involvement of nuclear stabilized EIN3 in drought tolerance. I think adding several of these controls would show that the drought resistant phenotype of the 35S: Δ NT-CTR1(ctr1-1) might be an artifact.

Response 14: Thank you for the comments. To address the comments and confirm the presented results, we performed several experiments that included additional transgenic lines and controls. We included a substantial number of newly acquired results that support the role of nuclear CTR1 in stress resilience to salt and drought stress.

1. Wild-type (WT)-like phenotypes of 35S: Δ NT-CTR1^{ctr1-1} line (#14) was unexpected to us as well, yet the results were repeated. To clarify misunderstanding of the reviewer, the seedling photos in the Supplemental Figure S1 are transgenic plants expressing WT CTR1 and CTR1 variants from a CTR1 native promoter in ctr1-2 mutant background, not CTR1 overexpression lines that we used to analyze stress responses.

To address the comment, we included an additional independent line of 35S: Δ NT-CTR1^{ctr1-1} line (#28) and tested the salt and drought response. The #28 line also expressed higher levels of EIN3 and displayed significantly enhanced salt and drought tolerance than the WT (**Fig. 6j** and **Supplementary Fig. 18a**). Furthermore, to eliminate the possibility that the enhanced stress resilience of the 35S: Δ NT-CTR1^{ctr1-1} line is specific to the truncated form of CTR1, we examined the salt and drought stress responses of seedlings expressing full-length CTR1 (35S:GFP-

CTR1^{ctr1-1}) (**Fig. 6j** and **Supplementary Fig. 18a**) and found that it showed similar levels of salt and drought resistance to *35S:ΔNT-CTR1^{ctr1-1}*. Our localization data and ethylene response kinetics analysis showed that *35S:GFP-CTR1^{ctr1-1}* displayed some levels of constitutive CTR1 nuclear localization and significantly slower growth recovery kinetics than the WT (**Fig. 3a-d** and **Fig. 4b**). Intriguingly, unlike *35S:ΔNT-CTR1^{ctr1-1}*, *35S:GFP-CTR1^{ctr1-1}* seedlings are significantly smaller than the WT and *35S:ΔNT-CTR1^{ctr1-1}*. Moreover, the hypocotyl length of *35S:GFP-CTR1^{ctr1-1}* is significantly shorter than the WT over a broad range of ACC concentrations (**Supplementary Fig. 18d**). Examination of EIN3 levels in *35S:GFP-CTR1^{ctr1-1}* also showed that it produced higher levels of EIN3 than WT (**Supplementary Fig. 18b-c**). We speculate that the unrelatedness of the higher levels of EIN3 and WT-like phenotype of *35S:ΔNT-CTR1^{ctr1-1}* could be related to their truncated, overexpression and an unknown role in the cytoplasm or the nucleus via interaction with other cellular processes. Given the significantly smaller seedling size of *35S:GFP-CTR1^{ctr1-1}* than that of the *35S:ΔNT-CTR1^{ctr1-1}*, the truncated form of CTR1 may have stronger effects than the full-length CTR1 in terms of seedling size via the unknown mechanism.

2. Besides, we examined the salt/drought stress response of *35S:GFP-CTR1^{ctr1-8}* and *ctr1-8* seedlings (**Fig. 6e** and **6k**). The localization data showed that CTR1^{CTR1-8} proteins do not translocate into the nucleus in response to ACC regardless of which promoters are used. Therefore, one would think that *35S:GFP-CTR1^{ctr1-8}* seedlings would show a similar salt/drought response to that of WT seedlings as it does not affect EIN3 levels in the nucleus. We found that overexpression of CTR1^{CTR1-8} protein did not enhance salt and drought tolerance, showing a similar stress response to the WT (**Fig. 6e, 6k, Supplementary Fig. 19** and **20**). However, *ctr1-8*, a moderate constitutive ethylene response mutant with a slower growth recovery kinetics, showed a strong salt/drought tolerance phenotype. Furthermore, the immunoblotting analysis showed that *35S:GFP-CTR1^{ctr1-8}* produced a similar level of EIN3 to that of the WT (**Supplementary Fig. 18c**) despite that it produced comparable levels of CTR1^{ctr1-8} protein to CTR1^{ctr1-1} in *35S:GFP-CTR1^{ctr1-1}*. These results show that enhanced stress resilience to salt and drought requires nuclear-localized CTR1.

3. To demonstrate that the enhanced stress tolerance to salt and drought by nuclear-localized CTR1 is not simply due to the overexpression of the CTR1 variant in the nucleus, we examined the salt and drought stress responses of two additional lines expressing CTR1^{ctr1-8} or CTR1^{ctr1-8}-SV40 protein from its native promoter in the *ctr1-8* background (*CTR1p:GFP-CTR1^{ctr1-8}/ctr1-8* and *CTR1p:GFP-CTR1^{ctr1-8}-SV40/ctr1-8*) (**Fig. 6f** and **6l**). We found that *CTR1p:GFP-CTR1^{ctr1-8}/ctr1-8* lines are significantly bigger than the *ctr1-8*, but slightly smaller than the WT, and showed similarly or only slightly enhanced salt/drought tolerance to the WT. However, *CTR1p:GFP-CTR1^{ctr1-8}-SV40/ctr1-8* showed comparable levels of stress resilience to the *ctr1-8* and displayed a similar size to the *ctr1-8*. We speculate that *CTR1p:GFP-CTR1^{ctr1-8}-SV40/ctr1-8* exhibited a similar stress response to the WT despite the slower recovery kinetics than that of the *ctr1-8* (**Fig. 4c**) because the levels of EIN3 in the *ctr1-8* already met the threshold levels that could confer stress resistance. These results are likely due to the forced nuclear-localization of CTR1^{ctr1-8} protein by SV40 NLS stabilizes EIN3 in the nucleus without affecting cytosol ethylene response. Whereas, CTR1^{ctr1-8} protein expressed from the transgene, which does not translocate into the nucleus, inhibits ethylene response in the cytoplasm without stabilizing EIN3

in the nucleus, thus rescuing the smaller size of *ctr1-8* and restoring the response to stress to the WT levels. Together, these results suggest that the effect of enhanced drought/salt resilience phenotypes of overexpression lines of CTR1 variants is not due to their overexpression.

4. Several control lines (*ctr1-1*, *ctr1-2*, *ebf2-3*, EIN3 OX, and complementation lines of *ctr1-2*) showed that correlation between enhanced salt and drought stress resilience to increased ethylene response (**Supplementary Fig. 19 and 20**).

Together, these results support the notion that nuclear-localized CTR1 confers stress resilience to Arabidopsis, and the increased abiotic stress resilience of *35S:ΔNT-CTR1^{ctr1-1}* is not an artifact. Furthermore, the analysis of additional transgenic lines included in the revision further supports the positive role of ethylene in drought stress tolerance in Arabidopsis. We included related information in the text.

Reviewer #2 (Remarks to the Author):

This manuscript describes novel findings that advance our understanding of ethylene signaling and open new avenues for further investigation. CTR1 has long been known to be a negative regulator of ethylene responses with the CTR1 protein associating with the ethylene receptors at the ER membrane, but here the authors show in Arabidopsis that ethylene induces translocation of CTR1 into the nucleus where CTR1 seems to function as a positive regulator of ethylene responses, possibly through an inhibitory interaction with EBFs, which in turn leads to increased levels of the transcription factor EIN3, a known positive regulator of ethylene responses. The authors have convincingly demonstrated that ethylene induces translocation of CTR1 into the nucleus, that this nuclear localization involves control by the physical interaction of the CTR1 N-terminal domain with the ethylene receptors. The nuclear translocation is independent of CTR1 kinase activity. Using kinetic growth analysis, the authors show that nuclear-localized CTR1 delays growth recovery upon the removal of ethylene, which inhibits plant growth. They correlate this delayed recovery to enhanced levels of the transcription factor EIN3, perhaps due to protein-protein interaction between CTR1 and EBF2, however the mechanism for the inhibition of EBF2 by CTR1 is not addressed in this study. The authors also show an interesting biological consequence of overexpressing the C-terminal domain of CTR1 in protecting Arabidopsis from abiotic stress. A variety of methods were used in this study, and the methodology is generally sound. In a few cases the interpretation and conclusions could be made clearer or with more caution.

Main suggestions:

The varying data/statements on the role of CTR1 in the nucleus are confusing and should be clarified. For example, the Abstract and Title indicate that CTR1 in the nucleus is a positive regulator of ethylene responses. However, the authors show that ethylene response does not require the nuclear localization of CTR1, given that the *ctr1-8* protein is not detected in the nucleus yet the *ctr1-8* mutant has a constitutive ethylene response. As the authors state, "This

finding indicated that CTR1 nuclear movement does not control the primary ethylene response but instead may influence ethylene response kinetics by fine-tuning nuclear ethylene signaling”(page 8). This sentence on page 8, and the lack of *ctr1-8* in the nucleus, seem at odds with the paper’s conclusion that “CTR1 enhances nuclear ethylene responses by stabilizing EIN3 (in the Discussion, page 13)”. Moreover, to explain why the high, CTR1-induced levels of EIN3 do not confer the known phenotype of EIN3 overexpression, the authors speculate that the overexpressed CTR1 acts to suppress ethylene responses in the cytoplasm (page 11). This again feels in conflict with the main conclusion provided in the title of the paper. I suggest that the nuances of CTR1 action be clarified in the Discussion section.

Response 15: We are sorry about the ambiguity of some of the content that may cause confusion. We concluded that the nuclear localization of CTR1 does not require ethylene response based on the ACC-induced nuclear localization of CTR1 in ethylene-insensitive mutants (*ein2-5* and *ein3eil1*) not by the inability of nuclear translocation of *ctr1-8* protein. The constitutive ethylene response of the *ctr1-8* mutant is mostly due to the unstable binding of *ctr1-8* proteins to the ER, resulting in the reduced suppression of EIN2, not because of its inability to translocate into the nucleus. Our data suggest that seedlings can show normal ethylene response without nuclear-localized CTR1 as long as CTR1 localizes to the ER, but nuclear-localized CTR1 can influence the recovery kinetics of seedlings after the removal of stress or ethylene by increasing the stability of EIN3 proteins in the nucleus.

The wild-type-like phenotypes of the *35S:ΔNT-CTR1^{ctr1-1}* line (#14) were unexpected to us as well, but the results were reproducible. We addressed this comment by providing additional results from the analysis of additional independent overexpression lines of the C-terminal kinase domain of CTR1(*ΔNT-CTR1^{ctr1-1}* #28) and a line expressing full-length CTR1, two lines expressing *ctr1-8* or *ctr1-8-SV40* from its native promoter in the *ctr1-8* mutant background, and several controls. The detailed response to this comment can be found in **Response 14** above.

We modified some text and added more discussion and information in the discussion to minimize the confusion on the nuclear role of CTR1 and ethylene response.

The model in Figure 7 shows a “Secondary Ethylene Response”, but I don’t think this wording was used anywhere in the paper other than in this figure. Please explain what is meant by a secondary response.

Response 16: Thank you for the comments. We used the term “Secondary Ethylene Response” in the model to indicate that the action of CTR1 in enhancing ethylene response in the nucleus via the stabilization of EIN3 is not a primary ethylene response that requires EIN2. Furthermore, the kinetics analysis of CTR1 and EIN2 in our study and others showed that CTR1 appears to move to the nucleus slower than EIN2. We include related discussion in the text (lines 367-375).

The Figure 7 legend states, “After the nuclear translocation of EIN2-CEND, CTR1 is released from the receptors and relocates to the nucleus via an unknown mechanism.” I don’t think

anything can be stated about the order and timing of the translocation of CTR1. In other words, the statement “after the nuclear translocation of EIN2-CEND” seems unrealistic even for a model. What is the basis for proposing this sequential order in the model? I presume there is almost always some basal ethylene, so the ethylene signaling system is not something that is fully turned off in the absence of ethylene and then turned on in a series of steps where CTR1 only goes to the nucleus after EIN2-CEND goes into the nucleus.

Response 17: We proposed the sequential order of CTR1 and EIN2 nuclear movement based on our kinetic analysis and previous work from Ecker’s laboratory. We agree that there would be no complete on and off of signaling pathways in the cells. Therefore, multiple hypotheses on the kinetics of EIN2 and CTR1 nuclear movement are possible. CTR1 may travel to the nucleus together with EIN2 after EIN2 is released from the ER. Alternatively, given the results that the nuclear movement of CTR1 is EIN2/EIN3-independent, upon ethylene perception by the receptors, CTR1 may translocate into the nucleus before EIN2 is released from the ER. Time-course imaging analysis, however, showed that CTR1 appears to be in the nucleus later than EIN2 after ACC treatment. In this submission, we minimized the emphasis on the slower CTR1 movement kinetics than EIN2 and added more discussion on the possible nuclear kinetics of CTR1 and EIN2 mentioned above (lines 367-375).

I think there should be a negative control for the Co-IP of Myc-CTR1 by GFP-EBF2, using some other Myc-tagged protein that is not expected to be pulled down by GFP-EBF2, to strengthen this interaction finding. Alternatively, there should be such a negative control for EBF1 and EBF2 in the Y2H and BiFC.

Response 18: We provided negative control data for EBF1 and EBF2 in BiFC (**Supplementary Fig. 15c-d**).

Why didn’t the authors also test or show co-IP of CTR1 with EBF1?

Response 19: Previous genetic studies have shown that EBF1 and EBF2 share redundant functions in ethylene signaling by destabilizing EIN3 proteins. The single mutant of *ebf1* or *ebf2* shows enhanced ethylene response by stabilizing EIN3, whereas the *ebf1-ebf2* double mutant shows constitutive ethylene response. However, previous work from Vierstra’s laboratory showed that *ebf1* and *ebf2* mutants showed different ethylene response kinetics, especially during the recovery phase after the removal of ethylene. The *ebf1* mutant shows comparable growth recovery to the WT, whereas the *ebf2* mutant displays significantly delayed growth recovery after ethylene removal (1). This result suggests a close functional link between EBF2 and the nuclear role of CTR1, which stabilizes EIN3 proteins by suppressing EBF2. With this rationale, we mainly focused on EBF2.

Reference

- (1) B. M. Binder *et al.*, The Arabidopsis EIN3 binding F-Box proteins EBF1 and EBF2 have distinct but overlapping roles in ethylene signaling. *Plant Cell* **19**, 509-523 (2007).

It would be helpful to know whether the EIN3 constructs are actually expressed in the Y2H and BiFC experiments (Figs 5a, b), since the authors use this data to rule out a direct interaction between CTR1 and EIN3.

Response 20: We provided the suggested immunoblotting results of Y2H and BiFC in **Supplementary Fig. 14b** and **15b**.

In the Discussion, the authors conclude, “We further discovered that the nuclear movement of CTR1 is tightly associated with a plant’s response to abiotic stress” This sentence should be revised because the authors did not demonstrate that resilience to abiotic stress is a normal role of native CTR1 nor that the nuclear movement itself is the cause of the resilience. For the stress experiments, the authors used only an overexpressed C-terminal domain of CTR1, and so they should adjust their conclusion to be more like their statement in the Abstract. (In addition, given that this effect was only shown in Arabidopsis, perhaps the conclusion should not be implied to be true for all plants.)

Response 21: In the original submission, we only examined the stress response of seedlings expressing the C-terminal kinase domain of CTR1 (*35S:GFP-ΔNT-CTR1^{ctr1-1}*) because we included the stress results as peripheral to the main story. In this revision, we included the results of the salt/drought response of seedlings expressing full-length CTR1 (*35S:GFP-CTR1^{ctr1-1}*) and also seedlings expressing CTR1 from its native promoter (*CTR1p:GFP-CTR1^{ctr1-8}/ctr1-8* and *CTR1p:GFP-CTR1^{ctr1-8}-SV40/ctr1-8*) and additional control lines.

The results showed that not only truncated CTR1, but also full-length CTR1 proteins, can induce stress resilience (**Fig. 6a, 6c, 6g, 6i, 6j, and Supplementary Fig. 18a, 19, and 20**).

Furthermore, the salt and drought responses of *CTR1p:GFP-CTR1^{ctr1-8}-SV40/ctr1-8* native promoter-driven lines demonstrated that the forced nuclear-localization of CTR1-8^{ctr1-8} protein by SV40 NLS confers a higher salt and drought tolerance phenotype without suppressing the constitutive ethylene response of the *ctr1-8* mutant, which was observed in *CTR1p:GFP-gCTR1^{ctr1-8}/ctr1-8* (**Fig. 6f** and **6l**). We observed that the seedling size of *CTR1p:GFP-gCTR1^{ctr1-8}/ctr1-8* was significantly bigger than *ctr1-8* but slightly smaller than WT (**Fig. 6f** and **6l**). This is likely that CTR1^{ctr1-8} protein expressed from the transgene does not translocate into the nucleus, thereby inhibiting ethylene response in the cytoplasm, whereas nuclear-targeted CTR1^{ctr1-8}-SV40 proteins stabilize EIN3 proteins, thus enhancing tolerance to abiotic stress. Together with ACC/ethylene-induced nuclear localization of full-length CTR1 expressed from its native promoter, these results support that ethylene-triggered nuclear localization of CTR1 and its associated stress resilience is a normal function of CTR1 in Arabidopsis plants. We included these new results in the text, figures, and discussion.

We also removed and substituted the sentences implying the extrapolation of the nuclear role of Arabidopsis CTR1 in other plants.

For the abiotic stress experiments, I think the authors should have included the *ctr1-1* mutant and a line that overexpresses EIN3 as controls to see whether overexpressing the C-terminal domain of CTR1 provides even greater protection. Could the overexpression construct have caused gene silencing of CTR1 in those lines that were analyzed?

Response 22: We examined the salt and drought responses of the suggested lines and other control lines (**Supplementary Fig. 19** and **20**). In general, seedlings expressing the C-terminal domain of CTR1 (*35S:GFP-ΔNT-CTR1^{ctr1-1}*) or full-length CTR1 (*35S:GFP-CTR1^{ctr1-1}*) showed a higher survival rate than the *ctr1-1*, *ctr1-2*, and *ebf2-3* on media containing 175mM NaCl. The

complementation line of *ctr1-2* by a *CTR1* genomic fragment showed a comparable survival rate to the WT. Furthermore, *35S:GFP-CTR1^{ctr1-8}* line showed a comparable survival rate to that of WT, similar to their response to salt in soil-grown conditions (**Fig.6e** and **Supplementary Fig.19**). However, the survival rate of both *35S:GFP-ΔNT-CTR1^{ctr1-1}* and *35S:GFP-CTR1^{ctr1-1}* seedlings is lower than that of EIN3 overexpression lines, which is likely due to that EIN3ox lines produce significantly higher levels of EIN3 proteins (**Supplementary Fig.18c and 19**).

Similarly, we examined the germination rate of various CTR1 transgenic and control lines on media containing different concentrations of mannitol (**Supplementary Fig. 20**). The results are consistent with the results from the salt response experiments in that constitutive ethylene response mutants showed a higher germination rate than WT in growth media with higher mannitol concentrations. Both *35S:GFP-ΔNT-CTR1^{ctr1-1}* and *35S:GFP-CTR1^{ctr1-1}* showed similar or slightly higher germination rates than *ctr1-1*, *ctr1-2*, *ebf2-3*, and EIN3ox. Given that we used the same EIN3ox lines used for survival on salt plate, this result may suggest that the threshold levels of EIN3 protein required for stress tolerance to salt and drought may differ in Arabidopsis seedlings. The complementation line of *ctr1-2* by a *CTR1* genomic fragment showed a similar germination rate to the WT, indicating that the genomic fragment of CTR1 complemented the *ctr1-2*. Consistent with the drought phenotypes on soil conditions, *35S:GFP-CTR1^{ctr1-8}* exhibited similar levels of germination throughout different concentrations of mannitol to the WT. We included these results and discussion in the text.

We did not find silencing of CTR1 in the lines we used over several generations.

The abiotic resilience could be tested with the *ctr1-8* version with and without the SV40 NLS in order to correlate the stress resilience with nuclear localization of CTR1-8.

Response 23: We used two native lines (*CTR1p:GFP-CTR1^{ctr1-8}/ctr1-8* and *CTR1p:GFP-CTR1^{ctr1-8}-SV40/ctr1-8*) and *ctr1-8* to show a correlation of stress resilience with nuclear-localized *ctr1-8* proteins (**Fig. 6f** and **6l**). Unlike native promoter-driven expression of SV40-fused CTR1-8, expression of CTR1-8 from its native promoter suppressed the constitutive ethylene response of the *ctr1-8* mutant, thus resulting in a WT-like salt and drought response. This is likely attributable to that CTR1^{ctr1-8} protein expressed from the transgene does not translocate into the nucleus, thereby inhibiting ethylene response in the cytoplasm and rescuing the ethylene response phenotype of the *ctr1-8*. Consistent with our hypothesis, the size of *CTR1p:GFP-CTR1^{ctr1-8}/ctr1-8* seedlings was significantly bigger than *ctr1-8* but slightly smaller than WT. By contrast, nuclear-targeted CTR1^{ctr1-8}-SV40 proteins stimulate ethylene response by increasing EIN3 levels as shown in **Fig. 5f**, thus enhancing salt and drought tolerance. These results showed that nuclear localization of CTR1 is linked to salt and drought stress resilience.

Minor comments:

Please describe what kind of mutations are used for the following experiment for those who are unfamiliar with the mutant alleles: "Disruption of either EIN2 or EIN3/EIL1 did not prevent the nuclear accumulation of CTR1 (Fig. 1g)."

Response 24: We indicate the related reference of EIN2 and EIN3/EIL1 mutations in the materials and methods section (line 456).

Since there is no quantification for the data in Figure 1h, the authors should not refer to “a large fraction” in this sentence: “a large fraction of GFP-CTR1 constitutively localized to the nucleus in etiolated seedlings, regardless of ACC treatment (Fig. 1h).”

Response 25: We added a quantification graph.

Cite Supplementary Figure 7 before the part of the sentence that says “both of which...” in the following sentence: “Similar kinetics of ethylene response and growth recovery was also observed in seedlings expressing active or inactive wild-type full-length CTR1 (35S:GFP-CTR1 and 35S:GFP-CTR1ctr1-1), both of which showed some level of constitutive CTR1 nuclear localization (Fig. 1h, 3c, and Supplementary Figure 7).”

Response 26: We moved the Supplementary 7 in the original submission to Fig. 4b in the revised version.

For the list of ACC responses on page 5, reference 24 does not show ACC in pollen tube growth.

Response 27: We removed a word “growth”.

Minor edits:

In the Figure 5 legend, do not capitalize “B”, and add the “S” to “35:GFP-...”for g.

In the legend for 5h, replace “showed enhanced ACC response than” with “showed an enhanced ACC response compared to”

Capitalize the first letter of “agrobacteria”.

Response 28: Thank you for finding the mistakes. We fixed accordingly.

Reviewer #3 (Remarks to the Author):

My comments to this work can be found from the attached file.

The article by Park and co-authors reports a very interesting, eye-catching finding on the cytoplasmic-nuclear transport of the Raf-like CTR1 protein, an essential negative regulator of the ethylene signaling pathway, and provides another aspect of ethylene signaling regulation via the F-box protein EBF1/EBF2. The finding of CTR1 nuclear localization is convincing and exciting. My only concerns about the inference made in this work are 1) effects of the overexpressed proteins on ethylene signaling appear to be marginal and conditional, and it is undetermined whether the reported effects are also conferred by the native function of CTR1, 2) data acquisition and analyses for the laser scanning confocal microscopy (LSCM) primarily involved “independent samples” but not “paired samples” to compare effects of a treatment or genetic backgrounds. Large sample-to-sample variations are prone to false negatives or false positives in cell biology studies, 3) subcellular localizations and effects of the several different

GFP-CTR1 variants involve data from independent transgenic lines (or events) in different genetic backgrounds, and it is unlikely to determine whether the different degrees of nuclear localization or effects are results of the protein levels or the treatment/genetic backgrounds, and 4) wording needs to be more precise to avoid misleading.

Specific comments:

1. According to the image data presented, as long as GFP-CTR1 exceeds a level, constitutive nuclear localization will be observed at various degrees. It is the level, instead of the transgene promoters (CTR1p or 35S), that determines the percentage of nuclear localization. The authors, however, complicated their data presentations at the first place and gave the authors to an impression that GFP-CTR1 is constitutively nuclear localized by the 35S:GFP-CTR1 but not the CTR1p:GFP-CTR1 transgene. As a fact, GFP-CTR1 by the CTR1p:GFP-CTR1 transgene also exhibited a level of constitutive nuclear localization. It is misleading to describe the GFP-CTR1 movement to be "constitutive". The wording "constitutive" needs to be revised for other GFPCTR1 variants in this work.

Response 29: We agree that when WT CTR1 is expressed in the *ctr1-2* background, its nuclear localization appears to be influenced by the levels of WT CTR1 in plants. We showed that ACC treatment substantially increases CTR1 stability (**Fig. 1I**), which is consistent with a previous study showing that ethylene treatment increases CTR1 levels (1). This ACC-induced increase in CTR1 protein abundance is somewhat similar to the effect of CTR1 overexpression lines, in which CTR1 localizes to the nucleus in the presence of silver nitrate. Thus, it is conceivable to think that if ACC-induced increase in protein levels of WT CTR1 is beyond the threshold that the ethylene receptors can hold them at the ER, the excess CTR1 would translocate into the nucleus even if its expression is regulated by its native promoter. However, the treatment of silver nitrate nearly eliminated nuclear localization of CTR1 (**Fig. 1j, Supplementary Fig. 2, and 4**). But, in the case of 35S CTR1 lines, CTR1 is still translocated to the nucleus due to its overexpression, which is not influenced by silver treatment. Overall, we find that Ethylene-induced stabilization of CTR1 could be part of the mechanism by which ethylene stimulates the movement of CTR1 to the nucleus, in addition to the displacement of CTR1 from the ER via the binding to receptors. Therefore, the reviewer's comment is somewhat consistent with our results on WT CTR1. We realized that we did not provide sufficient discussion regarding the role of ACC on CTR1 protein abundance and its nuclear translocation, and its link to the constitutive CTR1 nuclear localization in 35S lines. We added a related discussion in this submission (lines 108-112).

While the nuclear localization of WT CTR1 in the *ctr1-2* background appears to be influenced by its expression levels because ACC treatment increases the protein abundance of CTR1, we are afraid that we do not agree that the nuclear-localization of WT CTR1 in other genetic backgrounds and CTR1 variants is determined by the levels of CTR1 proteins. For example, the reviewer commented that "When the protein is in excess amount, its ER localization is expected. On the other hand, when GFP-CTR1-8 in transgenic lines is expressed at a low level, it is less likely to observed a nuclear signal, which does not mean the absence of nuclear localization" in comments below. To demonstrate that the levels of CTR1^{ctr1-8} do not determine its ER or nucleus localization, we provided immunoblotting analysis of native and 35S lines that

express WT CTR1 or CTR1^{ctr1-8} proteins used in this study (**Supplementary Fig. 8**), and their Z-stack images with and without ACC and silver nitrate treatment (**Supplementary Fig. 2**). The immunoblotting analysis showed that similar levels of WT-CTR1 and CTR1^{ctr1-8} proteins were expressed in both 35S overexpression and native lines, yet CTR1^{ctr1-8} exhibits different localization patterns from the WT-CTR1 protein in response to ACC and silver nitrate treatment. The fractionation analysis of WT (Col-0) and *ctr1-8* mutants using a CTR1 antibody further showed that endogenous CTR1^{ctr1-8} protein is not present in the nucleus regardless of ACC or silver treatment (**Fig. 2i** and **Supplementary Fig. 4**). Likewise, WT CTR1 expressed from its native promoter in the *etr2-3ers2-3ein4-4* mutant background constitutively localized to the nucleus even in the presence of silver nitrate despite that the expressed CTR1 level is lower than that in the WT and *ein3eil1* backgrounds (**Fig. 2J, 2k** and **Supplementary Fig. 8**). Together, these results demonstrate that the nuclear localization of CTR1 and CTR1 variants is not associated with their protein levels in plants.

We agree that the use of “constitutive” may lead to a misunderstanding of the localization of CTR1 in plants as we did not provide adequate discussion on ACC-induced stabilization CTR1 and its potential consequences on CTR1 nuclear translocation due to enhanced protein abundance. We added this to the text and also modified the sentences that explain the nuclear localization of CTR1 in the 35S lines (lines 108-112).

References:

1. Localization of the Raf-like Kinase CTR1 to the Endoplasmic Reticulum of *Arabidopsis* through Participation in Ethylene Receptor Signaling Complexes (2003). Gao et al., JBC.
2. Data acquisition and analyses from the laser scanning microscopy (LSCM) will be critical to the interpretation of GFP-CTR1 localizations in the study and need to be very carefully and rightly conducted.
 - Cells are in a three-dimensional shape, and an image acquired from a focal plane of the laser confocal microscopy (LSCM) only provides a slice of two-dimensional information. The constitutive nuclear fluorescence for lines with high GFP-CTR1 levels will be much evident than those with low levels, because strong nuclear fluorescence can be observed over a wide range of focal planes and weak fluorescence only observed within a narrower focal plane range (See the illustration below). This easily explains the observation of constitutive GFP-CTR1 nuclear localization of those 35S:GFP-CTR1 lines and partial nuclear localization of those CTR1p:GFP-CTR1 lines. This also explains the uneven fluorescence (i.e., cell-to-cell variations) among cells of an image sample (for example, Fig 1f and 1g, where the fluorescence is much stronger in cells on both sides and weak in the center of the AgNO₃-treated hypocotyl, and the opposite for the ACC-treated). There will also be transgenic lines, regardless of the promoters, exhibiting various levels of the GFP-CTR1 proteins. In other words, one will observe constitutive nuclear localization for CTR1p:GFP-CTR1 lines and partial nuclear localization for 35S:GFP-CTR1 lines. There will be *ctr1-2* lines exhibiting the WT phenotype while the GFP-CTR1 fluorescence

barely detectable. It may be misleading to report constitutive GFP-CTR1 nuclear localizations only for 35S:GFP-CTR1 lines.

- Following comment 1 (also the illustration), the cell-to-cell variation, generated from different focal plane sampling, will distort the observed nuclear localization percentage reported in Fig 1 due to false negatives. Please describe how was the % of nuclear localized CTR1 measured to ensure that the analyses were properly conducted. When the data are presented in percentage, it means that that are nuclei truly determined without the fluorescence (did the authors conduct this measurement?). The authors will need to report the number of individual nuclei counted and the number of nuclei exhibiting the fluorescence (make sure that those without fluorescence is determined over the z-axis for). When the number of observed nuclei is less than 100, fraction but not percentage should be used; otherwise, the percentage is an estimate but not an actual percentage and cannot be subject to statistical analyses. There will be no error bars for the percentage/fraction; once the percentage or fractions are measured, it will be goodness of fit instead of t-test to determine statistical significance for data such as Fig 2c and 2i.
- The GFP-CTR1 nuclear localization affected by a treatment presented in most of the figures was not determined from the same samples after a treatment (i.e., dependent samples or paired samples that reveal the changes before-and-after a treatment), and the LSCM images comparing “treated” and “non-treated” (independent samples) cannot truly reveal effects of a treatment (or the mutations) because of complexity associated with different focal plane sampling and sample-to-sample variations as explained in comments 1 and 2. Without in-depth analyses, results from independent sampling in cell biology studies are prone to false positives/negatives.
- When “paired samples” are involved for LSCM image comparisons, caution should be taken to avoid focal plane shift that also affects the observation and inference (note, as an example, there is focal plane shift for Fig 1i/ 1j and Supplementary Fig 4; the presented images were not taken from the same focal planes, as inferred from the different outlines of adjacent cells). The focal plane shift makes it unlikely to determine whether the observed nuclear signal was a result of cytoplasmic-nuclear trafficking or simply stronger fluorescence signal captured from different focal planes. For instance, the total fluorescence at 60 min was evidently much stronger than that at time 0 to 30 min for Fig. 1j. It is also recommended to present full images of a LSCM image (can be presented in expanded information) over the z-axis for some essential images, instead of a crop for a cell, so that more information can be available to assess the nuclear localization status from different cells. Besides, the cropped images for a single cell cannot be “representative”, and one cannot make inference from single samples in the absence of sufficient information from other cells.
- Involving independent transgenic lines (events), there need negative controls (nontreated) for the GFP-CTR1 localizations/percentage to contrast effects of the AgNO₃ and ACC treatment in the indicated mutant backgrounds in Fig 1 (including ctr1-2, etr1-1, ein2- 5, and ein3 eil1).
- Following the aforementioned comments and explanations on the problems associated with “independent sampling”, I strongly recommend the authors to carefully conduct “paired sampling” and avoid focal plane shift to provide better quantification and more reliable image

evidence to characterize the cytoplasmic-nuclear transport event. This comment is also important because different transgenic lines in those genetic backgrounds are involved in this study and there are always cell-to-cell variations for a same line. Analyzing paired samples will minimize the uncertainty resulted from different protein levels. To avoid focal plane shift, which always happens for prolonged observations, one will need to re-adjust the focal plane manually to the time 0 plane by referencing the cell outlines. In this study, ACC treatment was conducted for 2 hrs. Given that a short ACC treatment is sufficient to induction of the ethylene response, the GFP-CTR1 movement in response to ethylene for paired samples can be easily conducted (as reported in Fig 1j and 1l).

Response 30: Thank you for the comments. To respond to the concerns related to focal plane change during imaging of seedlings, we need to mention how we determined to use +ACC and +silver nitrate conditions for the experiments and single focal plane images. We found that etiolated seedlings expressing CTR1 in its native promoter without any treatment occasionally display low levels of nuclear-localized CTR1 in small portions of cells (**Supplementary Fig. 2**). This wasn't unexpected given Petri-dish growth conditions (a potential stress condition) and basal levels of ethylene in dark-grown seedlings. However, this low level of nuclear-localized CTR1 is nearly undetectable in seedlings grown on silver nitrate, as shown in **Fig. 1j**, **Supplementary Fig. 2**, and **4**, and ACC treatment strongly enhanced CTR1 nuclear localization, supporting the nuclear movement of CTR1 is ethylene-dependent. The key idea of the study is to demonstrate that ethylene triggers the nuclear translocation of CTR1 and determine the underlying mechanism using various CTR1 variants. To achieve this goal, the most critical factor is to create a condition that blocks ethylene signaling in order to determine if different CTR1 variants and WT CTR1 in different genetic backgrounds behave differently from the WT CTR1. Given this, no treated-seedling condition is not an essential part of the experiments as ACC clearly stimulates nuclear transport of most CTR1 variants but their nuclear translocation is distinctively regulated by silver nitrate treatment. Furthermore, in the presence and absence of ACC and silver nitrate, the localization of WT CTR1 and its variants in the cytosol and nucleus is very clear and quantitatively different, which does not necessitate Z-stack imaging. With this rationale, we used ACC-treated and silver-treated conditions and single focal plane imaging to examine the ethylene-induced nuclear localization of different CTR1 variants.

We measured the percentage of cytosolic vs. nuclear-localized CTR1 variants by using single focal plane images of 3–4 (some has close to 10) independent seedlings per genotype. As the localization of various CTR1 variants in response to ACC and silver nitrate is apparent and different, we scanned through seedlings expressing CTR1 or CTR1 variants and obtained the best representative focal plane images. We provided the requested raw excel data in the source files.

For the suggested use of "paired sampling", while we acknowledge the importance of paired sampling in certain cell biology contexts in which the changes in the localization of a given protein are subtle, the changes in the localization of CTR1, particularly its movement into the nucleus, are qualitatively clear in our experimental settings. In our study, we used multiple

approaches, including transgenic lines expressing mutant CTR1 from its native promoter in different backgrounds, overexpression lines, and biochemistry, to demonstrate that the nuclear localization of CTR1 is not the result of changes in its protein levels or of focal plane changes. For a similar reason, many cell biology studies in plants use similar approaches as ours if the changes are substantial (1-8). In addition, to repeat all the experiments in the manuscript using such an approach would involve repeating the bulk of the experiments starting nearly from scratch, which will take an inordinate amount of time (~ 2 years). Finally, the suggested "paired imaging" is not technically feasible (not impossible to do it, but it takes enormous time to get a decent image with minimum focal plane changes) in our experimental setting in which seedlings are in an aqueous solution; it would be extraordinarily difficult to locate the exact same focal plane after extended treatment times. Further, seedlings continue to grow during the 2 h of incubation time, which will also affect adjusting a focal plane. Overall, while we appreciate the reviewer's point, we do not feel that paired imaging is required to support the conclusions we draw in the manuscript. Second, for the comment that it is simply the level of CTR1 protein that impacts its localization to the nucleus is not correct. We provided additional data that supports that is not the case for our studies (**Response 29**). We also provided Z-stack imaging of key transgenic lines with different nuclear localization in the presence and absence of ACC/silver nitrate to confirm that the movement of CTR1 into the nucleus is not the result of changes in the focal plane (**Supplementary Fig. 2**)

To investigate if the presented data resulted from possible focal plane changes or prolonged laser exposure, we repeated the experiments several times and included the Z-axis in data processing to minimize the effects of focal plane change. The newly acquired data (merged 10 successive Z-stack images at each time point) using *CTR1p:GFP-CTR1/ctr1-2* showed that ACC appeared not to influence the intensity of CTR1 fluorescence in the cytoplasm but increased the fluorescence of CTR1 in the nucleus over time compared to that in the seedlings without ACC treatment at time 0 (**Fig. 1k** and **Supplementary Fig. 6**). Consistent with the results in the original submission, CTR1 amassed in the nucleus approximately 30 min after ACC treatment. Normally, we image seedlings by placing them on a microscope slide followed by ACC solution mount and obtain images at different time points. During this process, seedlings in ACC solution gradually change their position over time. Thus, we speculate that the increased GFP-CTR1 signals in Fig.1i in the original submission may likely be due to focal plane changes of the cells while locating the same cells at the previous time points. We are sorry about the inadequate quality of the data in the original submission. We replaced it with the newly generated data obtained with a more careful imaging process and Z-axis inclusion (**Fig. 1k**).

For Fig.1j, we performed this experiment in a similar way to that of Fig.1i in the original submission. Thus, we repeated the experiment as described above. The overall signals in both the nucleus and ER were gradually reduced until 40 min after ethylene removal, followed by a steep reduction in GFP signals at 60 min after ethylene removal (**Fig. 1I**). To further investigate the protein abundance of CTR1 after ethylene removal, we examined the protein abundance of GFP-CTR1 after ethylene removal using immunoblotting. Seedlings were pretreated with ethylene for 2 h and harvested at different time intervals after ethylene removal. As shown

in **Supplementary Fig. 7**, the steady-state level of GFP-CTR1 gradually reduces after the removal of ethylene.

For statistical comments on percentage-based data presentation, we have consulted a mathematical biologist in our department because we are aware of that there is an ongoing debate on using error bars on percentage-based data. Below is the consultation we received from him regarding your comments:

(i) Percentage is just a linear transformation of proportion. Whether it is presented as p or $p * 100$ or even p multiplied by any constant c that you can imagine (other than 0 or infinity I suppose), this has no effect on comparisons among treatments. It just moves the y axis up and down maintaining the exact same relative difference between treatments.

(ii) Any mean in any paper is always an estimate of the true mean and so the “actual percentage” is never known in any study that hopes to generalize beyond their sample. There is nothing special about counting 100 things to be able to do a linear transformation of $p * 100$. There is also nothing special about $n = 100$ to be able to apply any form of inferential statistics, whether goodness of fit or a t-test. You will have less power to detect small effect sizes, with smaller sample sizes, but if you are interested in big effect sizes then you can still detect them with as little as $n = 3$.

(iii) There is no reason why one can't have error bars on proportions, though they have a special equation, so you may want to recalculate them to be sure they are right. Additionally, if you want to express these as percentages, a linear transformation on p will also change the SD. With a proportion, p , that is transformed by multiplying by some constant c , and for a sample size of n , the standard deviation (SD) of $p * c$ is given by:

$$SD = \sqrt{c^2 \frac{p(1-p)}{n}}$$

So, if you want to express p as $100p$, then you set $c = 100$.

Based on this consultation, we have changed the SD of all % graphs based on the equation. We also used SE instead of SD as you suggested in latter comments.

References:

1. Nucleocytoplasmic trafficking and turnover mechanisms of BRASSINAZOLE RESISTANT1 in *Arabidopsis thaliana* (2021). Wang et al, PNAS.
2. Degradation of the Endoplasmic Reticulum by Autophagy during Endoplasmic Reticulum Stress in *Arabidopsis* (2012). Liu et al., Plant Cell.
3. Proper PIN1 Distribution Is Needed for Root Negative Phototropism in *Arabidopsis* (2014). Zhang et al., PLOS ONE.
4. A B-ARR-mediated cytokinin transcriptional network directs hormone cross-regulation and shoot development (2018). Xie et al., Nature Communication.

5. EIN2-directed translational regulation of ethylene signaling in Arabidopsis (2015). Li et al., Cell.
6. Ethylene-Induced Stabilization of ETHYLENE INSENSITIVE3 and EIN3-LIKE1 Is Mediated by Proteasomal Degradation of EIN3 Binding F-Box 1 and 2 That Requires EIN2 in *Arabidopsis* (2010). An et al., Plant Cell.
7. Activation of ethylene signaling is mediated by nuclear translocation of the cleaved EIN2 carboxyl terminus (2012). Wen et al., Cell Research.
8. Processing and Subcellular Trafficking of ER-Tethered EIN2 Control Response to Ethylene Gas (2012). Qiao et al., Science

3. The nuclear GFP-CTR1 rises from about 10% (AgNO₃) to 60% (ACC) (a 6-fold increase; Fig 1c), and the nuclear GFP-CTR1 is undetected for the silver treated (Fig 1i). Can a prolonged exposure detect GFP-CTR1 for the silver treated? A negative control (non-treated) will be needed as well to demonstrate inhibition of CTR1 nuclear transport by silver. This comment applies to both WT, *ein2* and *ein3 eil1*. In the western blot (Fig 1i), please explain why and which Hsc70 isoform was used as the internal reference and why the ER marker BiP2 not used. Hsc70 could exist in the cytoplasm and nucleus, and it is the ER marker that may better determine if there is any ER contamination in the nuclear fraction.

Response 31: The imaging of CTR1 localization in dark-grown seedlings is very tricky. It requires extra caution when handling seedlings and minimization of light exposure to reduce the stimulation of ethylene signaling that results in the nuclear localization of CTR1. We found that seedlings that were grown in media containing silver nitrate generally did not show nuclear-localized CTR1, but occasionally, a small fraction of cells may show the basal levels of nuclear-localized CTR1 during the handling of samples or prolonged laser exposure, which may account for 10% of nuclear-localized CTR1 in seedlings with silver nitrate treatment. Given this, no detection of nuclear-localized CTR1 in silver-treated seedlings is likely that seedlings were immediately processed to isolate protein extracts without going through extra handling that might cause stress on seedlings.

As suggested, we included Z-stack images and fractionation of seedlings without any treatment (**Fig. 1j** and **Supplementary Fig. 2**). Seedlings without any treatment sometimes showed low levels of CTR1 nuclear localization in some portions of cells, as mentioned above. However, the low level of nuclear-localized CTR1 is almost not detected in seedlings grown on media containing silver nitrate and the treatment of ACC significantly accentuates the levels of nuclear-localized CTR1. This suggests that ethylene controls CTR1 nuclear movement and silver nitrate blocks nuclear transportation. We also used BiP as suggested to confirm that the nuclear fraction is not contaminated with ER-localized CTR1.

4. In Materials and Methods, please describe whether the purified proteins for the in vitro assays were from soluble or re-solubilized inclusion bodies. The description for His6-EIN2WT is misleading because this work does not involve the full-length EIN2.

Response 32: All of the proteins used in the kinase assays were purified from soluble fractions.

We included this information in the section and changed “His6-EIN2WT” to “His6-EIN2-CEND”.

5. The conclusion that GFP-CTR1 constitutively localizes to the nucleus in *etr2 ein4 ers2* needs to be carefully made because the transgenes in WT and the mutant are expressed at a different locus (i.e., independent transgenic event), and levels of the GFP-CTR1 may differ and affect the inference. This argument agrees with the fact that GFP-CTR1 constitutively localizes to the nucleus when the transgene is driven by the 35S promoter in WT. Moreover, the constitutive nuclear localization of GFP-CTR1 in *etr2 ein4 ers2* appears to agree with the absence of ER-associated CTR1 reported by Gao and co-authors (Gao et al., 2003). On the other hand, the soluble fraction by Gao et al. involves 8,000×g and 10,000× centrifugations, which will greatly remove the nuclei, indicating that the detected CTR1 in the soluble fraction is from cytosol but not from the nucleus. This comment also applies to the CTR1-8 protein. *ctr1-8* is a relatively weak allele, producing a mild mutant phenotype throughout development, and CTR1-8 in part associates with the membrane. When the protein is in excess amount, its ER localization is expected. On the other hand, when GFP-CTR1-8 in transgenic lines is expressed at a low level, it is less likely to be observed a nuclear signal, which does not mean the absence of nuclear localization. The different degrees of nuclear GFP-CTR1 localization may in part arise from different protein levels of those independent transgenic lines. To confirm ethylene-induced CTR1 localization to the nucleus, it is critical to provide evidence without involving transgene expression (determining nuclear CTR1 and CTR1-8 by immunoblotting). This will not be a challenging comment because one of the authors has publications involving anti-CTR1 antibodies.

Response 33: We provided the results that demonstrated the unrelatedness of the expression levels of CTR1 proteins and their nuclear localization in above (**Response 29**). However, we acknowledge the reviewer’s point. Thus, we performed immunoblotting analysis using the antibody generated by the Kieber lab (1) for fractionation analysis of WT (Col-0), *ctr1-8*, *ctr1-1*, and *etr2-3ers2-3ein4-4* mutants. Similar to GFP-CTR1 or GFP-CTR1^{ctr1-1} in *ctr1-2* mutant, the nuclear-localization of endogenous CTR1 in WT and *ctr1-1* is dependent on ACC, and silver treatment nearly blocks its nuclear localization (**Fig. 3c** and **Supplementary Fig. 4**). In contrast to the WT endogenous CTR1, the fractionation analysis of the *ctr1-8* mutant showed that endogenous CTR1^{ctr1-8} protein is not present in the nucleus fraction of seedlings regardless of treatment (**Fig. 2i**). Furthermore, endogenous CTR1 proteins in the *etr2ers2ein4* was detected in the nucleus regardless of ACC and silver treatment, corroborating their constitutive nuclear localization (**Fig. 2k**).

References:

1. Localization of the Raf-like Kinase CTR1 to the Endoplasmic Reticulum of *Arabidopsis* through Participation in Ethylene Receptor Signaling Complexes (2003). Gao et al., JBC.

6. There lacks a non-treated control for the percentage of CTR1 nuclear localization to determine effect of the AgNO₃ or ACC treatment. It is important to biochemically determine the

subcellular localizations of the endogenous CTR1, CTR1-8, and CTR1-1 to support the model and the LSCM results proposed in this study (good anti-CTR1 antibodies are available from one of the coauthors). To determine subcellular localizations of endogenous CTR1-8, CTR1-1, or CTR1 (in the triple mutant), proteins from the soluble, membrane, nuclear fractions will need to be determined by immunoblot analyses. Previous studies on CTR1/CTR1-8 were only conducted for the membrane and soluble fractions (Gao et al., 2003), which do not address whether the protein is in the nucleus.

Response 34: For the comments "There lacks a non-treated control for the percentage of CTR1 nuclear localization to determine effect of the AgNO₃ or ACC treatment", we provide a response on why we used +ACC and +Silver nitrate conditions in **Response 29** and also provide Z-stack images of WT CTR1 (**Supplementary Fig. 2**).

We provided the results of fractionation analysis of endogenous CTR1 in WT, *ctr1-1*, *ctr1-8*, and *etr2ers2ein4* backgrounds (**Fig. 2i, 2k, Fig. 3c, and Supplementary Fig. 4**).

7. At present, the inference is interfered by excess amount of the GFP-CTR1 proteins. Biological effects of the overexpressed GFP-CTR1 are marginal and not easily detected, interference of the excess amount protein on native CTR1 function or ethylene signaling needs to be considered.

Response 35: We addressed the comments in the responses above.

8. The seedling growth recovery rate (Fig 4) was determined from transgenic lines involving independent transgenes at different loci. Because of the effects (or difference) are marginal, with large variations (data are means and SEs, $n \geq 6$, meaning that the SD for each sample mean will be approximately $SE \times \sqrt{6}$), and can only be observed conditionally, it is unlikely to determine whether the differences are results of different protein levels or the expressed proteins. Alternatively, excess amount of GFP-CTR1 or GFP-CTR1-1 may disturb the native CTR1 function (this comment applies to data for the ACC dose-response assay in Fig 5g and 5h). Studies on the GFP-CTR1-8-SV40 protein needs a positive control (GFP-CTR1-SV40) to determine whether the SV40 signal may impair function of the CTR1 or CTR1-8 protein. This control is important to addressing the biological significance of constitutive localization of CTR1 in the nucleus. Conversely, investigating whether an NES signal may impair CTR1 function is equally important. The conversion of seedling growth to the recovery rate may somehow distort the growth curves for different genotypes, and the rate could be misleading. For example, the seedling hypocotyl will be longer for WT than for *ctr1* mutants, thus a larger denominator for WT than for *ctr1* mutants; as a result, the actual growth recovery will be under estimated for WT and overestimated for *ctr1* mutants. This also explains for the higher growth rate for *ctr1-8* and the two transgenic lines than for WT (**Supplementary Fig 8b**). The conclusion for the different growth recovery reported in Fig 4 is somehow descriptive or arbitrary, lacking statistical testing. The authors will need to report the support for their inference with statistical evidence. With the conversion of growth recovery to recovery rate, the data are somehow distorted by the different denominators (especially for Fig 4b), and the authors will need to determine a reasonable way to analyze the data. I might not have problems for the interpretations for GFP-CTR1-8- SV40 of

Fig 4b; at least, the difference appears prominent. The description “the GFP-CTR1 transgene ...restored the slower growth recovery of *ctr1-8* to levels comparable to those of the WT” needs to be revised. The transgenic line appears to have a greater growth recovery rate than WT as shown in Fig 4b.

Response 36: We only indicated the minimal number of samples in the manuscript. In some cases, over 40 were used for analysis. We provided a table with the information (**Supplementary Table 1 and 2**) and indicated n for each genotype (**Fig. 4**)

The normalization of the growth rate does not distort the recovery rate. The same times of recovery are observed whether or not the data is normalized. The normalization of the data is to make it easier to visually compare the different seed lines. Additionally, there is no correlation between the growth rate in air and the growth recovery rate after the removal of ethylene. We have added growth rates in the air pre-treatment so that this can be more easily seen. As can be seen from the table (**Supplementary Table 1**), the only seed line with a growth rate significantly different from WT is the *ctr1-8*.

As pointed out, there is no distortion of data from normalizing the data. However, we have determined the recovery time of each seedling to 100% of the pre-treatment growth rates. We have then calculated the average time \pm SEM. We also used ANOVA to determine whether any of these recovery rates were statistically different from WT. This information is provided in **Supplementary Table 2** and we also added a graph in the main figure (**Fig. 4**).

For the comment about “the GFP-CTR1 transgene ...restored the slower growth recovery of *ctr1-8* to levels comparable to those of the WT” even without statistical analysis, the rate of recovery is only modestly faster than the WT, and hence the word “comparable” is appropriate. However, as you also asked for statistics, the modestly faster recovery is not statistically significant. We have therefore left the wording unchanged.

To address the concerns regarding the possibility that the use of SV40-NLS might affect the function of WT CTR1 and CTR1^{*ctr1-8*} proteins, we provided an ethylene response kinetics of seedlings expressing CTR1^{*ctr1-8*} or CTR1^{*ctr1-8*}-SV40 from its native promoter to examine if the addition of SV40 affects the ethylene response of CTR1^{*ctr1-8*}. Since we do not have lines expressing WT CTR1-SV40 NLS on hand, we used these two lines to show that the addition of SV40-NLS did not affect the ethylene response of CTR1^{*ctr1-8*} proteins. The response kinetics analysis showed that CTR1^{*ctr1-8*}-SV40 seedlings displayed comparable ethylene kinetics to that of CTR1^{*ctr1-8*} seedlings, indicating that SV40-NLS does not affect the role of CTR1^{*ctr1-8*} in ethylene response. We added the results in **Supplementary Fig. 13**.

The suggested experiment using NES-fused CTR1 is an excellent idea. However, we do not have the lines, so we did not add them in this revision. We feel that our data that includes multiple transgenic lines (**Response 29**) is sufficient to show that the nuclear role of CTR1 in stabilization of EIN3 and its associated stress resilience.

9. Levels of EIN3 shown in Fig 5d may not support the differences in the seedling hypocotyl lengths (Fig 5g). The levels of Fig 5e may not support the *ctr1-8* seedling hypocotyl lengths (Supplementary Fig 8a and 8b). EIN3 levels of 35S: Δ NT-*ctr1-1* at 0 ACC were similar to that of WT at 1-10 mM ACC, but the difference between the seedling hypocotyl lengths of WT and 35S: Δ NT-*ctr1-1* was not statistically significant at 0 ACC (Fig 5d and 5g). EIN3 levels of *ctr1-8* at 0 ACC are already much greater than that in WT treated with 10 mM ACC, but *ctr1-8* seedlings at 0 ACC are known longer than 10 mM ACC-treated WT (the authors did not report the actual seedling hypocotyl length for 10 mM ACC-treated WT). It is unexplained for why the similar level of EIN3 of 1-10 mM ACC-treated WT did not result in similar levels of the ethylene response.

Response 37: Thank you for the comments. Please see **Response 29** for the unrelatedness of higher EIN3 levels and wild-type-like phenotypes of 35S: Δ NT-CTR1^{*ctr1-1*} below.

In the source file, we provided raw data for *ctr1-8* hypocotyl lengths. The *ctr1-8* produced longer hypocotyls at 0 ACC (average 3.596135 mm) than that of WT at 10 mM ACC (average 2.684171mm), which is consistent with the reviewers' comments. We are not clear about why a higher EIN3 level in the *ctr1-8* does not provide a shorter hypocotyl length than the WT. However, based on the phenotypes of 35S: Δ NT-CTR1^{*ctr1-1*} and its inconsistency in the phenotype, this may be attributed to the excess cytosolic CTR1^{*ctr1-8*} protein, which does not translocate into the nucleus, unlike endogenous WT CTR1, and may interact with other cellular processes. Since there are no other references regarding the levels of EIN3 in the *ctr1-8* mutant relative to the WT, further study may be needed to understand this discrepancy, but it is beyond the scope of the current study.

10. Data for the resilience to abiotic stress by the 35S:GFP- Δ NT-CTR1^{*ctr1-1*} need to be carefully addressed. This evidence is only obtained from a single transgenic event, and it is to be determined whether it is the transgene insertion at certain locus (or loci; there may be more than one transgenes in this line) or the effects of GFP- Δ NT-CTR1-1 fragment. Moreover, EIN3 levels determined from 35S:GFP- Δ NT-CTR1^{*ctr1-1*} do not agree with the rosette phenotype shown in Fig 6. The highly elevated EIN3 is expected to result in strong rosette growth inhibitions, and the 35S:GFP- Δ NT-CTR1^{*ctr1-1*} plants are not different from WT. The section "The increased nuclear accumulation of CTR1 enhances stress tolerance to abiotic stress" is misleading. It the 35S:GFP- Δ NT-CTR1^{*ctr1-1*} transgene, its insertion at certain loci, or the truncated Δ NT-CTR1-1, instead of CTR1, that results in the phenotype. To support the argument by the authors, one will need data from the corresponding transgenic plants.

Response 38: Thank you for the comments. To address the comments and confirm the presented results, we performed several experiments that included additional transgenic lines and controls. We included a substantial number of newly acquired results that support the role of nuclear CTR1 in stress resilience to salt and drought stress.

1. To address the comment, we included an additional independent line of 35S: Δ NT-CTR1^{*ctr1-1*} line (#28) and tested the salt and drought response. The #28 line also expressed higher levels of EIN3 and displayed significantly enhanced salt and drought tolerance than the WT (**Fig. 6j** and **Supplementary Fig. 18a**). Furthermore, to eliminate the possibility that the enhanced stress resilience of the 35S: Δ NT-CTR1^{*ctr1-1*} line is specific to the truncated form of CTR1, we

examined the salt and drought stress responses of seedlings expressing full-length CTR1 ($35S:GFP-CTR1^{ctr1-1}$) (**Fig. 6j** and **Supplementary Fig. 18a**) and found that it showed similar levels of salt and drought resistance to $35S:\Delta NT-CTR1^{ctr1-1}$. Our localization data and ethylene response kinetics analysis showed that $35S:GFP-CTR1^{ctr1-1}$ displayed some levels of constitutive CTR1 nuclear localization and significantly slower growth recovery kinetics than the WT (**Fig. 3a-d** and **Fig. 4b**). Intriguingly, unlike $35S:\Delta NT-CTR1^{ctr1-1}$, $35S:GFP-CTR1^{ctr1-1}$ seedlings are significantly smaller than the WT and $35S:\Delta NT-CTR1^{ctr1-1}$. Moreover, the hypocotyl length of $35S:GFP-CTR1^{ctr1-1}$ is significantly shorter than the WT over a broad range of ACC concentrations (**Supplementary Fig. 18d**). Examination of EIN3 levels in $35S:GFP-CTR1^{ctr1-1}$ also showed that it produced higher levels of EIN3 than WT (**Supplementary Fig. 18b-c**). We speculate that the unrelatedness of the higher levels of EIN3 and WT-like phenotype of $35S:\Delta NT-CTR1^{ctr1-1}$ could be related to their truncated, overexpression and an unknown role in the cytoplasm or the nucleus via interaction with other cellular processes. Given the significantly smaller seedling size of $35S:GFP-CTR1^{ctr1-1}$ than that of the $35S:\Delta NT-CTR1^{ctr1-1}$, the truncated form of CTR1 may have stronger effects than the full-length CTR1 in terms of seedling size via the unknown mechanism.

2. Besides, we examined the salt/drought stress response of $35S:GFP-CTR1^{ctr1-8}$ and *ctr1-8* seedlings (**Fig. 6e** and **6k**). The localization data showed that $CTR1^{CTR1-8}$ proteins do not translocate into the nucleus in response to ACC regardless of which promoters are used. Therefore, one would think that $35S:GFP-CTR1^{ctr1-8}$ seedlings would show a similar salt/drought response to that of WT seedlings as it does not affect EIN3 levels in the nucleus. We found that overexpression of $CTR1^{CTR1-8}$ protein did not enhance salt and drought tolerance, showing a similar stress response to the WT (**Fig. 6e, 6k, Supplementary Fig. 19** and **20**). However, *ctr1-8*, a moderate constitutive ethylene response mutant with a slower growth recovery kinetics, showed a strong salt/drought tolerance phenotype. Furthermore, the immunoblotting analysis showed that $35S:GFP-CTR1^{ctr1-8}$ produced a similar level of EIN3 to that of the WT (**Supplementary Fig. 18c**) despite that it produced comparable levels of $CTR1^{ctr1-8}$ protein to $CTR1^{ctr1-1}$ in $35S:GFP-CTR1^{ctr1-1}$. These results show that enhanced stress resilience to salt and drought requires nuclear-localized CTR1.

3. To demonstrate that the enhanced stress tolerance to salt and drought by nuclear-localized CTR1 is not simply due to the overexpression of the CTR1 variant in the nucleus, we examined the salt and drought stress responses of two additional lines expressing $CTR1^{ctr1-8}$ or $CTR1^{ctr1-8}$ -SV40 protein from its native promoter in the *ctr1-8* background ($CTR1p:GFP-CTR1^{ctr1-8}/ctr1-8$ and $CTR1p:GFP-CTR1^{ctr1-8}-SV40/ctr1-8$) (**Fig. 6f** and **6l**). We found that $CTR1p:GFP-CTR1^{ctr1-8}/ctr1-8$ lines are significantly bigger than the *ctr1-8*, but slightly smaller than the WT, and showed similarly or only slightly enhanced salt/drought tolerance to the WT. However, $CTR1p:GFP-CTR1^{ctr1-8}-SV40/ctr1-8$ showed comparable levels of stress resilience to the *ctr1-8* and displayed a similar size to the *ctr1-8*. We speculate that $CTR1p:GFP-CTR1^{ctr1-8}-SV40/ctr1-8$ exhibited a similar stress response to the WT despite the slower recovery kinetics than that of the *ctr1-8* (**Fig. 4c**) because the levels of EIN3 in the *ctr1-8* already met the threshold levels that could confer stress resistance. These results are likely due to the forced nuclear-localization of $CTR1^{ctr1-8}$ protein by SV40 NLS stabilizes EIN3 in the nucleus without affecting cytosol ethylene response. Whereas, $CTR1^{ctr1-8}$ protein expressed from the transgene, which does not

translocate into the nucleus, inhibits ethylene response in the cytoplasm without stabilizing EIN3 in the nucleus, thus rescuing the smaller size of *ctr1-8* and restoring the response to stress to the WT levels. Together, these results suggest that the effect of enhanced drought/salt resilience phenotypes of overexpression lines of CTR1 variants is not due to their overexpression.

4. Several control lines (*ctr1-1*, *ctr1-2*, *ebf2-3*, EIN3 OX, and complementation lines of *ctr1-2*) showed that correlation between enhanced salt and drought stress resilience to increased ethylene response (**Supplementary Fig. 19 and 20**).

Together, these results support the notion that nuclear-localized CTR1 confers stress resilience to Arabidopsis, and the increased abiotic stress resilience of *35S:ΔNT-CTR1^{ctr1-1}* is not an artifact. Furthermore, the analysis of additional transgenic lines included in the revision further supports the positive role of ethylene in drought stress tolerance in Arabidopsis. We included related information in the text.

Other comments or suggestions:

1. The antibodies for EIN3 were from Agrisera, and there is no information about the validity of this commercial antibodies from the webpage (<https://www.agrisera.com/en/artiklar/ein3.html>). Many false positive reports are resulted from inappropriate antibodies (Baker, 2005), and can the authors provide any evidence that validate the anti-EIN3 antibodies including negative positive/controls (albeit the antibodies were previously cited in PNAS)? This comment is raised because EIN3 levels and degrees of the seedling hypocotyl growth inhibition are not tightly associated (see comment 8), and personally I had poor experience with the commercial antibodies. The increase immune signal in response to ACC does not mean it is the EIN3 protein. Similarly, please report the source of anti-EIN2 antibodies. Besides, please show the full-paged immunoblots for EIN2, EIN3, and CTR1 in supplementary files for the purpose of data transparency, with the molecular-weight markers.

Response 39: We provided a requested immunoblot that contains total protein extracts of WT, *ein3eil1*, and EIN3ox seedlings without any treatment (**Supplementary Figure 17**). The EIN3 overexpression lines were obtained from Dr. Jose Alonso's laboratory and were originally generated in Dr. Ecker's laboratory (1). We also provided uncropped x-ray images of the immunoblot results in the source files. For the EIN2 antibody, we did not use an EIN2 antibody in the study. The data presented in Figure 5i is an in vitro kinase assay of EIN2-CEND and other proteins (EIN3 and EBF) using p32 [ATP]. We included EIN2-CEND as a positive control of CTR1-mediated substrate phosphorylation. The phosphorylation of EIN2-CEND has been demonstrated in our previous work (2). And the EIN2-CEND bands in Fig 3f are not from immunoblotting using anti-EIN2 antibody. The immunoblotting was done using an anti-GFP antibody to detect GFP-fused EIN2-CEND (phos-tag analysis). To avoid confusion, we modified the name of the protein to GFP-EIN2-CEND.

Reference:

1. Activation of the ethylene gas response pathway in Arabidopsis by the nuclear protein ETHYLENE-INSENSITIVE3 and related proteins (1997), Q Chao et al., Cell
2. CTR1 phosphorylates the central regulator EIN2 to control ethylene hormone signaling from the ER membrane to the nucleus in Arabidopsis (2012), Li et al., PNAS

2. For the seedling hypocotyl measurements (not the growth rates in Supplementary Fig 8), data for ACC- or ethylene-treated WT as controls are absent. Besides, the graphic presentations for the dose-response curves are erroneous. The distance between each concentration point at the x-axis should not be equal because the dosage is not linear.

Response 40: We provided the hypocotyl measurement of WT in Fig. 5h and 5i and changed the graphic presentation as recommended.

3. The section title “The CTR1 N-terminus inhibits CTR1 nuclear trafficking” may be misleading. Nuclear trafficking still occurs for GFP-CTR1 that has the N-terminus. It is the association of CTR1 with the ethylene receptors via the N-terminus to retain the protein at the ER.

Response 41: The full-length CTR1 with a N-terminus translocates to the nucleus. Although the detailed mechanistic insights are not resolved, our examination of the subcellular localization of N-terminus truncated CTR1 suggests that ethylene-binding receptors cause the conformational changes of CTR1, which includes the N-terminus of CTR1, thus releasing CTR1 from the receptors. Furthermore, truncated CTR1 proteins were found in the nucleus regardless of treatment to increase the stability of EIN3 in the nucleus. Thus, technically, the N-terminal domain of CTR1 inhibits CTR1 nuclear trafficking by preventing its cytosolic release by direct interaction with the receptors in the absence of ethylene. We have therefore left the wording unchanged.

4. In introduction, the description for CTR1-mediated EIN2 phosphorylation and ethylene signaling repression may need updating another study showing full-length EIN2 localization to the nucleus, complex EIN2 cleavages, and weak association of EIN2 phosphorylation status and ethylene signaling activation to better describe other aspects of ethylene signaling (Wen et al., 2012; Zhang et al., 2020). Inference made from different aspects of findings, instead of limited to the canonical model that may change from time to time and prevent new findings, is important to development of new discovery.

Response 42: Thank you for the comments. We included the suggested reference (lines 64).

5. I noticed that some of the LSCM images are from cells with a relatively small size, and it appears that those images were taken from cells of tip regions (root tips or apical regions) of a seedling. Probably I was wrong; however, if any LSCM images were taken from cells of tip regions, background noises that gives false positive signals should be considered (Zhang et al., 2020).

Response 43: Most LSM images were taken from the areas below the hook and above the elongation zone of the seedlings.

6. Seeding hypocotyl measurement needs to be described in Materials and Methods; the WT seedlings appear to be unbelievably short (6mm). The seedling hypocotyl measurement is also problematic for WT in Fig 2e; it is only about 5 mm in length. Please refer to the typical seedling hypocotyl measurement by Hua and Meyerowitz, where the WT seedling is about ≥ 12 mm (Hua and Meyerowitz, 1998). A proper measurement may better unveil the difference in degrees of the ethylene response. When the hypocotyls are relatively short, minor differences cannot be unveiled and the inference will be made differently.

Response 44: Thank you for the comments. We repeated hypocotyl measurement data, which showed similar results as to the difference in WT and CTR1 transgenic lines in the original submission. We replaced it with the newly generated data.

7. For graphical data presentations, error bars should be presented on both sides of a bar, and the y-axis should start from 0. The data analyses involve inferential statistics, and it is SE instead of SD to be presented (Cumming et al., 2007).

Response 45: We fixed the data presentation as suggested.

8. Fig 2d, the percentage for nuclear localized CTR1, fix CTR1 to Δ NT-CTR1.

Response 46: We fixed the typo.

9. In addition to the co-expression, data for Supplementary Fig 11 needs controls for EIN2-C and ENAP1 alone to show their subcellular localizations in the absence of each other.

Response 47: We provided the suggested data in the Supplemental section (**Supplementary Fig. 16**).

10. The LSCM images for Supplementary Fig 4 for the nuclear EIN2 look more like at the ER network surrounding the nucleus. The nuclear EIN2 is characteristic of speckles that are revealed in independent LSCM images (Qiao et al., 2012; Zhang et al., 2020).

Response 48: Thank you for the comments. We agree that the current data may not clearly show the speckle of EIN2 in the nucleus. We provided an enlarged image of the nucleus at the 60-minute time point, which shows clumps of speckles in the nucleus (**Supplementary Fig. 5**).

Reviewer #1 (Remarks to the Author):

The revised manuscript of Park et al has significantly improved. The authors have done a lot of extra work, and provide compelling new data that makes the conclusions much stronger. The new results largely strengthen the original conclusions, while some elements were rephrased or presented differently in the light of the new findings. Particularly, I cheer the extra stress experiments, that provide a much better understanding that nuclear localized CTR1 can act as a signal to enable stress tolerance. The new Figure 6 and the data interpretation is now much stronger and convincing. Also, additional supplemental material is provided to strengthen some other techniques or methods/materials used in this study.

All my major concerns were addressed. I believe that even some new insights obtained in this revision round, open up new routes of future investigation (e.g. role of the ctr1-8 mutation in stress resilience). The new manuscript reads well and the data is nicely presented.

The overall impact of the findings presented in this study is huge to the field of ethylene signaling and responses. I consider this a major breakthrough in the field that will lead to many new exciting future studies.

Reviewer #2 (Remarks to the Author):

This revised manuscript has been much improved compared to the original version.

I have only a few minor comments:

1. There might be a better section title than "The CTR1 N-terminus inhibits CTR1 nuclear trafficking". This title is somewhat misleading given that full-length CTR1 (with the N-terminus) goes into the nucleus. Perhaps the title should be along the lines of "CTR1 nuclear trafficking is regulated through the CTR1 N-terminus"

2. It would be helpful if the data in Supplementary Figure 7 were quantified.

3. It's a little confusing how panels in some figures (e.g., Fig. 1 and Suppl. Fig. 2) are referred to non-consecutively in the text. Supplementary Fig. 2d-f are not explicitly cited anywhere in the main text. It might be better to separate the panels by the point they make, rather than grouping them because they are the same technique. For example, Suppl. Fig. 2a could be shown next to Suppl. Fig. 3. But this is relatively minor.

4. In supplementary Figure 5, is the same hypocotyl shown at each time point? If yes, change the legend so as not to refer to etiolated seedlings (plural).

5. When examining the time course of nuclear localization in Fig. 2k and 2l, did you control for the possibility of photo-bleaching?

6. Line 175 should say "localized to the nucleus IN the"

Reviewer #3 (Remarks to the Author):

please find my comments in the attached file.

Reviewer #3 Attachment on the following page.

The manuscript (347433) by Park and co-authors answered most the comments with addition of an overwhelming amount of data, which I highly appreciated. The salt resilience conferred by those different CTR1 variants is intriguing. I have no major concerns about the data quality. The inference for roles of the nuclear CTR1 made in this work is primarily based on ectopic transgene expression for CTR1 variants, which may exaggerate the effect of the endogenous level of nuclear CTR1 variants. My concern is: to what extent the salt resilience can be attributed to endogenous level of the nuclear CTR1? The authors claimed roles of the nuclear CTR1 in salt resilience; however, they never tested effects of the wild-type CTR1 on salt tolerance. Instead, the conclusion was drawn from investigations involving CTR1-8, CTR1-1, or the Δ NT-CTR1. These did not truly reflect native roles of the CTR1 protein in response to the stress, as a mechanism for fitness. *GFP-CTR1* transgenic plants were investigated for their subcellular localizations and growth recovery kinetics. I do not get the point for why these readily available materials were not tested for their model. The explanation for the inconsistency for the elevated EIN3 level and unaffected seedling phenotype for those CTR1 variant lines was too speculative. Part of the overwhelming amount of data presented in this work somehow compromises the depth of investigations (especially for that in the last section), overlooking other possibilities. There are also places where the description was vague and easily led to logical fallacy.

1. Western blots for CTR1 and CTR1-1 revealed only a small fraction of the total CTR1/CTR1-1 allocated to the nucleus after ACC treatment (*Supplementary Figure S4 and Figure 3C*). This agrees with the ACC-induced YFP-CTR1 accumulation in the nucleus (*Fig. 1C*) for *CTR1p:YFP-CTR1* plants. By contrast, a large fraction of the ectopically expressed GFP-CTR1 accumulates in the nucleus, regardless of the silver or ACC treatment for *35S:GFP-CTR1* plants (*Fig. 1h*), likely due to the ectopically expressed GFP-CTR1 reaching the ethylene receptor binding limit at the ER. Given that ACC induced GFP-CTR1 stabilization, determined by CHX treatment/western blots, the authors proposed that ACC stabilizes CTR1. As a result, the ACC-induced CTR1 stabilization led to higher CTR1 levels such that excess CTR1 may translocate to the nucleus. This inference appears not supported by the higher CTR1 levels for the non-treated than the ACC-treated wild type (*Supplementary Figure S4*) or *ctr1-1* plants (*Fig. 3C*). The unchanged % of nuclear localized GFP-CTR1 for silver- and ACC-treated seedlings, determined by laser scanning confocal microscopy (*Fig. 1h*), did not agree with the ACC-induced nuclear GFP-CTR1 accumulation determined by western blots (*Fig. 1j*).

Supplementary Figure 4

Levels of total CTR1/CTR1-1 are higher for the non-treated than the ACC-treated plants.

Nuclear GFP-CTR1 levels unchanged

The nuclear GFP-CTR1 % is similar for the silver- and ACC-treated seedlings. By contrast, western blots determine GFP-CTR1 accumulation in the nucleus for the ACC-treated but not for the non-treated plants.

2. The result for ACC-induced GFP-CTR1 stabilization should not be described as ACC-induced CTR1 stabilization (lines 108-109), because the authors did not conduct this experiment and the protein property for CTR1 could be changed by the GFP fusion. Similarly, there are places where experiments/results involving GFP-fused CTR1 variants were described as CTR1 in the main text. It is important to clearly and correctly report the exact proteins/genes involved for the corresponding experiments/results to prevent vagueness. It is the inference but not conclusion that is made for the CTR1 protein from experiments involving the GFP-fused CTR1 variants. It is also a logical fallacy when the different protein variants are equalized to make a conclusion.
3. Inference made in this study primarily involved results from single transgenic lines. It should be noted that different transgenic events may give different effects due to differential transgene expression. Guidelines for several scientific journals require involvement of multiple (at least 3) transgenic lines when transgenes are involved [the guideline by *New Phytologist* as an example: *Studies using transgenic organisms that employ a single transgenic primary event will not be considered (i.e. multiple independent lines are required)*]. This concern was raised in my previous

reviewing. The authors indicated in their rebuttal that several lines were investigated; however, the relevant data were not provided. Different expression levels conceivably affect the growth recovery kinetics, and the inference could be made differently due to different protein levels. For instance, the growth recovery kinetics differed greatly for *ctr1-8* (Fig. 4c) and *CTR1p:GFP-CTR1-8/ctr1-2* seedlings (supplementary Figure 12), indicative of the kinetics affected by CTR1-8 levels. Another example is that the seedling growth recovery kinetics differed for *CTR1p:GFP-CTR1/ctr1-2* and WT (Supplementary Figure S12), even if the transgene complements *ctr1-2*. Evidence for the growth recovery kinetics determined on the base of “time to reach pre-treatment growth rate” is only limited to a time point while the whole recovery curves left not compared, which are major part of the kinetics.

The growth recovery kinetics differ greatly for *ctr1-8* and *GFP-CTR1-8* seedlings, and the growth recovery kinetics for *GFP-CTR1/ctr1-2* seedlings and WT differed.

- I am concerned about the LSCM images for the GFP-CTR1 nuclear transport that could be resulted from a focal plan shift. For instance, the images for *Supplementary Figure S6* clearly revealed different focal plans. The cell outlines and the circled (red) nucleus change or disappear (see below).

Supplementary Figure 6

5. The authors did not truly answer for why the elevated EIN3 levels of *35S:GFP-ΔNT-CTR1-1* unable to produce a typical seedling phenotype, although a vague speculation is proposed. This comment also applies to data from Supplementary Figure 18, where EIN3 levels are highly induced by GFP-CTR1-1, GFP-ΔNT-CTR1-1, and GFP-CTR1-8, and the plants did not exhibit a prominent ethylene growth inhibition phenotypes as expected. Results and descriptions for the last section (starting from line 314) are somehow difficult to follow, without the information for the genetic background of the transgenic lines. I assume they are in the WT background if not specified, but I am not sure if my assumption was correct.
6. Lines 305-307. The description “The hypersensitive ACC response of *CTR1p:GFP-ΔNT-CTR1-1/ctr1-2* as compared to *CTR1p:GFP-CTR1-1/ctr1-2...*” lacks a control (*ctr1-2*) (Fig. 5h). It has been shown that *ctr1-2* seedlings are ethylene responsive. If *ctr1-2* and *CTR1p:GFP-ΔNT-CTR1-1/ctr1-2* seedlings exhibit a similar response curve to ACC, the GFP-ΔNT-CTR1-1 protein would have little effect on the ethylene response, as oppose to the propose positive role in ethylene responses.

Minors or optional:

1. This was raised in my previous comment. The laser scanning confocal imaging (LSCM) for EIN2-mCherry localization to the nucleus is somehow inconclusive (Supplementary Figure S5). The fluorescence primarily appeared a networking structure, reminiscent of the ER network that surrounds the nucleus. Alternatively, the indicated fluorescence might not represent a nucleus (the indicated nucleus appeared too large to be a nucleus). This could arise from focal plane shift as mentioned in my previous comment, inferred from the different shape for adjacent cells of the four images. I was able to observe an empty nucleus (blue circles; see below) in the first three images, and the nucleus disappeared for the 60 min image, implying a focal plan shift. The kinetics for EIN2 nuclear movement should not be determined from a single cell for the lack of evidence power for cell biology studies. Perhaps, evidence for the EIN2 movement is not essential to this work and could be removed because similar results have been reported in previous studies (Qiao *et al.*, 2012).

Supplementary Figure 5

2. The way for presenting those CTR1 variants was confusing. For examples, CTR1-1 was presented as CTR1^{CTR1-1} and CTR1-8 as CTR1^{CTR1-8}. Given that the CTR1-1 protein is encoded by the *ctr1-1* gene, presenting the protein as CTR1-1 is sufficient and clear.
3. Line 387 “CTR1 lacks a canonical NLS”. This statement may not be true. Sequence analyses (http://nls-mapper.iab.keio.ac.jp/cgi-bin/NLS_Mapper_form.cgi#opennewwindow) for the GFP-fused CTR1 variants revealed several peptide sequences that facilitate CTR1 localization in the cytoplasm and nucleus (see below). These may explain the dual localizations of CTR1. On the other hand, it is to be explained for why CTR1-8 is not localized to

the nucleus.

cNLS Mapper Result

Predicted NLSs in query sequence	
MVSKGEELFTGVVPIIVLVELDGDVNGHKFSVSGEGEDATYGKLTILKFICT	50
TGKLPVWPVTLVITFGYGVQCFARYPDHMKQHDFFKSAMPEGYVQERTIF	100
FKDDGNYKTRAEVKFEGDILVNRIELKGIDFKEDGNILGHKLEYNYNSHN	150
VYIMADKQKNGIKVNFKIRHNIEDGSQLADHYQQNTPIGDGPVLLPDNH	200
YLSYQSALSKDPNEKRDMVLLFVTAAGITLGMDELKVDNENPGRRSN	250
YTLISQFSDQVYSVTGAPPPHYDLSSENRSHNSGHTGKAAERGGF	300
DWDPSGGGGDHRLLHMQPHRVGHNTYASSLGLRQSSGSSFGESSLGDY	350
YHPILSAAANEIESVGFPPDDGFRGFGGGGDLRIQMAADSAGSSSGK	400
SWAQQTEESYQLALALRLSSEATCADDPNFLDPVPEALRTSPSSAE	450
TVSHRFVVMGCLSYDVPDGFYMMGLDPYIVTLCIDLHESGRIPSIES	500
LRAVDSGVDSLEAIVDRRSDPAFKELHHRVHDI SCSCITITKEVVDQLA	550
KLICHRMGPPVINGEDELVPMWKECIDGLKEIFKVVVPVIGSLVGLCRHR	600
ALLFKVLADIDLPCRIAGCKYCHRDAAASCLVKPGLDREYVLDVGLKRP	650
GHLVEPDSLLNGPSSISISSPLRFPKPYEPAVDFRLLAKQYFSDSQSL	700
HLVFPASDDMGFSHFHRQYDHPGGENDALAEHG66SLPPSAMHPPQHNM	750
RASHQIEAAPMHPPTISQYVPHRAHRELGLDGDMDIPWCDLNIKEKIGA	800
CSFGTVHRAEWHGSDVAVKILMEQDFHAERVNEFLREVAINKRLRHPNIV	850
LFMGAVTQPPNLSIVTEYLSRGLYRLHRSAGAREQLDERRRLSMAYDVA	900
KGMVYLNHRNPPIVHRDLKSPMLVDKYYTVKCDLGRSLKASTFLSSK	950
SAAGTPEWMAPEVLRDEPSNEKSDVYSFGVILWELATLQPPWGNLHPAQV	1000
YAAVCFKCKRLRIPRNLHPQVAIIIEGCWTRHPWKRPSPATINDLLRPLI	1050
KSAVPPPNRSDL	1062

Predicted monopartite NLS

Pos.	Sequence	Score

Predicted bipartite NLS

Pos.	Sequence	Score
83	DFFKSAMPEGYVQERTIFFKDDGNYKTRAE	3
595	GLCRHRALLFKVLADIDLPCRIAGCKY	4
598	RHRALLFKVLADIDLPCRIAGCKYCHRD	3.7
645	DLVCKPGLHVEPDSLLNGPSSISISSPLRFP	3.9
796	EKIGAGSFGTVHRAEWHGSDVAVKILME	3.8
815	DVAVKILMEQDFHAERVNEFLREVAINKRLR	3.3
836	REVAINKRLRHPNIVLFGAVTQPPNLSIVTEY	3
1032	EPWKRPSPATINDLLRPLIKSAVPPPNRSD	3.9

Peptide sequence facilitating GFP-CTR1 transport to the nucleus.

cNLS Mapper Result

Predicted NLSs in query sequence	
MVSKGEELFTGVVPIIVLVELDGDVNGHKFSVSGEGEDATYGKLTILKFICT	50
TGKLPVWPVTLVITFGYGVQCFARYPDHMKQHDFFKSAMPEGYVQERTIF	100
FKDDGNYKTRAEVKFEGDILVNRIELKGIDFKEDGNILGHKLEYNYNSHN	150
VYIMADKQKNGIKVNFKIRHNIEDGSQLADHYQQNTPIGDGPVLLPDNH	200
YLSYQSALSKDPNEKRDMVLLFVTAAGITLGMDELKVDNENPGRRSN	250
YTLISQFSDQVYSVTGAPPPHYDLSSENRSHNSGHTGKAAERGGF	300
DWDPSGGGGDHRLLHMQPHRVGHNTYASSLGLRQSSGSSFGESSLGDY	350
YHPILSAAANEIESVGFPPDDGFRGFGGGGDLRIQMAADSAGSSSGK	400
SWAQQTEESYQLALALRLSSEATCADDPNFLDPVPEALRTSPSSAE	450
TVSHRFVVMGCLSYDVPDGFYMMGLDPYIVTLCIDLHESGRIPSIES	500
LRAVDSGVDSLEAIVDRRSDPAFKELHHRVHDI SCSCITITKEVVDQLA	550
KLICHRMGPPVINGEDELVPMWKECIDGLKEIFKVVVPVIGSLVGLCRHR	600
ALLFKVLADIDLPCRIAGCKYCHRDAAASCLVKPGLDREYVLDVGLKRP	650
GHLVEPDSLLNGPSSISISSPLRFPKPYEPAVDFRLLAKQYFSDSQSL	700
HLVFPASDDMGFSHFHRQYDHPGGENDALAEHG66SLPPSAMHPPQHNM	750
RASHQIEAAPMHPPTISQYVPHRAHRELGLDGDMDIPWCDLNIKEKIGA	800
CSFGTVHRAEWHGSDVAVKILMEQDFHAERVNEFLREVAINKRLRHPNIV	850
LFMGAVTQPPNLSIVTEYLSRGLYRLHRSAGAREQLDERRRLSMAYDVA	900
KGMVYLNHRNPPIVHRDLKSPMLVDKYYTVKCDLGRSLKASTFLSSK	950
SAAGTPEWMAPEVLRDEPSNEKSDVYSFGVILWELATLQPPWGNLHPAQV	1000
YAAVCFKCKRLRIPRNLHPQVAIIIEGCWTRHPWKRPSPATINDLLRPLI	1050
KSAVPPPNRSDL	532

Predicted monopartite NLS

Pos.	Sequence	Score

Predicted bipartite NLS

Pos.	Sequence	Score
83	DFFKSAMPEGYVQERTIFFKDDGNYKTRAE	3
266	EKIGAGSFGTVHRAEWHGSDVAVKILME	3.8
285	DVAVKILMEQDFHAERVNEFLREVAINKRLR	3.3
306	REVAINKRLRHPNIVLFGAVTQPPNLSIVTEY	3
502	EPWKRPSPATINDLLRPLIKSAVPPPNRSD	3.9

Peptide sequence facilitating GFP-CTR1 ΔNT transport to the nucleus.

(b) Cut-off score

cNLS Mapper extracts putative NLS sequences with a score equal to or more than the selected cut-off score. Higher scores indicate stronger NLS activities. Briefly, a GUS-GFP reporter protein fused to an NLS with a score of 8, 9, or 10 is exclusively localized to the nucleus, that with a score of 7 or 8 partially localized to the nucleus, that with a score of 3, 4, or 5 localized to both the nucleus and the cytoplasm, and that with a score of 1 or 2 localized to the cytoplasm (see Supplemental Figure S1 in refs. 1 or 3 for detail).

- I appreciated that the authors were able to answer my concern for the CTR1 protein localization involving polyclonal antibodies for CTR1 (previous comment 5). GE Schaller previous showed localization of CTR1-8 to the cytoplasmic soluble fraction, with the total CTR1-8 level unchanged. Results from the western blot for CTR1-8 appeared odd. The western blot by Park and co-authors showed a greatly reduced level for the cytosol CTR1-8, as compared to the level from the total protein (*Fig. 2i*). By contrast, CTR1/CTR1-1 levels are similar for proteins from the total and cytoplasmic fraction. The cytoplasmic fraction did not involve ultracentrifugation that removes the membrane fraction, and the cytoplasmic fraction may represent proteins from the membrane and soluble fractions. Can the authors explain the results? The authors may need to describe in methods for how the total protein was prepared.

Supplementary Figure 4

- Western blots for CTR1 (*Supplementary Figure S4*) are a critical piece of evidence for nuclear localization of a small fraction of the total CTR1 and should be presented in Figure 1 described in the first section (*Ethylene-induced ER-to-nucleus translocation of CTR1*). Placing the data in *Supplementary Figure S4* may not draw sufficient attention when the amount of data is overwhelming.
- This was raised in my previous comment. The LSCM study for localization of the several different CTR1 variants involved silver or ACC treatment. Presenting localizations under the ambient condition (non-treated) is equally important as a control. However, this control (no treatment) is missing, except for that for western blots. The authors replied that there is no need for the control. Without the control, we cannot judge the respective effects of silver and ACC on CTR1 localization and

abundance.

7. I completely agree with the rebuttal for presenting the data in percentage for samples with a size much smaller than 100, where the small sample size produces a lower power of evidence. It is exactly my concern that whether the reduced power of evidence is sufficient to make a conclusion.
8. Evidence for the conclusion for GFP-CTR1 decrease after ethylene removal appear weak (*Supplementary Figure S7*). The extent of GFP-CTR1 reduction determined by western blots (*Supplementary Figure S7*) and LSCM (*Fig. 1i*) differed greatly. I agree that there appears to have a marginal change; however, the change is too marginal to be conclusive given that western blots can be affected by many factors and are not sufficiently sensitive/stable to detect marginal changes in protein levels. Even if the change is real, contribution of the marginal change to a biological activity is unlikely determined.

Fig. 1

Supplementary Figure 7

Changes in GFP-CTR1 levels are large for results determined by LSCM and marginal for results determined by western blots.

9. The seedling growth recovery by GFP-CTR1-8-SV40 is better tested in *ctr1-2* but not *ctr1-8* background.
10. I suggest the authors use CTR1 polyclonal antibodies for western blots involving GFP-fused CTR1 variants, instead of using the GFP antibodies. This may detect both the endogenous CTR1 and the ectopically expressed GFP-CTR1 variants, facilitating the revealing of levels of the two types of proteins.
11. Western blots show a much lower EIN3 level for ACC-treated WT were than for non-treated *ctr1-8* plants (*Fig. 5e*). These results appear abnormal, as *ctr1-8* is hylomorphic. Regardless, EIN3 levels for WT were still much lower than that for

ctr1-8 plants by the 10 μ M ACC treatment.

Qiao H, Shen Z, Huang S-sC, Schmitz RJ, Urich MA, Briggs SP, Ecker JR. 2012. Processing and Subcellular Trafficking of ER-Tethered EIN2 Control Response to Ethylene Gas. *Science* 338(6105): 390-393.

REVIEWER COMMENTS

Reviewer #1 (Remarks to the Author):

The revised manuscript of Park et al has significantly improved. The authors have done a lot of extra work, and provide compelling new data that makes the conclusions much stronger. The new results largely strengthen the original conclusions, while some elements were rephrased or presented differently in the light of the new findings. Particularly, I cheer the extra stress experiments, that provide a much better understanding that nuclear localized CTR1 can act as a signal to enable stress tolerance. The new Figure 6 and the data interpretation is now much stronger and convincing. Also, additional supplemental material is provided to strengthen some other techniques or methods/materials used in this study.

All my major concerns were addressed. I believe that even some new insights obtained in this revision round, open up new routes of future investigation (e.g. role of the ctr1-8 mutation in stress resilience). The new manuscript reads well and the data is nicely presented.

The overall impact of the findings presented in this study is huge to the field of ethylene signaling and responses. I consider this a major breakthrough in the field that will lead to many new exciting future studies.

Response: Thank you very much for taking the time to review our manuscript and for the positive comments.

Reviewer #2 (Remarks to the Author):

This revised manuscript has been much improved compared to the original version.

I have only a few minor comments:

1. There might be a better section title than “The CTR1 N-terminus inhibits CTR1 nuclear trafficking”. This title is somewhat misleading given that full-length CTR1 (with the N-terminus) goes into the nucleus. Perhaps the title should be along the lines of “CTR1 nuclear trafficking is regulated through the CTR1 N-terminus”

Response: Thank you for the suggestion. We have changed the title to “CTR1 nuclear trafficking is regulated through the CTR1 N-terminus”

2. It would be helpful if the data in Supplementary Figure 7 were quantified.

Response: We found that the quality of the loading control is not adequate for quantification. Thus, we replaced the data with better loading controls and quantified the relative band intensities of GFP-CTR1.

3. It's a little confusing how panels in some figures (e.g., Fig. 1 and Suppl. Fig. 2) are referred to non-consecutively in the text. Supplementary Fig. 2d-f are not explicitly cited anywhere in the main text. It might be better to separate the panels by the point they make, rather than grouping them because they are the same technique. For example, Suppl. Fig. 2a could be shown next to

Suppl. Fig. 3. But this is relatively minor.

Response: Thank you for the suggestions. We have put them together to collectively show the subcellular localization of seedlings expressing different CTR1 variants from 35S or native promoters without ACC or silver treatment and to demonstrate that the different localization of different CTR1 variants is not due to focal plane changes. We have also clearly cited all the noted figures in the text.

4. In supplementary Figure 5, is the same hypocotyl shown at each time point? If yes, change the legend so as not to refer to etiolated seedlings (plural).

Response: The images are time-course of a same seedling. Due to the comments of reviewer 3, we removed the figure as previous work already published EIN2 movement kinetics.

5. When examining the time course of nuclear localization in Fig. 2k and 2l, did you control for the possibility of photo-bleaching?

Response: We think reviewer is referring Fig. 1k and 1l. We did examine if the results of Fig. 1i were affected by photobleaching. We imaged GFP-CTR1 expressed in different seedlings (non-paired sampling) at different time points after ethylene removal to eliminate the effect of extended laser exposure. We found that GFP-CTR1 fluorescence was gradually reduced after ethylene withdrawal, similar to the paired samples (Fig. 1i), although we noticed that the time-course images of an identical seedling after ethylene removal showed a slightly faster reduction of GFP fluorescence than seedlings imaged at different times after ethylene removal. This indicates that there might be some minor photobleaching in paired sampling approaches. However, despite this marginal difference in reduction kinetics, both approaches showed that GFP fluorescence in both the cytosol and the nucleus was reduced after ethylene removal, suggesting that ethylene influences GFP-CTR1 abundance. Likewise, seedlings imaged at different time points after ACC treatment (non-paired sampling) showed somewhat enhanced GFP-CTR1 fluorescence, as shown in Supplementary Figure 2, compared to the paired sampling approach. This suggests that there was some photobleaching that diminished the effects of ACC on GFP-CTR1 fluorescence in paired sampling approaches (Fig. 1k), but GFP-CTR1 still translocated into the nucleus at a similar time point (~30 min) in the non-paired sampling approach. The difference between GFP-CTR1 fluorescence intensities in the presence and absence of ACC in the identical experimental setting (paired sampling with prolonged laser exposure) (Fig. 1k and 1l) is also correlated to the results that ACC increases CTR1 protein abundance. We have not included the results of non-paired sampling approach since it seems redundant and gives the same results, but if the reviewer think it is necessary, then we will include the results upon request.

6. Line 175 should say “localized to the nucleus IN the”

Response: Thank you for catching the error. We fixed it accordingly.

Reviewer #3 (Remarks to the Author):

The manuscript (347433) by Park and co-authors answered most the comments with addition of an overwhelming amount of data, which I highly appreciated. The salt resilience conferred by those different CTR1 variants is intriguing. I have no major concerns about the data quality. The inference for roles of the nuclear CTR1 made in this work is primarily based on ectopic transgene expression for CTR1 variants, which may exaggerate the effect of the endogenous level of nuclear CTR1 variants. My concern is: to what extent the salt resilience can be attributed to endogenous level of the nuclear CTR1? The authors claimed roles of the nuclear CTR1 in salt resilience; however, they never tested effects of the wild-type CTR1 on salt tolerance. Instead, the conclusion was drawn from investigations involving CTR1-8, CTR1-1, or the Δ NTCTR1. These did not truly reflect native roles of the CTR1 protein in response to the stress, as a mechanism for fitness. GFP-CTR1 transgenic plants were investigated for their subcellular localizations and growth recovery kinetics. I do not get the point for why these readily available materials were not tested for their model. The explanation for the inconsistency for the elevated EIN3 level and unaffected seedling phenotype for those CTR1 variant lines was too speculative. Part of the overwhelming amount of data presented in this work somehow compromises the depth of investigations (especially for that in the last section), overlooking other possibilities. There are also places where the description was vague and easily led to logical fallacy.

Response: Thank you for your comments and valuable inputs. We revised the stress response data based on the original data presented, which focused on the inactive form of CTR1 as ethylene signaling inactivates CTR1, and the kinetics data showed that kinase activity is not required for CTR1 nuclear translocation. Furthermore, we added additional overexpression lines to further support the link between CTR1 nuclear translocation and stress responses. We also investigated the correlation of CTR1 nuclear localization and stress responses using native promoter-driven lines to remove the possibility that the effects of nuclear CTR1 protein on stress responses are not an artifact (Figs. 6f and 6l). We believe the presented data is sufficient to link CTR1 nuclear localization and stress response, but acknowledge the reviewer's point. To address the comments, we examined the germination rate of two independent lines of WT CTR1 overexpression (ox) seedlings on plates containing salt because the survival rates of CTR1-1, CTR1-8, and Δ NTCTR1 overexpression lines in soil-based experiments correlate to their germination and survival on salt plates. Similar to CTR1-1ox seedlings, WT CTR1ox lines showed a higher germination rate than WT in medium containing higher salt concentrations (**Supplemental Fig. 18e**).

We are not sure to what extent the nuclear localization of CTR1 contributes to salt stress tolerance. However, the constitutively nuclear localized CTR1-8-SV40 protein expressed from a native promoter conferred significant tolerance relative to the control plants expressing CTR1-8 protein, which does not translocate into the nucleus. This indicates that nuclear-localized CTR1 has a substantial influence on salt stress tolerance. Furthermore, Supplemental Fig. 20 showed that the enhanced nuclear localization of CTR1 proteins, whether inactive or truncated, confers better survival on salt plants than constitutive ethylene response mutants (i.e., *ctr1-1*, *ctr1-2*, and *ebf2-3*). Plants with EIN3 overexpression showed significantly higher survival than all other genotypes tested. This indicates a strong correlation between EIN3 levels and salt tolerance.

Thus, it is plausible to conclude that CTR1-mediated EIN3 stabilization in the nucleus plays a significant role in salt stress response.

We do not know the precise mechanism for the discrepancy between EIN3 levels and phenotypes of lines expressing CTR1 variants (CTR1-1, and Δ NTCTR1), but provide our best speculation based on the results. All lines with disharmony of the EIN3 levels and plant sizes have one thing in common in that CTR1 variant proteins (CTR1-1 and Δ NTCTR1 proteins) localize to the nucleus. These results indicate that the increased abundance of the CTR1 protein in the nucleus or cytoplasm (both needs to travel from cytosol to the nucleus) may interact with an unknown pathway that influences plant development, such as the size of plants, by lessening the effects of EIN3. Given the uncertain outcomes of the perturbation of complex signaling pathways, this speculation is just one of many possible scenarios that could underlie the phenomenon. At this time, we cannot provide a more precise or complete answer to this question, but further studies on the cytoplasmic and nuclear roles of CTR1 may reveal the mechanism behind the phenomenon.

Western blots for CTR1 and CTR1-1 revealed only a small fraction of the total CTR1/CTR1-1 allocated to the nucleus after ACC treatment (Supplementary Figure S4 and Figure 3C). This agrees with the ACC-induced YFP-CTR1 accumulation in the nucleus (Fig. 1C) for CTR1p:YFP-CTR1 plants. By contrast, a large fraction of the ectopically expressed GFP-CTR1 accumulates in the nucleus, regardless of the silver or ACC treatment for 35S:GFP-CTR1 plants (Fig. 1h), likely due to the ectopically expressed GFP-CTR1 reaching the ethylene receptor binding limit at the ER. Given that ACC induced GFP-CTR1 stabilization, determined by CHX treatment/western blots, the authors proposed that ACC stabilizes CTR1. As a result, the ACC-induced CTR1 stabilization led to higher CTR1 levels such that excess CTR1 may translocate to the nucleus. This inference appears not supported by the higher CTR1 levels for the non-treated than the ACC-treated wild type (Supplementary Figure S4) or *ctr1-1* plants (Fig. 3C). The unchanged % of nuclear localized GFP-CTR1 for silver- and ACC-treated seedlings, determined by laser scanning confocal microscopy (Fig. 1h), did not agree with the ACC-induced nuclear GFP-CTR1 accumulation determined by western blots (Fig. 1j). The nuclear GFP-CTR1 % is similar for the silver- and ACC-treated seedlings. By contrast, western blots determine GFP-CTR1 accumulation in the nucleus for the ACC-treated but not for the non-treated plants.

Response: Thank you for the comments. We stated that ACC-induced stabilization and its potential role in CTR1 nuclear translocation as a possible mechanism that may explain the constitutive nuclear localization of CTR1 in overexpression lines. As indicated by the reviewer, we do not have compelling evidence to support this speculation, and it does not align with the fractionation data. Thus, it is possible that ACC-induced increases in CTR1 abundance may have nothing to do with the mechanism directing CTR1 nuclear translocation or that ACC stabilizes CTR1 with slower kinetics. To address the reviewer's concerns, we revised the text to minimize the emphasis on the implications of ACC-induced stabilization and its potential role in CTR1 nuclear localization.

To correct the confusion of Figs 1h and 1i (previous Fig. 1j): Fig. 1h shows images of WT CTR1 overexpression lines, which are expected to be localized in the nucleus regardless of ACC or

silver nitrate treatment; Fig.1i depicts a fractionation analysis of seedlings expressing WT CTR1 expression from a native promoter in a *ctr1-2* background, demonstrating that CTR1 translocates into the nucleus in the presence of ACC.

1. The result for ACC-induced GFP-CTR1 stabilization should not be described as ACC-induced CTR1 stabilization (lines 108-109), because the authors did not conduct this experiment and the protein property for CTR1 could be changed by the GFP fusion. Similarly, there are places where experiments/results involving GFP-fused CTR1 variants were described as CTR1 in the main text. It is important to clearly and correctly report the exact proteins/genes involved for the corresponding experiments/results to prevent vagueness. It is the inference but not conclusion that is made for the CTR1 protein from experiments involving the GFP-fused CTR1 variants. It is also a logical fallacy when the different protein variants are equalized to make a conclusion.

Response: The ACC-induced increase in CTR1 protein levels has been previously demonstrated by other group (Gao et al, 2003), in which they showed that ACC treatment increases endogenous CTR1 protein abundance without affecting CTR1 mRNA levels. Although we used transgenic lines to examine the influence of ACC on the protein abundance of GFP-fused CTR1, the effect of ACC on GFP-CTR1 is similar to what was previously shown (Gao et al, 2003). Further, the *GFP-CTR1* transgenic fragment containing the native promoter fully complements *ctr1-2*, which indicates that GFP-CTR1 is functional and likely regulated similarly to endogenous CTR1. Together, these results support the conclusion that ACC positively affects CTR1 protein abundance.

We carefully went over the text and corrected the protein and gene names corresponding to the associated experiments.

Reference:

Gao, Z., Chen Y.F., Randlett M.D., Zhao X.C., Findell J.L., Kieber J.J, and Schaller G.E. (2003) Localization of the Raf-like kinase CTR1 to the endoplasmic reticulum of Arabidopsis through participation in ethylene receptor signaling complexes. **J. Biol. Chem** 278: 34725-34732.

2. Inference made in this study primarily involved results from single transgenic lines. It should be noted that different transgenic events may give different effects due to differential transgene expression. Guidelines for several scientific journals require involvement of multiple (at least 3) transgenic lines when transgenes are involved [the guideline by New Phytologist as an example: Studies using transgenic organisms that employ a single transgenic primary event will not be considered (i.e. multiple independent lines are required)]. This concern was raised in my previous reviewing. The authors indicated in their rebuttal that several lines were investigated; however, the relevant data were not provided. Different expression levels conceivably affect the growth recovery kinetics, and the inference could be made differently due to different protein levels. For instance, the growth recovery kinetics differed greatly for *ctr1-8* (Fig. 4c) and CTR1p:GFP-CTR1-8/*ctr1-2* seedlings (supplementary Figure 12), indicative of the kinetics affected by CTR1- 8 levels. Another example is that the seedling growth recovery kinetics differed for CTR1p:GFP-CTR1/*ctr1-2* and WT (Supplementary Figure S12), even if the

transgene complements *ctr1-2*. Evidence for the growth recovery kinetics determined on the base of “time to reach pre-treatment growth rate” is only limited to a time point while the whole recovery curves left not compared, which are major part of the kinetics.

Response: For most native promoter-driven CTR1 transgenic lines, we initially confirmed the subcellular localization of WT CTR1 (or CTR1 variants) and protein expression using several independent T2 lines. Among those T2 lines, we continued to work with two independent homozygous lines per genotype to check their complementation of *ctr1-2* in the dark (presented the data in the main figures and supplemental information). Since two independent lines of most lines showed similar levels of *ctr1-2* complementation and phenotypes, we chose one line as a representative to perform ethylene response kinetics and stress responses. Similarly, several independent lines of various CTR1 overexpression (ox) lines were examined for their localization at T2 stages, all of which showed similar trends of ACC-induced subcellular localization of CTR1. Among those lines, we selected two independent lines, but mainly worked with one line for ethylene response kinetics and stress experiments, though we presented the results of two lines of Δ NTCTR1-1 ox (#14 and #28) for stress experiments. In this revision, we included the ethylene kinetics data for another independent line of 35S ox used in Fig. 4 (**Supplementary Fig. 10**). While there are some levels of variation in the recovery kinetics of two lines of some genotypes, probably due to the different expression levels of CTR1 variant proteins, both lines of each genotype showed similar slower recovery kinetics than the wild type. We found that the additional CTR1-1 ox line (#13 in **Supplementary Fig. 10d**) which showed slower kinetics than the other CTR1-1 ox line (#5), expresses significantly more CTR1-1 protein. To show that these additional independent lines also show similar stress responses, we examined the germination rate of WT CTR1 ox (#4, #5) and CTR1-1ox (#5 and #13) on plates with different NaCl concentrations (**Supplementary Fig. 18e**). As expected, additional lines of WT CTR1 ox and CTR1-1 ox showed a higher germination rate than WT. Furthermore, CTR1-1ox#13, which showed slower recovery kinetics than CTR1-1 ox#5, exhibited significantly higher germination rate than other lines, which is consistent with its slower recovery kinetics (**Supplementary Fig. 19**). Additionally, we included ethylene response kinetics of CTR1-8 ox lines (3 independent lines). All three CTR1-8 ox lines showed similar ethylene response kinetics to that of the wild type seedlings, which is congruent with its inability for nuclear translocation and WT-like stress responses. Based on the correlation between recovery kinetics and stress responses, these independent lines of each genotype of 35S lines likely show similarly enhanced stress responses on soil conditions compared to the wild type. We acknowledge the reviewer’s concerns. However, our studies also extensively analyzed various transgenic lines expressing CTR1 variants in different genetic backgrounds and various overexpression lines to systemically investigate the nuclear role of CTR1 and the nuclear translocation mechanism. We believe that this is a powerful approach to revealing the role of a protein of interest in plants. The analysis of all these lines collectively pointed out that the nuclear localization of CTR1 is regulated by ACC/ethylene and that nuclear-localized CTR1 positively influences stress resilience to drought and salinity stresses. Together with the ample amount of biochemical results presented, we feel that our data is solid and consistent with the conclusion of the manuscript.

Regarding *ctr1-8* versus *CTR1p:GFP-CTR1-8/ctr1-2* seedlings, the *ctr1-2* mutant does not have any changes in ethylene response and recovery kinetics before and after ethylene removal due to its constitutive active ethylene response (high EIN3 levels). Therefore, addition of the *ctr1-8* transgene would restore the response kinetics of the *ctr1-8* by inhibiting EIN2 at the ER but would not completely rescue the recovery kinetics of the *ctr1-8* due to the preexisting high levels of EIN3 in the nucleus. For wild type versus *CTR1p:GFP-CTR1/ctr1-2*, we agree that the difference could be due to the transgene, but these transgenes are in a different genetic context. As such, there are multiple factors that could be affecting response kinetics. The main point of supplemental figure 12 is to show that the wild-type transgene substantially rescues recovery kinetics, whereas the *ctr1-8* transgene does not.

Regarding time for recovery after removal of ethylene: We analyzed recovery time as suggested in the initial reviews. By definition, a recovery time is a single time. We did it in two ways. One is to determine the time to pre-treatment growth rates for each seedling, which is what we included in the revised manuscript. The other is to determine the time of recovery based on the time to maximal growth rate after the removal of ethylene. This second method gave the same results as far as determining which were significantly different from wild type, and therefore, we did not include this. Since our response in the previous rebuttal seems to be insufficient for the reviewer's expectation, we have now also determined the rate of growth recovery based on a linear regression of each seedling's growth recovery between times 3.5 and 4.75 h to provide another measure of recovery, and this too showed the same lines were statistically different from wild type. We have not included this analysis since it seems redundant and gives the same result.

3. I am concerned about the LSCM images for the GFP-CTR1 nuclear transport that could be resulted from a focal plan shift. For instance, the images for Supplementary Figure S6 clearly revealed different focal plans. The cell outlines and the circled (red) nucleus change or disappear (see below).

Response: As we mentioned in the previous rebuttal letter, in our experimental conditions where seedlings are in aqueous solution, it is extremely difficult to get images of the same seedlings without any minimal alteration of their positions, even if we manually adjust the focal plane. Thus, we included the Z-stacks to minimize focal plane changes and to pinpoint the cells that do not have nuclear-localized GFP-CTR1 and traced the cells to observe CTR1 localization changes over time. Cells are dynamic living entities with constant changes in cytoplasmic activities such as microtubule organization and organelle-organelle interaction on top of cell-to-cell variations. Some of the variation in the images may be attributed to these factors in addition to the fact that ER exclusively surrounds the nucleus. The main idea of the experiments is to examine the kinetics of CTR1 translocation into the nucleus, and the data suggests that it happens within 30 minutes in our experimental setting.

4. The authors did not truly answer for why the elevated EIN3 levels of 35S:GFP- Δ NT-CTR1-1 unable to produce a typical seedling phenotype, although a vague speculation is proposed. This comment also applies to data from Supplementary Figure 18, where EIN3 levels are highly induced by GFP-CTR1-1, GFP- Δ NTCTR1-1, and GFP-CTR1-8, and the plants did not exhibit a

prominent ethylene growth inhibition phenotype as expected. Results and descriptions for the last section (starting from line 314) are somehow difficult to follow, without the information for the genetic background of the transgenic lines. I assume they are in the WT background if not specified, but I am not sure if my assumption was correct.

Response: Please see the response above to this question. In addition to the aforementioned response, we want to correct that *35S:GFP-CTR1-8* does not express high levels of EIN3. It expressed moderately similar levels of EIN3 as wild-type plants in our immunoblot quantification (**Supplementary information 18**), which explains the similar plant size and levels of stress response to the wild-type plants. We also included ethylene response kinetics of *35S:GFP-CTR1-8*, which showed comparable ethylene response and recovery kinetics to the wild type (**Supplementary information 19**). For a description of the transgenic lines used in the text, we mentioned most overexpression lines by adding "35S" to the name of the lines. All overexpression lines are expressed in the wild-type background. We indicated this in the materials and methods in this revision.

5. Lines 305-307. The description "The hypersensitive ACC response of CTR1p:GFP- Δ NT-CTR1-1/*ctr1-2* as compared to CTR1p:GFP-CTR1-1/*ctr1-2*..." lacks a control (*ctr1-2*) (Fig. 5h). It has been shown that *ctr1-2* seedlings are ethylene responsive. If *ctr1-2* and CTR1p:GFP- Δ NT-CTR1-1/*ctr1-2* seedlings exhibit a similar response curve to ACC, the GFP- Δ NT-CTR1-1 protein would have little effect on the ethylene response, as oppose to the propose positive role in ethylene responses.

Response: We performed this experiment to examine the effect of truncated, constitutive nuclear-localized CTR1 over ethylene-regulated full-length CTR1 on ethylene response using native promoter-driven expression lines to remove the effects of ectopic expression of the CTR1 proteins in plants. We understand the reviewer's concerns. However, the effects of the ethylene responsiveness of *ctr1-2* are already equally applied to both lines as a common genetic background. Therefore, we think that the current data is sufficient to provide information on the positive effects of constitutive nuclear-localized CTR1 on ethylene responses.

Minors or optional:

1. This was raised in my previous comment. The laser scanning confocal imaging (LSCM) for EIN2-mCherry localization to the nucleus is somehow inconclusive (Supplementary Figure S5). The fluorescence primarily appeared a networking structure, reminiscent of the ER network that surrounds the nucleus. Alternatively, the indicated fluorescence might not represent a nucleus (the indicated nucleus appeared too large to be a nucleus). This could arise from focal plane shift as mentioned in my previous comment, inferred from the different shape for adjacent cells of the four images. I was able to observe an empty nucleus (blue circles; see below) in the first three images, and the nucleus disappeared for the 60 min image, implying a focal plan shift. The kinetics for EIN2 nuclear movement should not be determined from a single cell for the lack of evidence power for cell biology studies. Perhaps, evidence for the EIN2 movement is not essential to this work and could be removed because similar results have been reported in previous studies (Qiao et al., 2012).

Response: We removed Supplemental Figure 5 as advised.

2. The way for presenting those CTR1 variants was confusing. For examples, CTR1- 1 was presented as CTR1CTR1-1 and CTR1-8 as CTR1CTR1-8. Given that the CTR1-1 protein is encoded by the ctr1-1 gene, presenting the protein as CTR1-1 is sufficient and clear.

Response: Thank you for the suggestion. We changed the nomenclature of the mutant CTR1 proteins as suggested throughout the text.

3. Line 387 “CTR1 lacks a canonical NLS”. This statement may not be true. Sequence analyses (http://nls-mapper.iab.keio.ac.jp/cgi-bin/NLS_Mapper_form.cgi#opennewwindo) for the GFP-fused CTR1 variants revealed several peptide sequences that facilitate CTR1 localization in the cytoplasm and nucleus (see below). These may explain the dual localizations of CTR1. On the other hand, it is to be explained for why CTR1-8 is not localized to the nucleus.

Response: Thank you for providing this information. We were also aware of the potential NLSs of CTR1 predicted by the cNLS mapper and have already tested the highest scoring peptide for its role as an NLS. The cut-off score of 3-4 is relatively relaxed, which includes weak NLS peptides. However, we selected the peptides with the highest score (3.9) and tested whether the peptide sequences drive the nuclear localization of 3x and 3xGFP by fusing the NLS sequences. The results showed that it does not change the localization ratio (cytosol vs nucleus) of both 3x GFP and 3xGFP-NLS, which indicates it is not a functional NLS itself. Furthermore, none of the predicted peptides include G354, which is substituted for E in the CTR1-8 protein. Therefore, we also excluded the possibility that it might affect the nuclear localization of CTR1-8. Based on these results, we indicated that CTR1 does not have canonical NLS.

4. I appreciated that the authors were able to answer my concern for the CTR1 protein localization involving polyclonal antibodies for CTR1 (previous comment 5). GE Schaller previous showed localization of CTR1-8 to the cytoplasmic soluble fraction, with the total CTR1-8 level unchanged. Results from the western blot for CTR1-8 appeared odd. The western blot by Park and co-authors showed a greatly reduced level for the cytosol CTR1-8, as compared to the level from the total protein (Fig. 2i). By contrast, CTR1/CTR1-1 levels are similar for proteins from the total and cytoplasmic fraction. The cytoplasmic fraction did not involve ultracentrifugation that removes the membrane fraction, and the cytoplasmic fraction may represent proteins from the membrane and soluble fractions. Can the authors explain the results? The authors may need to describe in methods for how the total protein was prepared.

Response: We are sorry about the quality of the immunoblot. The reason that the cytosol fraction of CTR1-8 protein is lower than the level of the total fraction is likely due to the unequal loading of the cytosolic fraction as shown in the BIP control. Because of the scarcity of CTR1 antibody, we could not repeat the experiment for better quality. Nonetheless, the blot clearly showed that CTR1-8 does not translocate into the nucleus in response to ACC (the loading of nuclear fraction is relatively consistent among different treatment). We included information for protein extraction for the fractionation experiments in the materials and methods.

5. Western blots for CTR1 (Supplementary Figure S4) are a critical piece of evidence for nuclear localization of a small fraction of the total CTR1 and should be presented in Figure 1 described in the first section (Ethylene-induced ER-to-nucleus translocation of CTR1). Placing

the data in Supplementary Figure S4 may not draw sufficient attention when the amount of data is overwhelming.

Response: We included Supplemental Figure S4 to the main text (Fig. 1j).

7. This was raised in my previous comment. The LSCM study for localization of the several different CTR1 variants involved silver or ACC treatment. Presenting localizations under the ambient condition (non-treated) is equally important as a control. However, this control (no treatment) is missing, except for that for western blots. The authors replied that there is no need for the control. Without the control, we cannot judge the respective effects of silver and ACC on CTR1 localization and abundance.

Response: To address this, in previous responses, we provided six representative lines that show different ACC-induced localization of various CTR1 (WT CTR1, CTR1-8, Δ NT-CTR1) expressed from 35S and native promoter in the requested condition (no ACC, with ACC, with silver treatment) in Supplementary Fig.2. These results are consistent with the immunoblotting results, demonstrating that CTR1 nuclear movement is regulated by ACC/ethylene and the *ctr1-8* mutation inhibits the translocation of CTR1-8 protein.

7. I completely agree with the rebuttal for presenting the data in percentage for samples with a size much smaller than 100, where the small sample size produces a lower power of evidence. It is exactly my concern that whether the reduced power of evidence is sufficient to make a conclusion.

Response: Thank you for the comments.

8. Evidence for the conclusion for GFP-CTR1 decrease after ethylene removal appear weak (Supplementary Figure S7). The extent of GFP-CTR1 reduction determined by western blots (Supplementary Figure S7) and LSCM (Fig. 1i) differed greatly. I agree that there appears to have a marginal change; however, the change is too marginal to be conclusive given that western blots can be affected by many factors and are not sufficiently sensitive/stable to detect marginal changes in protein levels. Even if the change is real, contribution of the marginal change to a biological activity is unlikely determined.

Response: We speculate that the potential reason for the discrepancy between LSCM (Fig.11 (previous Fig. 1i)) and western blot is probably the difference in sample preparation and unequal loading of the samples based on the loading controls. For LSCM, we performed paired imaging using an identical seedling over time, whereas the immunoblotting was done with seedlings harvested at different time points after ethylene removal, as it is not possible to use the same seedling to check the alteration of the protein levels. To minimize the discrepancy between LSCM (Fig.1i) and western blot, we repeated the experiments and provided new data with better loading control.

9. The seedling growth recovery by GFP-CTR1-8-SV40 is better tested in *ctr1-2* but not *ctr1-8* background.

Response: We used this line to examine the effect of constitutively localized CTR1-8 protein expressed from a native promoter on stress response by comparing it to that of CTR1-8 because CTR1-8 does not translocate to the nucleus. Therefore, using *ctr1-8* background is

reasonable for the purpose of the experiment as it allows us to investigate the effect of nuclear-localized CTR1-8 on stress response without any potential consideration derived from using a different genetic background. The suggested line would also be a good line to test the role of nuclear CTR1, but it could be tested in future studies as such a line is not available at the moment.

10. I suggest the authors use CTR1 polyclonal antibodies for western blots involving GFP-fused CTR1 variants, instead of using the GFP antibodies. This may detect both the endogenous CTR1 and the ectopically expressed GFP-CTR1 variants, facilitating the revealing of levels of the two types of proteins.

Response: We already provided fractionation analyses that endogenous CTR1 and GFP-CTR1 behave in an identical manner when responding to ACC and silver nitrate and thus feel that using CTR1 antibodies would provide no additional information that strengthen our conclusion of the study.

11. Western blots show a much lower EIN3 level for ACC-treated WT were than for non-treated *ctr1-8* plants (Fig. 5e). These results appear abnormal, as *ctr1-8* is hylomorphic. Regardless, EIN3 levels for WT were still much lower than that for *ctr1-8* plants by the 10 μ M ACC treatment.

Response: We think the upward smearing streaks of the EIN3 bands in *ctr1-8* make the EIN3 bands appear more stronger than their actual intensity. Multiple repeats of *ctr1-8* and WT immunoblots showed that *ctr1-8* at zero ACC showed generally similar levels of EIN3 as that of WT at 10 μ M ACC. We replaced the blot with another replicate that more clearly shows this. Several repeats of this experiments consistently showed that *ctr1-8* at 0 μ M ACC expressed similar levels of EIN3 to that expressed in WT at 10 μ M ACC treatment. This is odd as indicated by the reviewer, as *ctr1-8* with no ACC treatment displayed longer hypocotyls than WT with 10 μ M ACC. As we previously mentioned, we don't know the precise mechanism behind this. We think this appears to be similar to what we have observed for lines overexpressing CTR1-1 and Δ NTCTR1. Given that CTR1-8 protein is not translocating into the nucleus, one of the speculations is that the increased protein abundance of CTR1-8 protein in the cytoplasm may interact with other cellular processes that alleviate the effect of EIN3 on hypocotyl growth and plant size.

Reviewer #3 (Remarks to the Author):

Results from this study draw a picture for possible roles of CTR1 in the different subcellular compartments, extending our knowledge outside the present framework for roles of CTR1 in ethylene signaling. This may lead to investigations and findings to an unexplored territory. Despite of the ambiguity as mentioned in the previous comments for some results, part of which was not fully answered, the uncertainty is acceptable because experiments help unveil truth from different perspectives and may not be necessarily perfect. It is thus important to rigorously contrast positive and negative evidence or opinions to draw interpretations, instead of presenting positive opinions only to give an answer. The authors have answered most of my concerns, and I only have a few comments to this manuscript. No new experiments will be needed.

1) I agree with the statement that CTR1 unlikely has a canonical NLS peptide; however, based on evidence described in the rebuttal, the possibility for presence of non-canonical NLS peptides cannot be excluded. I suggest several possibilities to be fully discussed in the Discussion section, by contrasting positive and negative evidence. In addition to the presence of several peptides with a relaxed score, the authors may argue that CTR1-8 is cytoplasmic, favoring the argument for absence of non-canonical NLSs.

2) This work did not investigate roles of wild-type CTR1 in salt resistance, and the authors revised the last section, focusing on roles the kinase-defective CTR1 variants in stress resistance. Because experiments in this section primarily involved kinase-defective CTR1 but not wild-type CTR1, I agree with the interpretation that the stress resilience appears to be independent of the kinase activity. The inconsistency is that the Co-IP experiment involved the wild-type CTR1 but not the kinase-defective CTR1-1 (Fig. 5C), implying a role of the wild-type CTR1 in interacting with or regulating EBF2. This agrees with the result that CTR1-8-SV40 *ctr1-8* plants produced higher levels of EIN3, exhibited delayed growth recovery, and were saline resistant (according to previous studies by J Kieber, CTR1-8 is presumably kinase active). These added uncertainty about the relevance of CTR1 kinase activity to the stress resistance induced in the nucleus. Perhaps, in-depth discussions will be needed to address this ambiguity. For instance, the excess amount of CTR1 or CTR1-8 may not be effectively activated by the ethylene receptors, and, presumably, they are thus kinase-inactive and able to regulate EBF2. This scenario may explain the how the ethylene-induced CTR1 transport to the nucleus may negatively regulate EBF2, a question raised in my previous comment. In other words, upon ethylene treatment, the ethylene receptors cannot activate CTR1, and the transport of the kinase-inactive CTR1 to the nucleus may lead to the fine-regulation of EBF2. It is equally important to state that the kinase activity for CTR1 is undetermined upon ethylene treatment. Nevertheless, the present evidence favors the model presented by the authors.

3) As for the inconsistency for the elevated EIN3 level and weaker constitutive ethylene response phenotype, I suggested the authors report the issue and leave the question open, instead of giving a highly speculative explanation (lines 300-302).

4) This is optional. As mentioned in my previous comment, mutant version of CTR1 can be presented in a concise way. For examples, CTR1-8 instead of CTR1*ctr1-8*, and CTR1-1 instead of CTR1*ctr1-1*.

5) The prominent salt resistance conferred by those CTR1 variants may not truly reflect effect of the endogenous levels of CTR1. In the rebuttal, the authors stated that the magnitude of contribution to salt resistance by the endogenous level of CTR1 cannot be determined. The present data do support the proposed model; meanwhile, the uncertainty may need to be fully discussed.

Reviewer #3 (Remarks to the Author):

Results from this study draw a picture for possible roles of CTR1 in the different subcellular compartments, extending our knowledge outside the present framework for roles of CTR1 in ethylene signaling. This may lead to investigations and findings to an unexplored territory. Despite of the ambiguity as mentioned in the previous comments for some results, part of which was not fully answered, the uncertainty is acceptable because experiments help unveil truth from different perspectives and may not be necessarily perfect. It is thus important to rigorously contrast positive and negative evidence or opinions to draw interpretations, instead of presenting positive opinions only to give an answer. The authors have answered most of my concerns, and I only have a few comments to this manuscript. No new experiments will be needed.

1) I agree with the statement that CTR1 unlikely has a canonical NLS peptide; however, based on evidence described in the rebuttal, the possibility for presence of non-canonical NLS peptides cannot be excluded. I suggest several possibilities to be fully discussed in the Discussion section, by contrasting positive and negative evidence. In addition to the presence of several peptides with a relaxed score, the authors may argue that CTR1-8 is cytoplasmic, favoring the argument for absence of non-canonical NLSs.

Response: Thank you for the comments. We have included suggested discussion in the text (line 430-433).

2) This work did not investigate roles of wild-type CTR1 in salt resistance, and the authors revised the last section, focusing on roles the kinase-defective CTR1 variants in stress resistance. Because experiments in this section primarily involved kinase-defective CTR1 but not wild-type CTR1, I agree with the interpretation that the stress resilience appears to be independent of the kinase activity. The inconsistency is that the Co-IP experiment involved the wild-type CTR1 but not the kinase-defective CTR1-1 (Fig. 5C), implying a role of the wild-type CTR1 in interacting with or regulating EBF2. This agrees with the result that CTR1-8-SV40 ctr1-8 plants produced higher levels of EIN3, exhibited delayed growth recovery, and were saline resistant (according to previous studies by J Kieber, CTR1-8 is presumably kinase active). These added uncertainty about the relevance of CTR1 kinase activity to the stress resistance induced in the nucleus. Perhaps, in-depth discussions will be needed to address this ambiguity. For instance, the excess amount of CTR1 or CTR1-8 may not be effectively activated by the ethylene receptors, and, presumably, they are thus kinase-inactive and able to regulate EBF2. This scenario may explain how the ethylene-induced CTR1 transport to the nucleus may negatively regulate EBF2, a question raised in my previous comment. In other words, upon ethylene treatment, the ethylene receptors cannot activate CTR1, and the transport of the kinase-inactive CTR1 to the nucleus may lead to the fine-regulation of EBF2. It is equally important to state that the kinase activity for CTR1 is undetermined upon ethylene treatment. Nevertheless, the present evidence favors the model presented by the authors.

Response: In our study, we included the inactive version of CTR1 to investigate the role of kinase activity in CTR1's nuclear translocation and stress responses. Ethylene is known to

inactivate the ethylene receptors that regulate CTR1 activity, so we speculate that the inactivation of CTR1 may contribute to the stress tolerance observed in plants expressing the active form of CTR1 (CTR1-8-SV40) under conditions of salinity and drought, as noted by the reviewer. We were able to confirm the interaction between the inactive CTR1 and EBF2 using yeast two-hybrid analysis, but it is still not clear how CTR1 activity is regulated during an ethylene-mediated stress response. We have addressed this issue in more detail in our revised manuscript (lines 395–400).

3) As for the inconsistency for the elevated EIN3 level and weaker constitutive ethylene response phenotype, I suggested the authors report the issue and leave the question open, instead of giving a highly speculative explanation (lines 300-302).

Response: As suggested, we left the question open.

4) This is optional. As mentioned in my previous comment, mutant version of CTR1 can be presented in a concise way. For examples, CTR1-8 instead of CTR1ctr1-8, and CTR1-1 instead of CTR1ctr1-1.

Response: We agreed that the suggested format would be a more concise way to name the CTR1 mutants. Therefore, we made the necessary changes to the protein name of the mutant CTR1 in the previous response. However, we kept the superscripted, lowercase format for transgenic lines with CTR1 mutations.

5) The prominent salt resistance conferred by those CTR1 variants may not truly reflect effect of the endogenous levels of CTR1. In the rebuttal, the authors stated that the magnitude of contribution to salt resistance by the endogenous level of CTR1 cannot be determined. The present data do support the proposed model; meanwhile, the uncertainty may need to be fully discussed.

Response: We added a related discussion in the text (line 416-420).